# Olfactory marker protein directly buffers cAMP to avoid depolarization-induced silencing of olfactory receptor neurons

Noriyuki Nakashima[1,2 ✉], Kie Nakashima[2,3], Akiko Taura[4,5], Akiko Takaku-Nakashima[1,6], Harunori Ohmori[2,7] & Makoto Takano[1]

Olfactory receptor neurons (ORNs) use odour-induced intracellular cAMP surge to gate cyclic nucleotide-gated nonselective cation (CNG) channels in cilia. Prolonged exposure to cAMP causes calmodulin-dependent feedback-adaptation of CNG channels and attenuates neural responses. On the other hand, the odour-source searching behaviour requires ORNs to be sensitive to odours when approaching targets. How ORNs accommodate these conflicting aspects of cAMP responses remains unknown. Here, we discover that olfactory marker protein (OMP) is a major cAMP buffer that maintains the sensitivity of ORNs. Upon the application of sensory stimuli, OMP directly captured and swiftly reduced freely available cAMP, which transiently uncoupled downstream CNG channel activity and prevented persistent depolarization. Under repetitive stimulation, $OMP^{-/-}$ ORNs were immediately silenced after burst firing due to sustained depolarization and inactivated firing machinery. Consequently, $OMP^{-/-}$ mice showed serious impairment in odour-source searching tasks. Therefore, cAMP buffering by OMP maintains the resilient firing of ORNs.

[1] Department of Physiology, Kurume University School of Medicine, 67 Asahi-machi, Kurume, Fukuoka 830-0011, Japan. [2] Department of Physiology and Neurobiology, Faculty of Medicine, Kyoto University, Yoshida-Konoe, Sakyo-ku, Kyoto 606-8501, Japan. [3] Laboratory of Developmental Neurobiology, Graduate School of Biostudies, Kyoto University, Yoshida Hon-machi, Kyoto 606-8501, Japan. [4] Department of Otolaryngology, Head and Neck Surgery, Graduate School of Medicine, Kyoto University Hospital, 54 Kawaracho, Shogoin, Sakyo-ku, Kyoto 606-8507, Japan. [5] Department of Medical Engineering, Faculty of Health Science, Aino University, 4-5-4 Higashioda, Ibaraki, Osaka 567-0012, Japan. [6] Post Graduate Training Program, The University of Tokyo Hospital, 7-3-1, Hongo, Bunkyo-ku, Tokyo 113-8655, Japan. [7] Department of Physiology, School of Medicine, Kanazawa Medical University, 1-1 Daigaku, Uchinada, Kahoku, Ishikawa 920-0293, Japan. ✉email: nakashima_noriyuki@med.kurume-u.ac.jp

Sensory sensitivities are regulated by balancing maximal amplitude and temporal resolution of cellular responses, depending on stimulus patterns. Adaptation adjusts the basal states of cells by reducing sensitivities to constant stimuli over time in signal transduction cascades. Furthermore, sensitivities to repetitive stimuli are distinctly maintained by regulating response kinetics in sensory cells[1–15].

Among the unique sensations, olfactory receptor neurons (ORNs) show prominent adaptive capacity that enables the cancellation of environmental smells[7–14]. Odourant-induced cAMP surges occur in small ciliary compartments and effectively activate cyclic nucleotide-gated (CNG) $Ca^{2+}$-permeable channels. The resultant $Ca^{2+}$ accumulation leads to feedback inhibition of CNG channels via $Ca^{2+}$/calmodulin[7–9,15–17]. In addition to inhibition of CNG channels, ORNs undergo adaptation by modulating olfactory signalling-associated molecules, including calmodulin-dependent kinase, adenylate cyclase (AC) and phosphodiesterase (PDE)[7,9–12,18].

Meanwhile, odour identification among background odours and odour-source searching requires sufficient sensitivity[18–23]. During olfactory behaviour, sniffing causes adaptation in a frequency-dependent manner, as high-frequency sniffing attenuates olfactory responses[24]. Repetitive olfactory neural responses have been proposed to require the rapid termination of CNG channel activity to avoid subsequent feedback adaptation in the cilia upon the application of each stimulus[7–9,14,16,17,25–32]. Recently, some odourant receptors have been demonstrated to simultaneously sense mechanical stimuli in addition to odourants through a common signalling process that mobilises the $G\alpha_{olf}$ protein and ACIII, resulting in cAMP production as a second messenger[33–35]. This process is important for sniff-coupled temporal coding[20]. ORNs should reconcile sensitivity and adaptation under the continual application of sensory stimuli—e.g., during repetitive sniffing—although the cellular mechanisms involved in this process have not been elucidated.

Here, we reveal the identity of olfactory marker protein (OMP), a histological marker of mature ORNs[36], as a cAMP-binding protein. We discovered that OMP plays a distinct cytosolic role in olfactory signal transduction by capturing surplus cAMP in parallel with conventional signal transduction and sharpening CNG channel activity under sensory stimuli. We further demonstrated that the responses of ORNs to repetitive sensory stimulation were rapidly silenced without this cAMP-buffering mechanism by OMP.

## Results

**OMP binds directly to cAMP.** Despite various hyposmic phenotypes demonstrated by mice lacking OMP, no direct interactions have been identified between OMP and cAMP signalling proteins[26–32]. Upon analysing the amino acid sequence of OMP[36–38], we identified a cyclic nucleotide-binding (CNB) domain-like motif[39] (Fig. 1a: boxed; Supplementary Fig. 1a–f). The docking simulations predicted that OMP possesses two CNB pockets, i.e., pockets A and B, that have free energy of binding (ΔG) values of −36.8 kJ/mol and −29.7 kJ/mol for cAMP, respectively (Fig. 1b–d; Supplementary Movies 1, 2). The amino acid residues comprising these pockets of OMP are highly preserved across vertebrates (Fig. 1e–l; Supplementary Fig. 1a–j).

We assessed the direct physical interaction between OMP and cAMP in vitro by performing bioluminescence resonance energy transfer (BRET) experiments, in which luminescence energy from luciferase was transferred to fluorescent molecules in very close proximity (Fig. 2a, b). Renilla luciferase (Rluc, 36 kDa, energy donor, emission peak at 480 nm) was fused to OMP (Rluc-OMP, 55 kDa; Supplementary Fig. 2a, b) such that BRET would occur in

tight proximity to fluorescent 8NBD-cAMP (energy acceptor, emission peak at 536 nm) (Fig. 2b; Supplementary Fig. 2a, b). Upon the addition of the luciferase substrate coelenterazine, wild-type Rluc-OMP luminesced with a single spectral peak at 480 nm (Ctrl in Fig. 2c); the background thermal noise was subtracted (Supplementary Fig. 3a, b). Another peak gradually emerged at 536 nm in a dose-dependent manner after the addition of 8NBD-cAMP, indicating that BRET occurred between Rluc-OMP and 8NBD-cAMP (Fig. 2c, sequentially from Ctrl to 10 μM). This peak-to-peak ratio was defined as the total BRET ratio (Fig. 2c). For analysis, the relative shift in the BRET ratio after subtracting the nonspecific BRET ratio was redefined as the ΔBRET signal of Rluc-OMP$^{Wt}$ (Wt, Fig. 2d; Supplementary Fig. 3c–g). The ΔBRET signal was saturated with 8NBD-cAMP at concentrations of 5 μM or higher (Fig. 2d) and decreased dose-dependently by the addition of a competitor cAMP (ΔΔBRET, Fig. 2b, e), yielding an apparent cAMP-binding affinity of −37.3 kJ/mol for Rluc-OMP$^{Wt}$ (see 'Methods' and Supplementary Fig. 3g–i). The time required to reach a steady state after 8NBD-cAMP application in the BRET assay was approximated by an association time constant of 1.5 s (Supplementary Fig. 3j).

We next created fusion proteins using the mutant OMP with up to three amino acid changes at important residues in pockets A and B (pocket A, G126E; pocket B, R136E) or the interpocket wall (pockets A/B, D90E) (Fig. 1e–h, Fig. 2f; Supplementary Fig. 2a, b). The cytosolic expression of these Rluc-OMP proteins was confirmed by immunohistochemistry analysis of OMP (Fig. 2g), and the size was confirmed by western blotting (Fig. 2h). The ΔBRET signal dropped over a high-affinity range (0.1–1 μM) for all mutants (green box in Fig. 2i) and over a low-affinity range (2–10 μM) for all mutants, except Rluc-OMP$^{G126E}$ (yellow box in Fig. 2i). Therefore, pocket A-mutated Rluc-OMP$^{G126E}$ retained the low-affinity interaction, whereas the pocket B mutants lost their interaction with 8NBD-cAMP (* in Fig. 2i), suggesting that pocket A has a conventional CNB motif that binds cAMP at a higher affinity than pocket B (Supplementary Fig. 1g–j), and that pocket B possibly contributes to steric interactions with pocket A or the overall conformational changes of OMP[37,38].

**OMP directly buffers cytosolic cAMP actions on CNGA2 channels.** Odourants evoke phasic cAMP production in the cilia of ORNs, where CNG channels are localised[7–9,11,14,15,17,40,41]. Previous reports showed that the odourant stimulation of OMP-KO ORNs induces prolonged CNG currents, causing strong adaptation and delaying the recovery of CNG channel activity[26–28,31,32]. Therefore, OMP in the cilia has been inferred to interrupt AC activity for the rapid termination of cAMP signalling. However, a direct interaction between OMP and AC has not been proven[1,31,32].

We attempted to measure the activity of CNG channel subunit A2 (CNGA2) in the presence of OMP because CNGA2 is suitably insensitive to $Ca^{2+}$ feedback adaptation and remains open when cytosolic cAMP is elevated[9,16,42].

We transfected the designed plasmids into HEK293T cells (Fig. 3a). Successfully transfected cells were fluorescently visualised with GFP and DsRed together with OMP and CNGA2 expression, respectively (Fig. 3b; Supplementary Fig. 2a, b). Channel activity was observed only in HEK293T cells expressing CNGA2 (Supplementary Fig. 4a–d) and was sensitive to the application of an AC activator, forskolin (Supplementary Fig. 4e–j).

The stepwise single-level channel activity of CNGA2 was confirmed in a cell-attached recording held at −80 mV (Fig. 3c). The unitary conductance estimated from these current signals was similar to that previously reported[40] (Supplementary Fig. 4a, c, d).

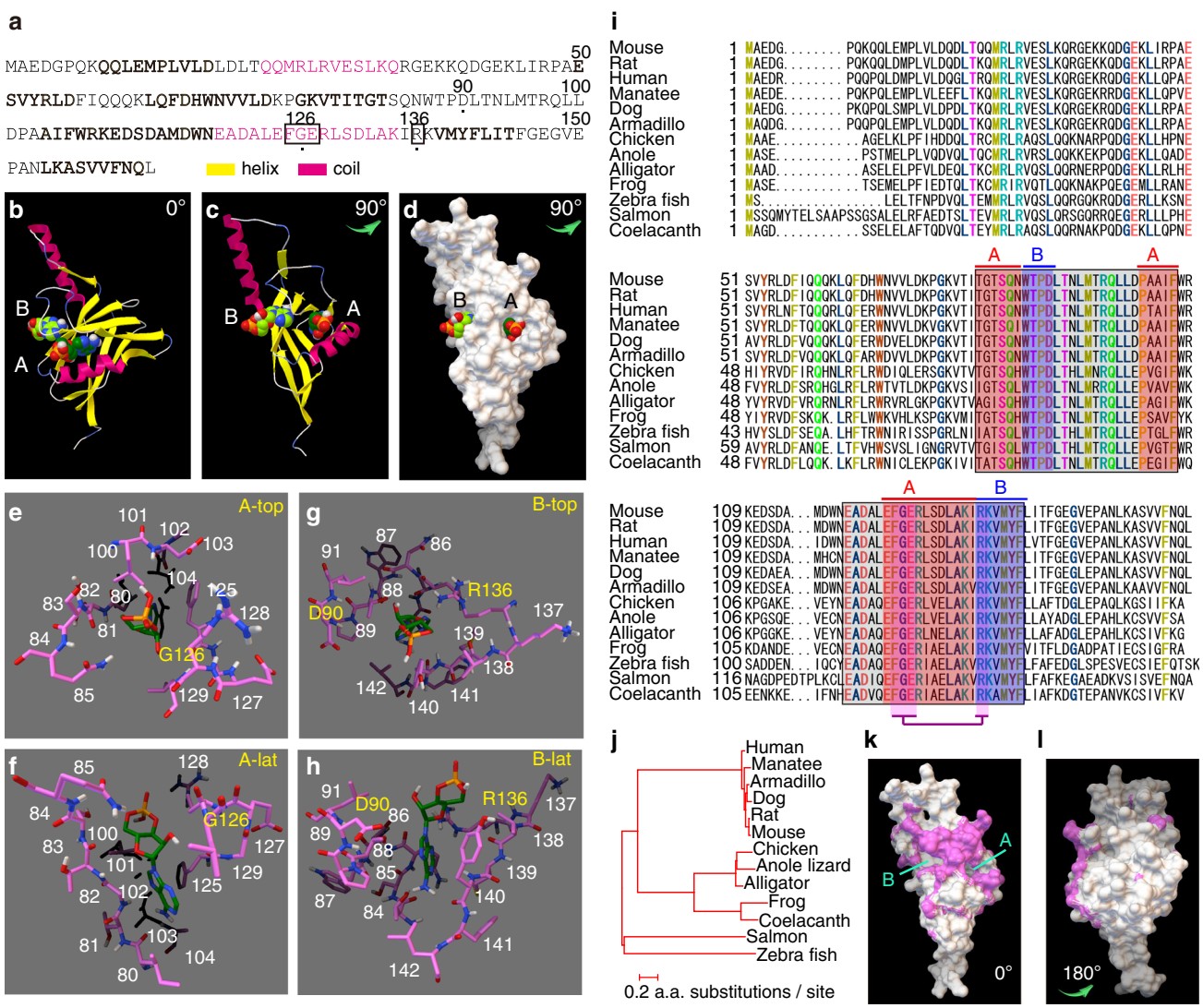

**Fig. 1 OMP and cAMP docking simulation. a** Primary structure of *M. musculus* OMP. **b**, **c** Docking results of OMP (ribbons) and two cAMP molecules (balls). **d** The calculated surface electron clouds of OMP are shown in white with two cAMP molecules docked. **e**–**h** cAMP and the surrounding residues at sites A and B viewed from the top (**e**, **g**) and laterally (**f**, **h**). The residues shown in yellow were mutated in the following experiments. **i** Comparison of OMP primary structures among different vertebrate species. The highly conserved residues are indicated with coloured letters. Residues comprising sites A and B are highlighted in red and blue, respectively. The classical consensus sequence of CNBD is highlighted by a purple box. **j** Evolutionary relationships of OMP. **k**, **l** The conserved residues on the surface of the OMP space-filled structure on the face (**k**) and back (**l**) are coloured pink.

We then recorded the whole-cell CNG currents from the transfected cells under voltage clamp. The cells were filled with photolytic caged cAMP (1 mM) upon rupture of the patch membrane, and were held at −80 mV. Inward CNG currents were induced by a 1-s UV flash. The presence of OMP did not significantly affect CNGA2 activation kinetics (245 ± 34 ms and 299 ± 54 ms; mean ± s.e.m., with and without OMP, $n = 16$ and 12, respectively; two-sided unpaired $T$ test, $P = 0.38$). CNG currents 3 s after the peak showed only a slight decay without OMP (light purple, Ctrl, Fig. 3d, e). With OMP, the CNG currents rapidly decreased with a decaying time constant of 1.67 ± 0.35 s (mean ± s.e.m., $n = 11$, light blue; OMP ( + ), Fig. 3d, e).

In the presence of a PDE inhibitor (PDEi), the activation time constants of CNG currents were unaffected with (266 ± 55 ms) or without OMP (292 ± 86 ms, mean ± s.e.m., $n = 4$ each; unpaired $T$ test, $P = 0.78$, Fig. 3f, g). The CNG channel currents were sustained for 3 s in the absence of OMP (Ctrl, Fig. 3d–g). In contrast, CNG currents were similarly deactivated in the presence of OMP, although some residual CNG currents remained at 3 s in

the presence of PDEi. This result may indicate that OMP interacted with basal cAMP, which was increased by PDEi, leading to a diminished cAMP-buffering capacity. The deactivation time constant was only slightly slower (2.26 ± 0.49 s, mean ± s.e.m., $n = 4$) than that observed in the absence of PDEi (two-sided unpaired $T$ test, $P = 0.17$). Furthermore, the time constant was comparable with the association time constant predicted by the BRET assay (1.5 s, Supplementary Fig. 3j).

These results suggest that OMP swiftly eliminates cAMP to deactivate CNG channels; furthermore, the cAMP-buffering capacity of OMP depends on the size of the basal cAMP pools, and a substantial population of OMP might be pre-equilibrated. Therefore, OMP alone could sequester cAMP surges. This situation can be explained by a simple kinetic model, where OMP forms an independent reserve compartment to buffer cAMP near the membrane compartment wherein cAMP is initially produced ($C_1$ and $C_2$ in Fig. 3h, i), based on the assumption that the cAMP elimination rate by PDE is slower than cAMP buffering by OMP, and that the speed of cAMP

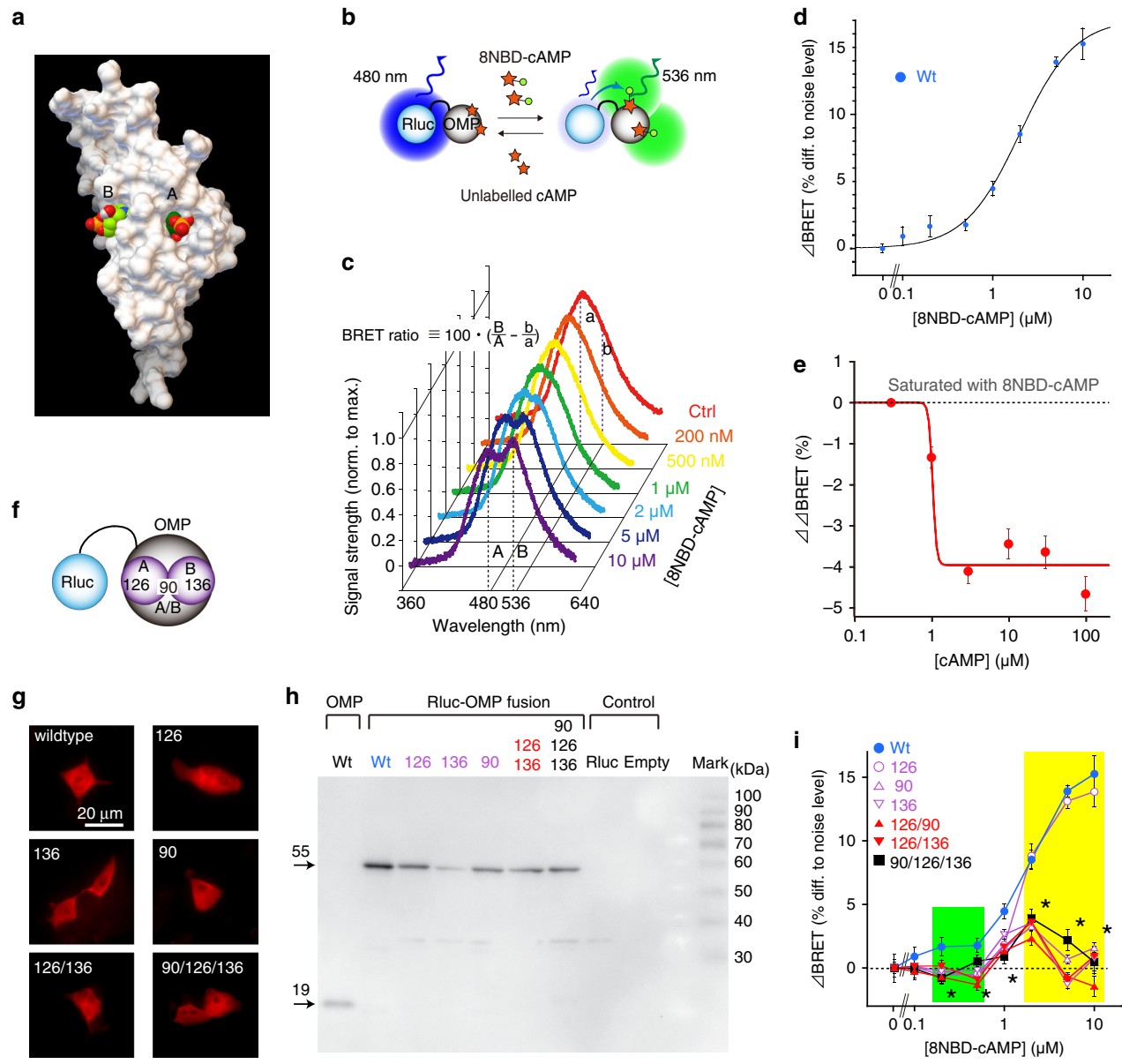

**Fig. 2 BRET assay for assessing direct interactions between OMP and cAMP. a** Computer-simulated docking conformation between cAMP and OMP. **b** Schematic diagram of BRET between Rluc-OMP and 8NBD-cAMP and a competitive assay in addition to unlabelled cAMP. **c** Representative BRET-emission spectra in the presence of different concentrations of 8NBD-c AMP. **d** ΔBRET for Rluc-OMP$^{Wt}$. $n = 8, 6, 25, 26, 23, 24$ and 24 plates. **e** The decrease in ΔBRET (ΔΔBRET) for cAMP as a competitor. $n = 3$ plates. **f** Schematic localisation of mutated residues in Rluc-OMP. **g** Immunocytological confirmation of the cytosolic expression of the mutated Rluc-OMP fusion proteins. **h** Western blot confirmation of the sizes and concentrations of OMP and mutant Rluc-OMP fusion proteins. **i** ΔBRET for the various mutated Rluc-OMP proteins. Wt is the same as that shown in (**d**) for comparison. *$P < 0.001$. Experiments were repeated three times with similar results in (**g**, **h**). Statistics: (**i**) one-way ANOVA at different concentrations of 8NBD-cAMP with post hoc one-sided Dunnett's multiple-comparison test for OMP$^{WT}$ ($n = 32$) vs OMP$^{Mut}$: 0 μM, $F_{(6, 96)} = 0.0386$, $P_{ANOVA} > 0.99$, $P_{Dunnett} > 0.99$ for all; 0.1 μM, $F_{(6, 97)} = 3.048$, $P_{ANOVA} = 0.00921$, $P_{Dunnett} < 0.01$ for OMP$^{126}$ and OMP$^{126/136}$ and >0.1 for the others. In all, 0.2 μM, $F_{(6, 97)} = 7.539$, $P_{ANOVA} < 0.001$, $P_{Dunnett} < 0.01$ for all; 0.5 μM, $F_{(6, 105)} = 16.05$, $P_{ANOVA} < 0.001$, $P_{Dunnett} = 0.0265$ for OMP$^{90/126/136}$ and <0.01 for the others; 1 μM, $F_{(6, 98)} = 72.17$, $P_{ANOVA} < 0.001$, $P_{Dunnett} < 0.001$ for all; 2 μM, $F_{(6, 103)} = 78.78$, $P_{ANOVA} < 0.001$, $P_{Dunnett} = 0.9131$ for OMP$^{126}$ and <0.0001 for the others; 5 μM, $F_{(6, 78)} = 47.84$, $P_{ANOVA} < 0.001$, $P_{Dunnett} = 0.7257$ for OMP$^{126}$ and <0.001 for the others; 10 μM, $F_{(6, 87)} = 47.41$, $P_{ANOVA} < 0.001$, $P_{Dunnett} = 0.0834$ for OMP$^{126}$ and <0.001 for the others. $n = 16, 11, 15, 8, 8$ and 8 for OMP$^{126}$, OMP$^{90}$, OMP$^{136}$, OMP$^{126/90}$, OMP$^{126/136}$ and OMP$^{90/126/136}$, respectively. Mean ± s.e.m.

association with OMP is preferentially faster than dissociation during this phase (Fig. 3j; see Supplementary Note for mathematical modelling). Consequently, OMP prevents the excessive current influx through CNG channels in the small ciliary compartment of ORNs (Fig. 3k, l). Otherwise, Na$^+$/Ca$^{2+}$ overload results in strong depolarisation and excessive adaptation of CNG channels[7–9], which reduces neuronal responses. Hence,

we next investigated the impact of repetitive sensory stimulation on neuronal activity in a semi-intact preparation ex vivo.

**Cyclic AMP buffering by OMP is essential for resilient neuronal responses.** We measured the firing of ORNs via a loose patch clamp in slices of olfactory epithelia obtained from $OMP^{+/GFP}$ heterozygous or $OMP^{GFP/GFP}$ homozygous knockout

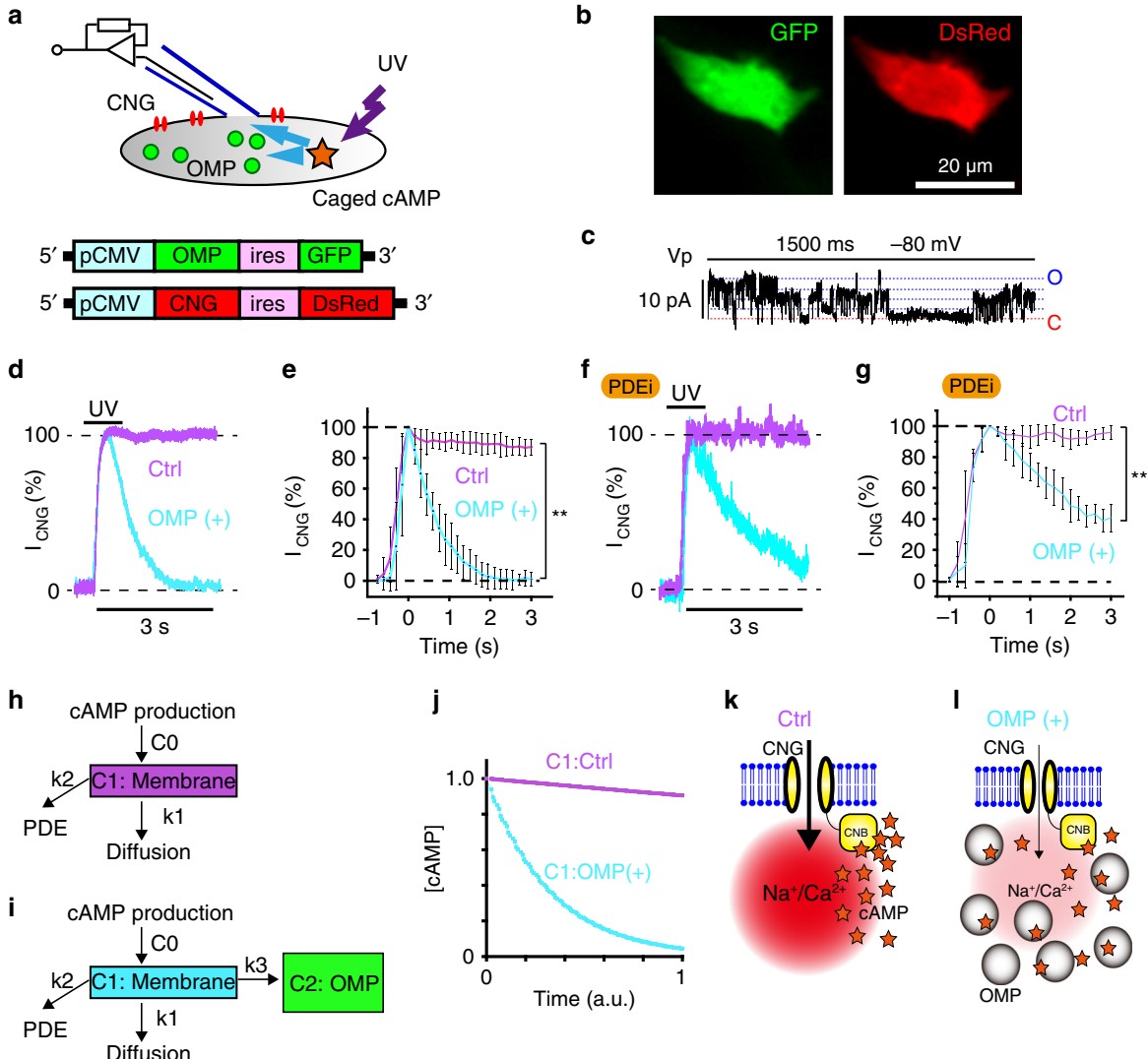

**Fig. 3 OMP sequesters a cytosolic cAMP surge. a** Scheme of the photo-uncaging of cAMP under whole-cell patch-clamp recording and the vectors used for the experiment in (**d–g**). Transfected HEK293T cells coexpressed GFP and DsRed with OMP and CNGA2, respectively, under the control of IRES. **b** Confirmation of fluorescent protein expression in transfected cells. Experiments were repeated three times with similar results. **c** CNGA2 channel activities in the cell-attached configuration. The red dashed line indicates the closed state. The blue dashed lines indicate the open state. **d** Representative UV-induced CNG currents from one cell in the absence (Ctrl) or presence of OMP, overlaid for each condition. **e** Normalised CNG currents (% to peak) induced by the photo-uncaging of cAMP; $n = 13$ and 11 cells for Ctrl and OMP($+$) cells, respectively. Statistics: one-sided unpaired $T$ test; activation time constant, $P = 0.38$; decay %, $P = 0.0001$. **f** Representative UV-induced CNG currents from one cell in the presence of PDEi: overlaid for Ctrl and OMP($+$). **g** Normalised CNG currents (% to peak) induced by the photo-uncaging of cAMP in the presence of PDEi. Mean ± s.d.; $n = 4$ and 4 cells for Ctrl and OMP($+$) cells, respectively. One-sided unpaired $T$ test; activation time constant, $P = 0.78$; decay %, $P = 0.0002$; $n = 4$ and 4 cells for recordings with and without OMP, respectively. **h** Single-compartment model without OMP. **i** Two-compartment model with OMP. k1–k3, the diffusion constant (k1), cAMP-eliminating activity of PDE (k2) and cAMP capture of OMP (k3). Assuming that k3 was much larger than k2, cAMP dissociation from OMP was ignored within this duration. **j** Simulated cAMP in the membrane compartment (C1). a.u. arbitrary unit of time. Time and [cAMP] are given in arbitrary units in the figures. **k, l** Schematics of the CNG channel activation time course; (**k**) prolonged Na$^+$/Ca$^{2+}$ influx without OMP and (**l**) small Na$^+$/Ca$^{2+}$ influx through rapidly deactivated CNG channels via cAMP buffering with OMP. Mean ± s.d.

mice[25] (Het and KO neurons, respectively). Both Het and KO neurons expressed GFP and were easily visible and targeted under a fluorescence microscope[25]. The KO neurons showed spontaneous firing at rates similar to those of the Het neurons (5.0 ± 3.8 and 5.3 ± 2.5 Hz; mean ± s.d., $n = 70$ and 47 for KO and Het neurons, respectively, three mice each, Fig. 4a). The firing frequency showed different patterns of distribution ($P = 0.0002$, F test, Fig. 4a). ORNs showed various firing patterns, including irregular firings with different spike amplitudes or regular firings (Supplementary Fig. 5a–c). Even ORNs that fired regularly sometimes showed burst firings, which were accompanied

by diminished spike amplitudes, most likely due to membrane depolarisation and inactivation of voltage-gated sodium channels[43].

In subsequent experiments, we investigated the sensory responses from the ORNs with firing frequencies within ± 1 s.d. of the mean (~2–8 Hz) and evaluated changes in the average firing frequencies of ORNs that were sensitive to sensory stimuli. When we applied a single stimulation with odourant mixtures for 200 ms towards cilia of ORNs, several ORNs increased their firing with an initial delay (Fig. 4b). Amongst these odourant-sensitive ORNs, Het neurons showed a transient increase in firing with a

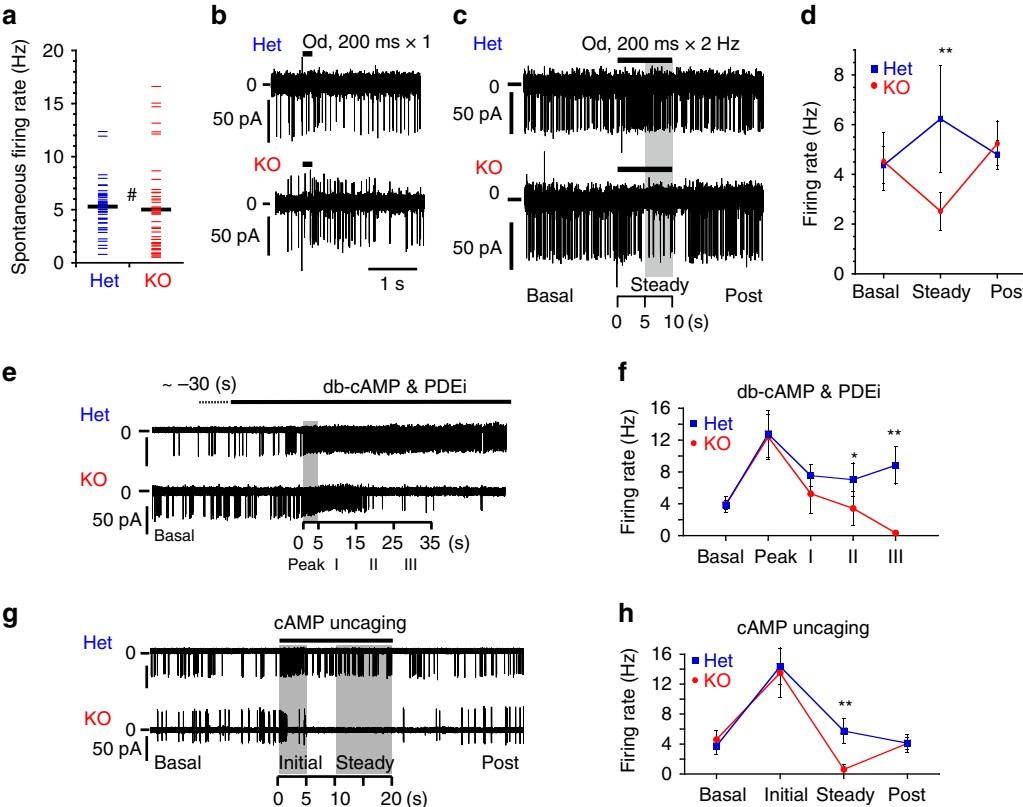

**Fig. 4 OMP-KO neurons are silenced by repetitive stimuli. a** Spontaneous firing rates of Het and KO neurons. Means: black bars. **b** Representative traces of firings induced by a single-odour stimulus in Het and KO neurons. Odourant mixtures contained amyl acetate, cineole, limonene, acetophenone, heptanal and lilial at a concentration of 100 μM each. **c** Representative traces of firings induced by repetitive odour stimuli in Het and KO neurons. **d** Time courses of the average firing rates of Het and KO ORNs shown in (**c**). Steady indicates the period between 5 and 10 s in (**c**). **e** Representative traces of spontaneous firings of Het and KO neurons under the loose patch-clamp configuration in the presence of db-cAMP and PDEi. The grey bars indicate the region with maximal frequency after perfusion. **f** Time courses of the average firing rates of ORNs shown in (**e**). The peak was determined at the position of the maximal frequency within a 5-s bin after perfusion of the test solution; other segments were determined within 10-s bins. Mean ± s.d. **g** Representative traces of spontaneous firing of Het and KO neurons during UV uncaging of membrane-permeable cAMP. **h** Time courses of the firing rates of the ORNs shown in (**g**). The initial rate was determined within a 5-s bin; other segments were determined within 10-s bins. Statistics: **a** two-sided unpaired $T$ test; $P = 0.548$ (#), $n = 70$ and 47 ORNs from three Het and three KO mice, respectively. **d** Two-sided unpaired $T$ test; the $P$-values for basal, steady and post are 0.4, 0.007 and 0.2. $n = 5$, each. **f** Two-sided unpaired $T$ test; the $P$-values for basal max, i, ii and iii are 0.89, 0.84, 0.11, 0.028 and 0.000039, respectively; $n = 5$ ORNs each from three Het or three KO mice. **h** Two-sided $T$ test; the $P$-values for basal, initial, steady and post are 0.18, 0.58, 0.0000022 and 0.95, respectively; $n = 7$ Het neurons and 8 KO neurons from three mice each. Mean ± s.d.

mean inter-spike interval (ISI) of $25 \pm 10$ ms (a minimal ISI of $10 \pm 9$ ms through a maximum ISI of $43 \pm 21$ ms, $n = 6$), whilst KO neurons slowly increased firing and lasted longer than Het neurons at an ISI of $131 \pm 20$ ms (a minimal ISI of $53 \pm 31$ ms through a maximum ISI of $216 \pm 55$ ms, $n = 6$). These prolonged responses in KO neurons were consistent with previous reports using OMP-KO mice[27,29,32] (Supplementary Fig. 5d, e). Next, we repetitively applied odourant mixtures at a duration of 200 ms and 2 Hz for 10 s; odour-responding Het neurons showed evoked firings at a sustained steady level after the initial strong increase in firing (Het, Fig. 4c, d), whilst the firing of KO neurons first increased and then decreased to below the basal level as the time of stimulation progressed (steady firing of KO neurons, Fig. 4c, d). Consistently, previous studies also demonstrated that repetitive odour stimulation caused the delayed recovery of electro-olfactography (EOG) in KO mice[27,29] (Supplementary Fig. 5d, e), which implies that receptor potentials are diminished, and that the generation of action potentials is impaired with repetitive stimuli in KO mice. Taken together, OMP is necessary to maintain evoked firing during odourant stimulation.

To imitate olfactory signalling via cAMP[17,40,41], we directly increased cAMP by coapplying membrane-permeable dibutyryl-cAMP (db-cAMP, 3 mM) and an inhibitor of PDE, 3-isobutyl 1-methylxanthine (IBMX, 30 μM). IBMX alone elevates basal cAMP pools and stably boosts spontaneous firing frequency[41]. Upon bath application of db-cAMP and IBMX, Het neurons showed an initial increase in spontaneous firing, followed by adaptation to the moderately elevated firing level (Het, Fig. 4e, f). However, KO neurons burst-fired and were then nearly silenced (KO, Fig. 4e, f). Approximately 30 s of perfusion with solution was required to induce the effect, at which time the maximal concentration was reached in the recording chamber, although the concentration could not be clearly determined inside the cells (Fig. 4e). Accordingly, we employed a strategy to trigger a cAMP surge via UV uncaging of membrane-permeable caged cAMP with temporal precision. UV irradiation increased the firing of the Het neurons (Het, Fig. 4g, h). The KO neurons instantly increased their firing upon UV irradiation, but the firing rate declined promptly and was even silenced during the UV flash (KO, Fig. 4g, h). These results indicate that OMP is necessary to maintain firing under cAMP overload.

In addition to sensing ambient odours, animals can navigate to an odour source by continual sniffing. Sniffing is accompanied by repetitive airflow in the nasal cavity in vivo[18–23]. This

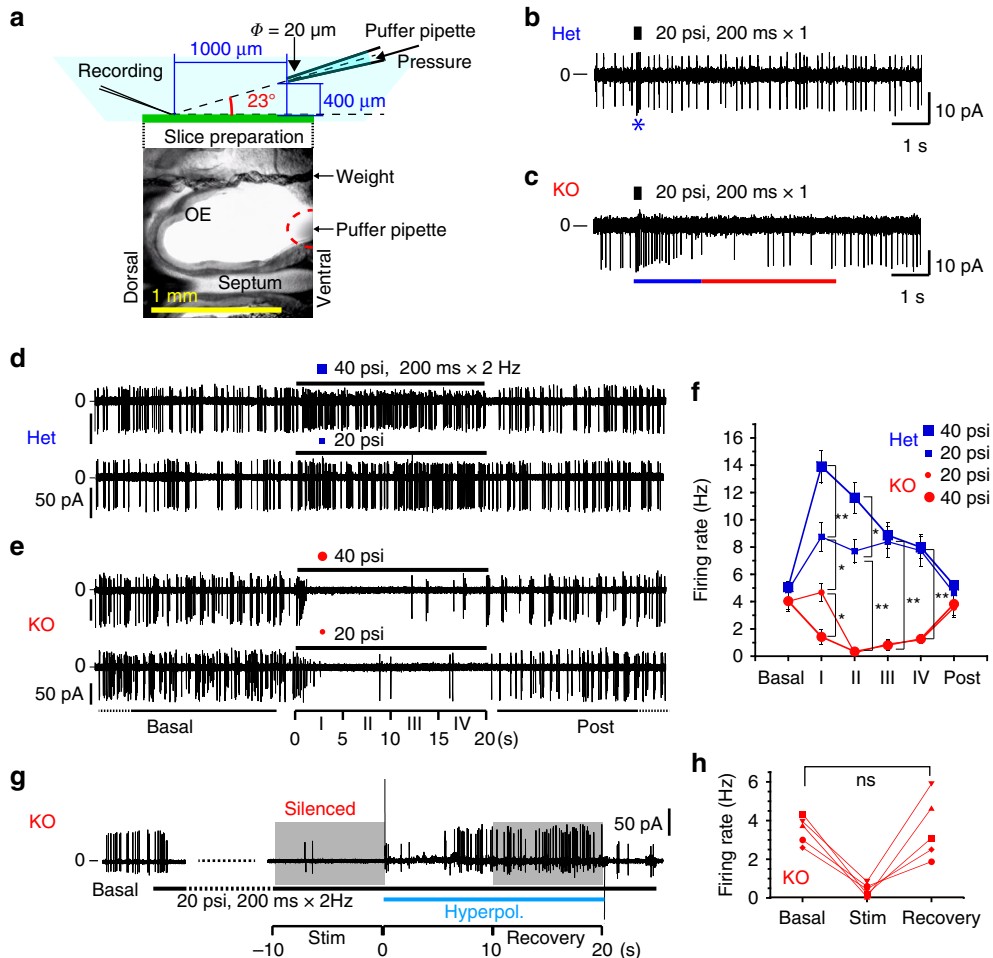

**Fig. 5 OMP-KO neurons are silenced by mechanical stimulation. a** Setup for electrical recordings of ORN firing under mechanical puff stimulation. **b**, **c** Representative traces of firings induced by a single mechanical stimulus in (**b**) Het and (**c**) KO neurons. psi = pound-force per square inch. To define the boundaries of the accelerated phase (* or a blue bar) or the depressed phase (a red bar), the average of two adjacent ISIs was compared with ISI$_{Basal}$. **d**, **e** Representative traces of firings induced by repetitive mechanical stimuli in (**d**) Het and (**e**) KO neurons. A stimulating puff with a duration of 200 ms at 2 Hz was applied for 20 s. **f** Time courses of the spontaneous- and mechanically induced firing rates of ORNs. The spontaneous firing rates before and after stimulation (basal and post) were calculated in 1-min bins, and the parastimulatory firing rates were calculated in 5-s bins (I–IV). $n = 11$ and 8 KO neurons; 9 and 13 Het neurons for 20 and 40 psi. *$P < 0.05$; **$P < 0.01$. **g**, **h** Representative trace and summary of the recovery from silencing in KO ORNs by a hyperpolarisation pulse from outside with a sudden increase in Vp to + 60 mV. Statistics: (**f**) one-way ANOVA and post hoc Bonferroni multiple-comparison test for different stimulation strengths (20 and 40 psi); $n = 9$–13 ORNs from 3 Het mice and 9–11 ORNs from 3 KO mice. Basal, i, ii, iii and iv, and Post: $F_{(6, 39)} = 1.317$, $P < 0.0001$; $F_{(6, 39)} = 31.69$, $P < 0.0001$; $F_{(6, 39)} = 47.54$, $P < 0.0001$; $F_{(6, 39)} = 28.61$, $P < 0.0001$; $F_{(6, 39)} = 24.65$, $P < 0.0001$; $F_{(6, 39)} = 1.674$, $P = 0.19$, respectively. Bonferroni test: *$P < 0.05$; **$P < 0.01$; $P > 0.05$ for all the other pairs. **h** Paired two-sided $T$ test; no significant difference (ns), $n = 5$ recordings. Mean ± s.d.

airflow causes mechanical stress on the olfactory epithelium and stimulates ORNs. Thus, we postulated that airflow-induced mechanical stress during sniffing would also affect the olfaction of KO mice in parallel with chemical stimulation by odourants. Thus, we further compared the properties of mechanosensitive ORNs with or without OMP. We induced mechanical turbulence on the spontaneously firing ORNs by applying the flow of an external solution from puffer pipettes placed approximately 1000 μm from the recording site (Fig. 5a).

Under a single-puff stimulus (200-ms duration at 20 psi, 1 pulse), the firing rate of mechanosensitive Het neurons instantly increased and then quickly returned to the basal level (Fig. 5b); the inter-spike interval (ISI) of Het neurons was shortened to 47 ± 39 ms from the basal ISI (222 ± 30 ms, measured during a 5-s pre-stimulatory period). The increased firing rate was continued for 117 ± 22 ms (mean ± s.d., $n = 4$, blue asterisk in Fig. 5b). On the other hand, the firing rate of KO neurons was increased for

1.7 ± 0.5 s by puff stimulation (blue bar in Fig. 5c). The firing activity of KO neurons was then suppressed (red bar in Fig. 5c) for the next 4.7 ± 0.5 s and then returned to the basal level. The ISI was 150 ± 43 ms during the transient phase, 575 ± 260 ms during the suppression phase and 243 ± 86 ms after recovery to the basal phase (mean ± s.d., $n = 3$). In KO neurons, the amplitude of spikes progressively diminished along with a high rate of firing activity during the transient phase after puff stimulation, indicating that the rise speed of action potentials was decreased. When sporadic firing activity returned, the spike size also returned to normal (Fig. 5c).

To emphasise the effects of repetitive stimuli, we applied a 20-s long puff at 2 Hz to ORNs. In the Het neurons, mild puff stimulation (20 psi) increased the firing rate (Het, 20 psi, Fig. 5d; Supplementary Fig. 5f, g; Supplementary Movie 3). Stronger puff stimulation (40 psi, Supplementary Movie 4) induced a larger initial burst. The firing rate initially increased and then deceased

adaptively to the level induced by 20-psi puff in 10 s, regardless of the intensity of puff stimulation (phase III, both 20 and 40 psi, Fig. 5d–f). Thus, the Het neurons sustained evoked firing under continual mechanical stress. In contrast, the KO neurons were silenced soon after the initial burst at the onset of puff stimulation until the end of the stimulus (KO, 20 and 40 psi, Fig. 5e, f; Supplementary Fig. 5f, g). This mechanosensitivity was abolished in the presence of an AC inhibitor (5-(3-bromophenyl)-1,3-dimethyl-5,11-dihydro-1H-indeno[2,1:5,6]pyrido[2,3-d]pyrimidine-2,4,6-trione (BPIPP, 10 µM) or SQ 22,536 (100 µM); Supplementary Fig. 5h–l), confirming that the mechanically induced activity was the result of cAMP production as previously demonstrated[33–35].

Although the firing of KO neurons remained silent during mechanical stimulation, the external electrical field stimulation that was applied to hyperpolarise the membrane potential of the soma recovered the firing and amplitude of action potentials to basal levels (Fig. 5g, h). This finding suggests that the firing machinery around the soma, i.e., voltage-gated sodium channels, was inactivated by membrane depolarisation during puff application. These results indicate that cAMP buffering by OMP prevents the silencing of ORNs due to sustained membrane depolarisation, which is most likely a consequence of $Na^+$ and $Ca^{2+}$ overload through the open CNG channels.

**OMP haploinsufficiency results in a smaller cAMP-buffering capacity.** As indicated in Fig. 3g, the cAMP-buffering capacity appears to be dependent on the quantity of available OMP inside ORNs. We next compared $OMP^{+/+}$ wild-type ORNs (WT neurons) and Het neurons, as KO neurons were quickly silenced under cAMP overload (Fig. 4b–h, Supplementary Fig. 6a, b). The expression of OMP was compared by immunofluorescence analysis, confirming that WT neurons had significantly stronger OMP immunoreactivity than Het neurons in the dendritic areas (Fig. 6a, b). First, we applied mechanical stimulation, and both the WT and Het neurons showed increased firing rates (upper traces, Pre, Fig. 6c–e). Then, we applied a PDEi (30 µM IBMX or 10 µM rolipram) to elevate the basal cAMP pool, which resulted in significant and stable increases in the basal firing rates of both WT and Het neurons, as previously shown[41,43] (basal, Fig. 6c, e; Supplementary Fig. 6c–f). Furthermore, WT neurons maintained generally stable firing activity even during mechanical stimulation (lower trace, Fig. 6c, Supplementary Fig. 6c; Stim, Fig. 6e, Supplementary Fig. 6e) and recovered swiftly to the pre-stimulatory level upon termination of the stimulus (WT, I–III, lower panel, Fig. 6e, Supplementary Fig. 6e). In contrast, upon PDE inhibition, Het neurons exhibited a decreased firing rate when mechanically stimulated, but were not silenced (Het, Stim, lower panel, Fig. 6d, e, Supplementary Fig. 6d, f). Furthermore, the recovery of firing was delayed for tens of seconds from the end of stimulation in the presence of PDEi (I–III; Fig. 6d, e, Supplementary Fig. 6d, f). A similar trend in firing changes was observed by odourant stimulation: WT neurons showed a sustained increase in firing under repetitive odourant stimulation at rest and in the presence of PDEi, while Het neurons, which showed odourant sensitivity at rest, exhibited decreased firing in the presence of PDEi at steady state during odourant application (Fig. 6f–h; Supplementary Fig. 6g–j). Taken together, Het neurons exhibited a smaller buffering capacity for the cAMP load than WT neurons upon PDEi administration, indicating that Het neurons with decreased OMP expression showed haploinsufficiency.

These observations were similar to those in KO neurons (compare Figs. 3d–g, 4b–h; Supplementary Fig. 6a, b). Under repetitive stimuli, PDE principally contributes to the gradual elimination of the remaining cAMP, as previously reported[8],

while OMP confers tolerance to the rapid cAMP load to maintain repetitive responses.

The results presented thus far depict a scenario, in which the cAMP-buffering capacity of OMP depends on its expression level and provides a cAMP signalling bypass that avoids ORN silencing to ensure resilient olfactory responses under repetitive stimuli.

**OMP-deficient mice lose olfactory sensitivity during odour-source searching.** Previous studies utilising odour-presenting assays have suggested that OMP-KO mice show various olfactory impairments in odour sensitivity, selectivity and discrimination[1,25–32].

We performed in vivo extracellular recordings from the olfactory epithelium nerve layer of anaesthetised mice[44] using a large-tip glass electrode (Fig. 7a) and applied air-puff stimulation into the nostril (Fig. 7a–c). The stable multiunit firing activity comprised several different action potential clusters with different amplitudes (Supplementary Fig. 7a–f). These collective firing activities were quantified by thresholding[45,46] above the noise level defined by +3 s.d. of basal signal fluctuations during the 20-s pre-stimulatory period (Supplementary Fig. 7d, e). The puff stimulation applied to the anaesthetised mice affected the overall collective activity during stimulation (Fig. 7b–g). When limonene and food odours were applied via intranasal air puff, the olfactory nerves of WT and Het mice exhibited increased multiunit firing during stimulation, whereas the nerves of KO mice exhibited decreased firing (L( + ) and Fd, Fig. 7h). Air puffs without odours (air, Fig. 7h) also increased firing in WT and Het mice and decreased firing in KO mice.

Then, we examined the vulnerability of OMP-deficient mice to cAMP by performing an odour-source localisation test. First, mice were trained to associate a reward with limonene (L( + )) for the subsequent experiments. WT, Het and KO mice similarly preferred the sucrose solution (Fig. 8a–c). Using sucrose as a reward, we trained water-deprived mice to associate sucrose water with L( + ) for 8 days (Fig. 8d). Mice were forced to locate the sucrose solution suspended together with the limonene-odour emitting container in room air or in the presence of a background odour (Fig. 8e–h). In room air, KO mice took longer to reach the spout of the reward-containing bottle than WT and Het mice (room air, Fig. 8i). In addition, KO mice showed a tendency to locate the incorrect spout (room air, Fig. 8j). Next, the mice were forced to locate the reward bottle in a cage in the presence of a background odour (1 v/v%), and the bottle locations were switched. The background odour would increase the basal cAMP level in limonene-sensitive ORNs (Fig. 8f, h). Het and KO mice spent more time searching around the incorrect bottle, which was situated in the test cage on the same side as that used in the previous experiments (1% background, Fig. 8i). The number of wrong licks of the empty bottle was increased in Het and KO mice (1% background, Fig. 8j), indicating that the odour-source localisation of KO mice by sniffing was further impaired. The vulnerability of Het mice was unveiled in the presence of a constant basal sensory stimulation applied as a background odour. However, the bottle and spouts might also serve as a visual cue to search and confuse the odour orientation.

Hence, we reconsidered the olfaction abilities of OMP-KO mice based on sequential behavioural aspects in hidden odour searching, passive sensation and active searching by sniffing. Food-deprived mice were placed on a platform with four holes, all of which were plugged with two tissue wads to eliminate visual aids (upper panel in Fig. 9a). Naive mice were expected to sniff out the correct hole containing the food reward. Because of the possibility that the mouse arbitrarily removes one wad, the mouse needed to remove two wads successively within 2 min to access the food reward (lower panel in Fig. 9a). This learning session was

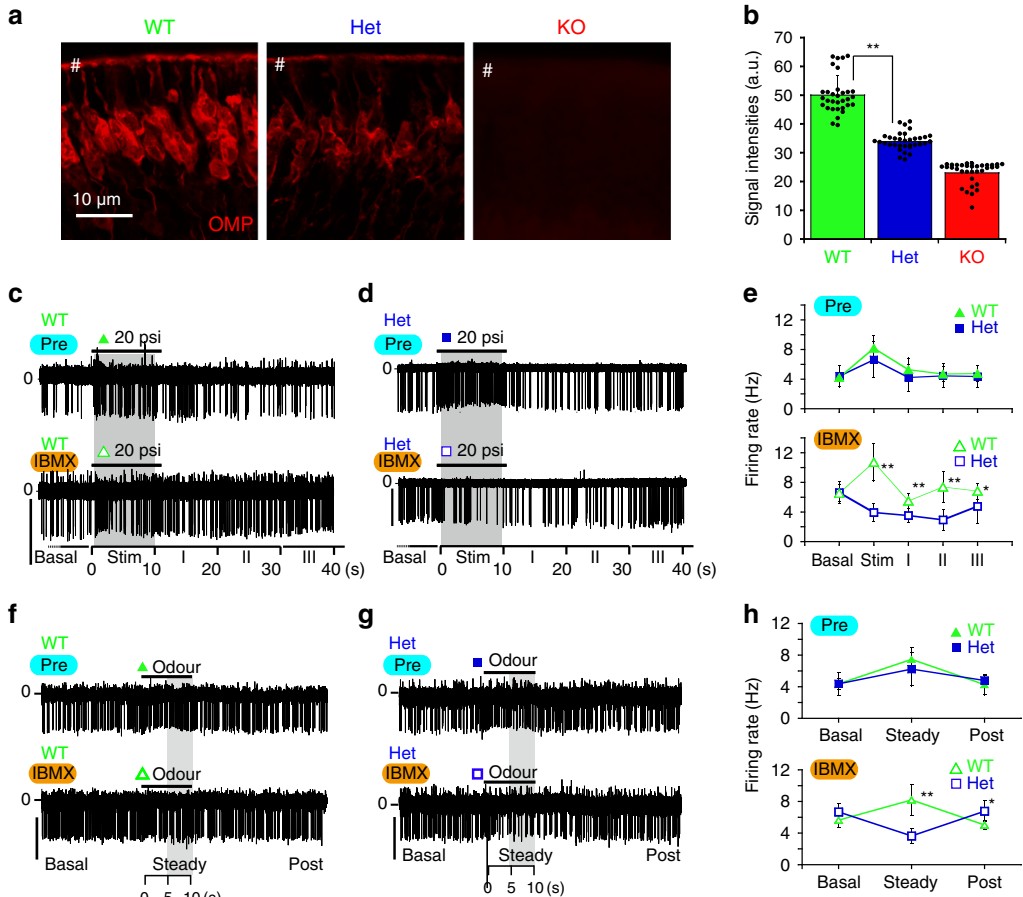

**Fig. 6 OMP confers tolerance to cAMP overload. a** OMP immunoreactivity in WT, Het and KO neurons. # indicates apical dendritic regions of ORNs. **b** OMP-immunoreactive signal intensities were measured at # in (**a**). a.u. arbitrary unit. Signals of KO neurons were used to set a nonspecific background level. $n = 34$ and 32 ROIs. **c**, **d** Representative traces of spontaneous and mechanically induced firings (200-ms duration at 2 Hz for 10 s) of (**c**) WT neurons and (**d**) Het neurons before (Pre) and after (IBMX) the application of a PDEi. The spontaneous firing rate before stimulation (Basal) was calculated in 1-min bins, and the parastimulatory firing rates were calculated in 10-s bins (Stim and I–III). **e** Time courses of the mechanically induced and spontaneous firing rates of ORNs before (Pre) and after (IBMX) the application of a PDEi. **f**, **g** Representative traces of spontaneous and odourant-induced firings (200-ms duration at 2 Hz for 10 s) of (**f**) WT neurons and (**g**) Het neurons before and after the application of IBMX. **h** Time courses of the odourant-induced and spontaneous firing rates of ORNs shown in (**f**) and (**g**). $n = 5$, each. Steady indicates the period between 5 and 10 s in (**c**). Basal and post indicate the 5-s bins, which are ~10 s prior or 10 s posterior to a stimulatory period. Scales: 20 pA. Mean ± s.d. *$P < 0.05$; **$P < 0.01$. Statistics: **b** a priori two-sided unpaired $T$ test; $P_{\text{WT vs Het}} = 2.8 \times 10^{-19}$, **e** a priori two-sided unpaired $T$ test; $P_{\text{WT vs Het}}$ values for the Basal, Stim, I, II and III periods: 0.78, 0.19, 0.25, 0.74 and 0.58 for the control (Pre) and 0.92, 0.00000092, 0.00073, 0.000056 and 0.041 in the presence of IBMX, respectively; $n = 7$ and 11 for WT and Het, respectively. **h** A priori two-sided unpaired $T$ test; $P_{\text{WT vs Het}}$ for the control (Pre) are 0.5, 0.16 and 0.22. $P_{\text{WT vs Het}}$ in the presence of IBMX for the basal, steady and post periods are 0.07, 0.001 and 0.02. Mean ± s.d.

repeated five times by changing the location of the food reward to discriminate the contribution of olfaction-based searching from that of memory-based searching.

During the initial trial, WT, Het and KO mice actively wandered around the platform to learn the test scheme (1st, Fig. 9b, c). During this learning period, WT and Het mice remained at the correct hole containing the food reward longer than the KO mice (2nd to 5th of Fig. 9b). The WT and Het mice exhibited higher success rates of removing the two wads out of the hole to reach the reward (double dig) than the KO mice (coloured bars, Fig. 9c). Moreover, the KO mice continued to search among the incorrect holes as the sessions progressed (hot spots at locations without food, Fig. 9b) and sometimes removed the wrong wads (black bars, Fig. 9c), indicating that the KO mice sensed the ambient odours passively but exhibited hyposmia during odour-source searching by sniffing.

**OMP-deficient mice present temporary hyposmia under cAMP challenge.** In addition, the task-trained mice were challenged by intranasal administration of an olfactory-specific PDE4 inhibitor[47,48] to elevate the basal cAMP level in ORNs (rolipram, 300 µM, 10 µL/nostril). As expected, the WT and Het mice reached the correct holes containing the food reward, while the KO mice rarely succeeded, either making errors (Fig. 9d) or remaining still on the platform (PDEi session of KO, Fig. 9b). However, the PDEi-challenged Het mice very frequently made errors in removing the wads, appearing to search for food in a random manner (Fig. 9e). These results indicate that the WT mice retained olfaction after PDEi challenge. The Het mice energetically but arbitrarily attempted to sniff the food source in momentary hyposmia, while the KO mice passively sensed too few scents to continue searching, apathetically removed one or a few wads or made random errors.

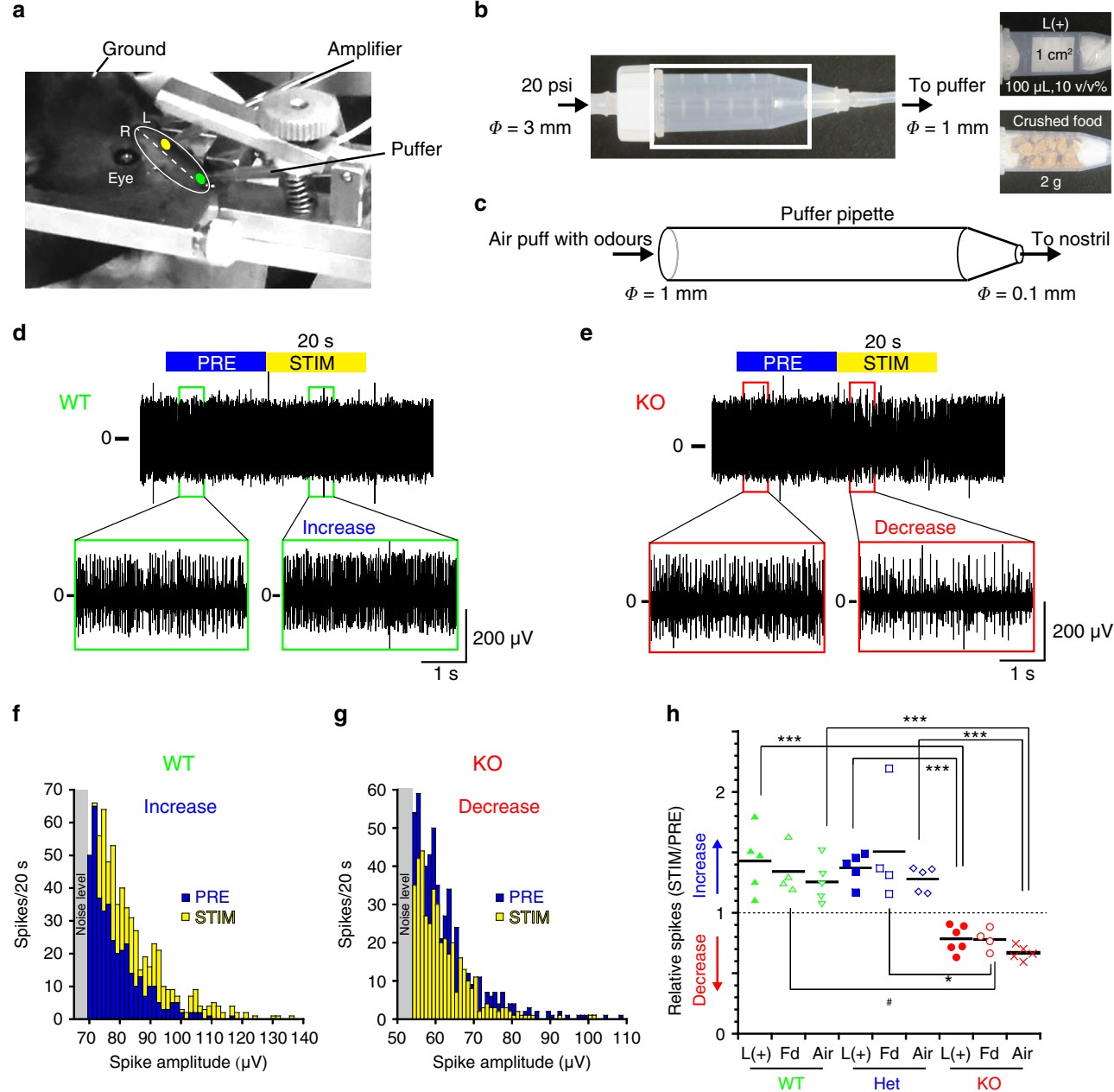

**Fig. 7 OMP-KO nerves show rapid adaptation in response to repetitive stimulation. a** Experimental setup of an in vivo multiunit recording from the nerve layer of the olfactory epithelium using a recording pipette inserted through a hole in the left nasal bone (a yellow dot). Odours were applied through a puffer pipette inserted into the left nostril (a green dot). **b** A 5-mL plastic tube containing scented filter paper or food as an odour application chamber was interconnected between a pressure pump and a puffer. **c** An approximately 0.2-psi air puff was applied from the puffer pipette into the nostril. Repetitive air puffs with 100-ms durations followed by 300-ms intervals at 20 psi were applied for 20 s. ϕ: diameters. **d** Multiunit responses from the nerve bundle layer of WT mice show sustained increases in firing during stimulation. **e** Multiunit responses from the nerve bundle layer of KO mice show a decrease during stimulation. **f, g** Representative histograms of spike amplitudes before (PRE) and during (STIM) odour-puff stimulation for 20 s in (**f**) WT mice and (**g**) KO mice. **h** Relative changes in the collective firing frequencies during nasal air-puff stimulation with L( + ) or food (Fd) or without odours (Air) to the resting frequencies (PRE) in (**h**) WT, Het and KO mice. Means: black bars. *$P < 0.05$; **$P < 0.01$; ***$P < 0.001$. Statistics: **h** one-way ANOVA and the post hoc Bonferroni multiple-comparison test. L( + ); $F_{(3, 13)} = 23.06$, $P < 0.001$ and $P_{WT vs Het} > 0.999$, $P_{WT vs KO} = 0.0001$, $P_{Het vs KO} = 0.0003$. $n = 5$, 5 and 6 for WT, Het and KO. Food; $F_{(3, 9)} = 6.602$, $P = 0.017$ and $P_{WT vs Het} > 0.999$, $P_{WT vs KO} = 0.076$ and $P_{Het vs KO} = 0.021$. # means $P_{WT vs KO} = 0.045$ in the post hoc Dunnett test. $n = 4$ each for WT, Het and KO. Air; $F_{(3, 12)} = 41.08$, $P < 0.0001$ and $P_{WT vs Het} = 0.32$, $P_{WT vs KO} < 0.0001$ and $P_{Het vs KO} < 0.0001$. $n = 5$ each for WT, Het and KO.

To further verify the presence of hyposmia during sniffing in the OMP-deficient mice, we placed a food chunk directly in front of the task-trained, starved mice in a test chamber (Fig. 10a–d). With this visual aid, the WT, Het and KO mice swiftly approached the food, sniffed eagerly and began to eat (Control in Fig. 10a, d; Supplementary Movies 5, 6). In contrast, following an intranasal PDEi challenge to elevate basal cAMP, all strains of mice took longer to approach the visible food by sniffing (Fig. 10c). The WT mice eventually started to eat the food with a few erroneous digs (WT, Fig. 10a, c, d; Supplementary Movie 7),

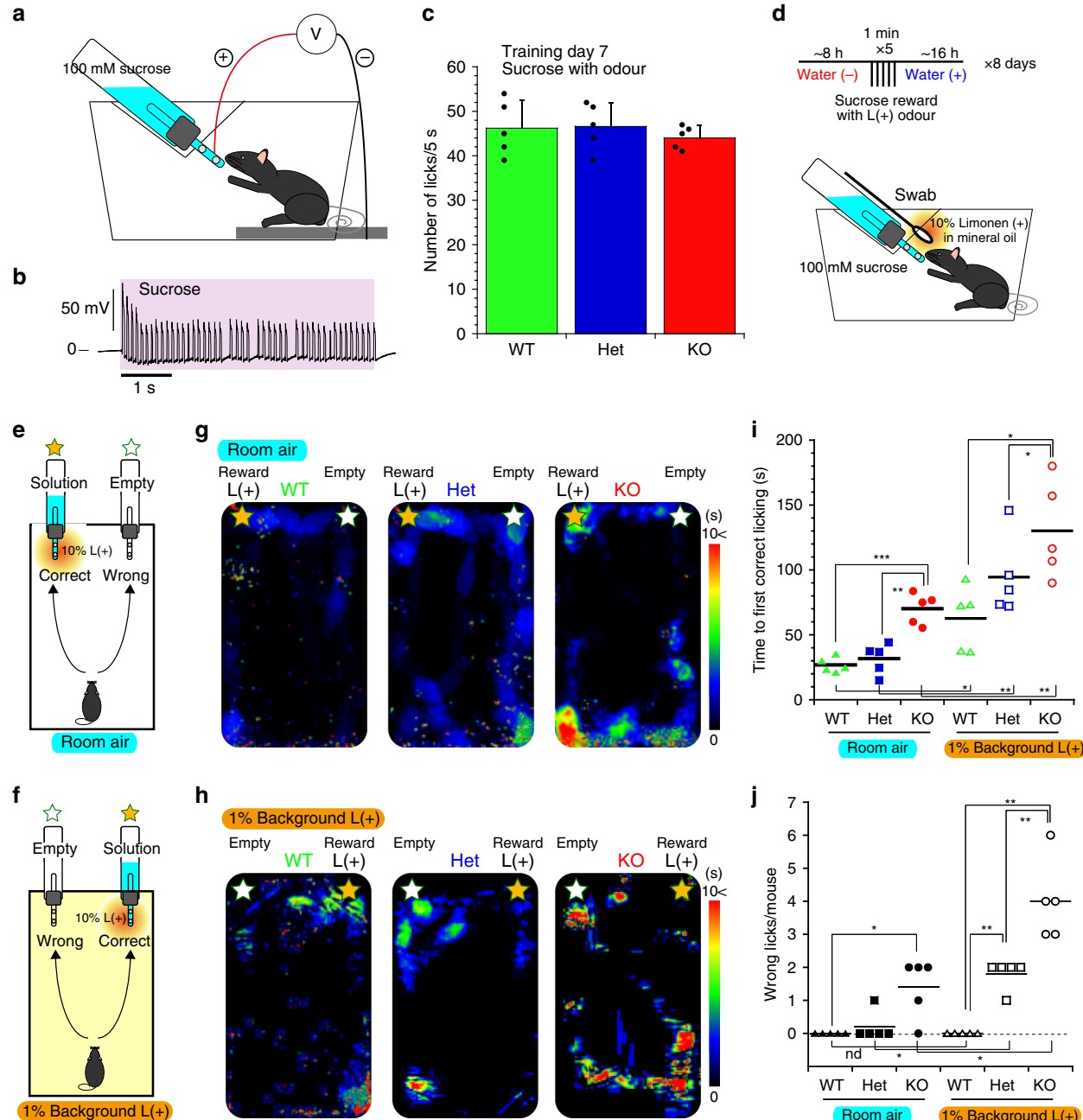

**Fig. 8 OMP-deficient mice lose olfactory sensitivity to background odours. a** The sucrose preference measurement by a licking metre. **b** Preference of mice for sucrose. Licking was counted for 5 s from the first lick. **c** Summary of licks during 5 s. Mean ± s.d. **d** Training paradigm to associate a sucrose reward with the limonene-(+) odour. A swab containing L(+) was presented with a sucrose solution. **e, f** Schemes of 10% L(+)-odour-source searching associated with a sucrose reward in (**e**) room air and (**f**) an atmosphere with a 1% L(+) background odour. **g, h** Representative heatmaps of the times mice spent reaching and licking the correct spout with the sucrose reward. Bottles with or without a reward were switched in different sessions (**g, f**). **i** Time to the first correct licking of the reward bottle. **j** The number of wrong licks of the empty bottle before the first correct lick. Means are shown in black bars in (**i, j**). $n = 5$ each for WT, Het and KO in (**c, i, j**). *$P < 0.05$; **$P < 0.01$; ***$P < 0.001$. Statistics: room air; **i** one-way ANOVA and post hoc Bonferroni test; $F(3, 12) = 27.26$, $P < 0.0001$ and $P_{WT\ vs\ Het} > 0.999$, $P_{WT\ vs\ KO} < 0.0001$, $P_{Het\ vs\ KO} = 0.0002$. **j** Kruskal–Wallis test; $P = 0.018$. The Mann–Whitney $U$ test for further comparison; $P_{WT\ vs\ Het} = 0.42$, $P_{WT\ vs\ KO} = 0.023$, $P_{Het\ vs\ KO} = 0.053$. Background odour; **i** one-way ANOVA and post hoc Bonferroni test; $F(3, 12) = 5.85$, $P = 0.017$ and $P_{WT\ vs\ Het} = 0.4$, $P_{WT\ vs\ KO} = 0.015$, $P_{Het\ vs\ KO} = 0.29$. **j** Kruskal–Wallis test; $P = 0.0013$. Mann–Whitney $U$ test; $P_{WT\ vs\ Het} = 0.0056$, $P_{WT\ vs\ KO} = 0.0071$, $P_{Het\ vs\ KO} = 0.0092$. Pre/post comparison; **i** one-sided $T$ test without (pre) and with (post) a background odour; $P_{WT,\ pre/post} = 0.015$, $P_{Het,\ pre/post} = 0.0035$, $P_{KO,\ pre/post} = 0.0099$. **j** Mann–Whitney $U$ test; $P_{WT,\ pre/post} = $ not determined (nd), $P_{Het,\ pre/post} = 0.01$, $P_{KO,\ pre/post} = 0.01$.

while the KO mice wandered around far less actively and sometimes stopped without digging any holes or even eating the visible food (KO, Fig. 10c, d), indicating that the KO mice challenged with PDEi visually noticed a novel object but were not able to recognise it as food due to severe hyposmia. Surprisingly, following PDEi administration, the Het mice also noticed the object but did not begin to eat; rather, they restlessly wandered and sniffed around the arena. These PDEi-challenged Het mice

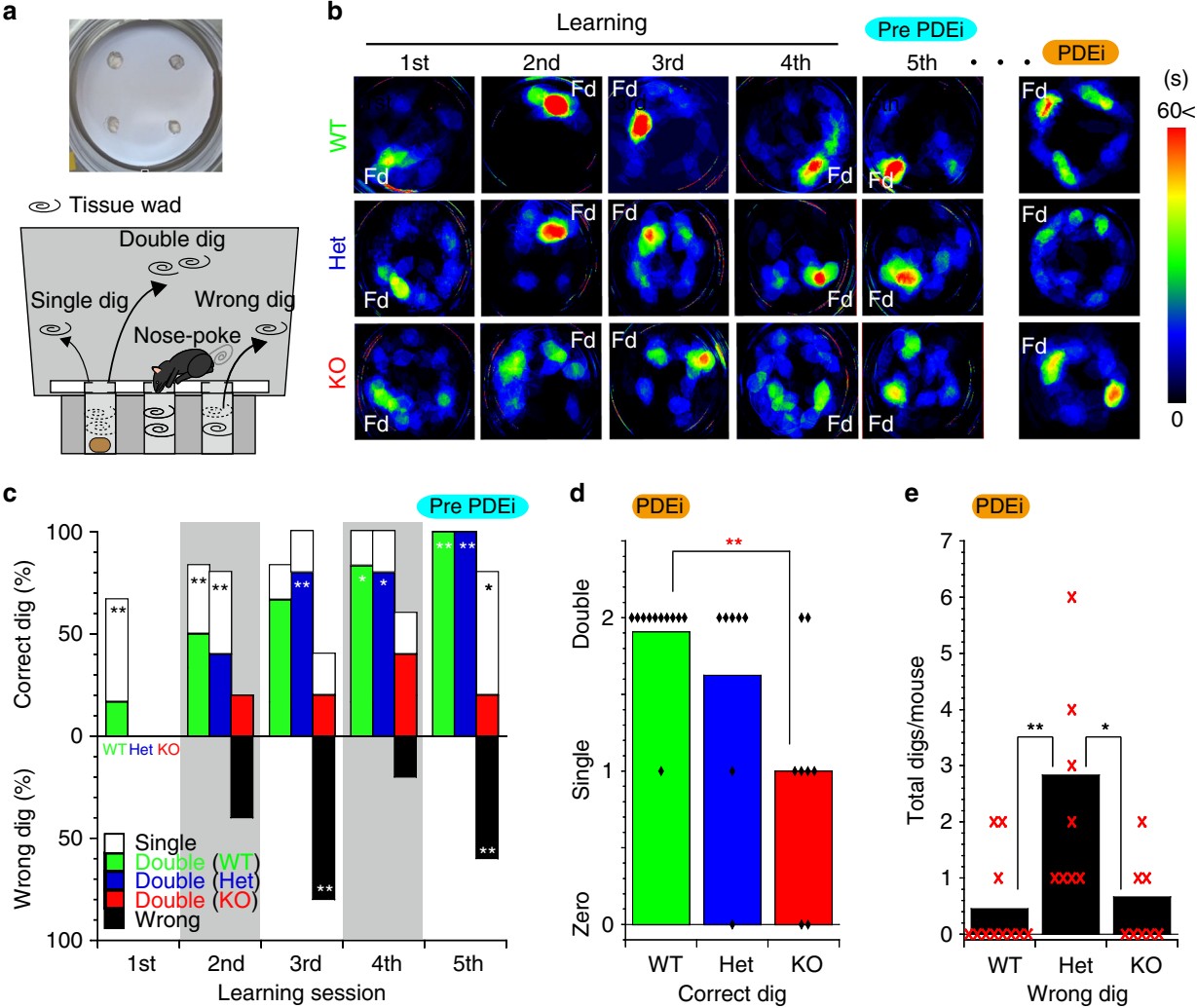

**Fig. 9 OMP-deficient mice lose olfactory sensitivity during sniffing. a** Schematic representation of the odour search assay. Platform for the odour-source-locating assay (upper). Testing scheme for the odour-source-locating assay (lower). A starved mouse was tasked with finding a piece of food hidden in one of the four holes by successively removing two tissue wads. **b** Heatmaps of the time spent by animals during each session. Fd food reward, PDEi session with intranasal PDEi administration. **c** Success-error histogram during each learning session. $n = 6$, 5 and 5 animals in the WT, Het and KO groups, respectively. **d** The numbers of correct digs by mice challenged with PDEi. **e** The number of wrong digs by mice challenged with PDEi. Diamonds and X-marks indicate the numbers of mice used in (**d**, **e**); 9, 7 and 7 for WT, Het and KO mice, respectively. *$P < 0.05$, **$P < 0.01$. Statistics: **c** one-way ANOVA and post hoc Bonferroni test, Double dig, $F(3, 15) = 17.88$, $P = 0.0002$ and $P_{WT\ vs\ Het} > 0.999$, $P_{WT\ vs\ KO} = 0.0004$, $P_{Het\ vs\ KO} = 0.0006$. Single dig, $F(3, 15) = 1.117$, $P = 0.3567$; $P_{WT\ vs\ Het} > 0.999$, $P_{WT\ vs\ KO} = 0.6178$, $P_{Het\ vs\ KO} = 0.6741$. Wrong dig, $F(3, 15) = 6.703$, $P = 0.00999$: $P_{WT\ vs\ Het} > 0.999$, $P_{WT\ vs\ KO} = 0.0186$, $P_{Het\ vs\ KO} = 0.0243$. $n = 6$, 5 and 5 for WT, Het and KO mice. **d** Two-sided Kruskal–Wallis test; $P = 0.012$. Mann–Whitney $U$ test for further comparison; $P_{WT\ vs\ Het} = 0.36$, $P_{WT\ vs\ KO} = 0.0044$, $P_{Het\ vs\ KO} = 0.097$. $n = 11$, 8 and 8 for WT, Het and KO mice. **e** Two-sided Kruskal–Wallis test; $P = 0.0055$. Mann–Whitney $U$ test for further comparison. $P_{WT\ vs\ Het} = 0.0048$, $P_{WT\ vs\ KO} = 0.8$, $P_{Het\ vs\ KO} = 0.012$. $n = 11$, 8 and 8 for WT, Het and KO mice.

occasionally began to dig empty holes in an arbitrary manner as if they were searching for the hidden odour source (Fig. 10b–d; Supplementary Movie 8). Thus, PDEi administration unveiled haploinsufficiency as momentary hyposmia in Het mice during continual sniffing, in line with the findings in the ex vivo experiments (Fig. 6c–h).

Overall, our data indicate that cAMP buffering by OMP is essential for both the maintenance of repetitive responses and for basal sensitivity during olfaction.

## Discussion

Previous biophysical studies have indicated that cAMP diffusion within the olfactory cilia is restricted at the site of production[17], and that the speed of diffusion is slowed down at high concentrations of cAMP within ORNs[49]. Cyclic AMP buffering by

OMP is a potential entity for this regulation of focal cAMP signalling.

Single-puff stimulation at 20 psi often causes a delayed and prolonged increase in firing, similar to EOG or patch-clamp recording results described in previous reports[27–29,31,32]. Based on these delayed kinetics, previous studies have implied that OMP is involved in sensory transduction, while the sites of action of OMP have remained an open question[27]. Therefore, the direct cAMP buffering action can explain the mechanisms by which OMP sharpens olfactory signalling. The loss of OMP leads to prolonged and wider spectral responses of ORNs under a single stimulus and eventually to the adaptive loss of responses under longer or stronger stimulation due to persistent membrane depolarisation, inactivation of firing machinery and simultaneous diminution of the driving force to generate receptor potentials.

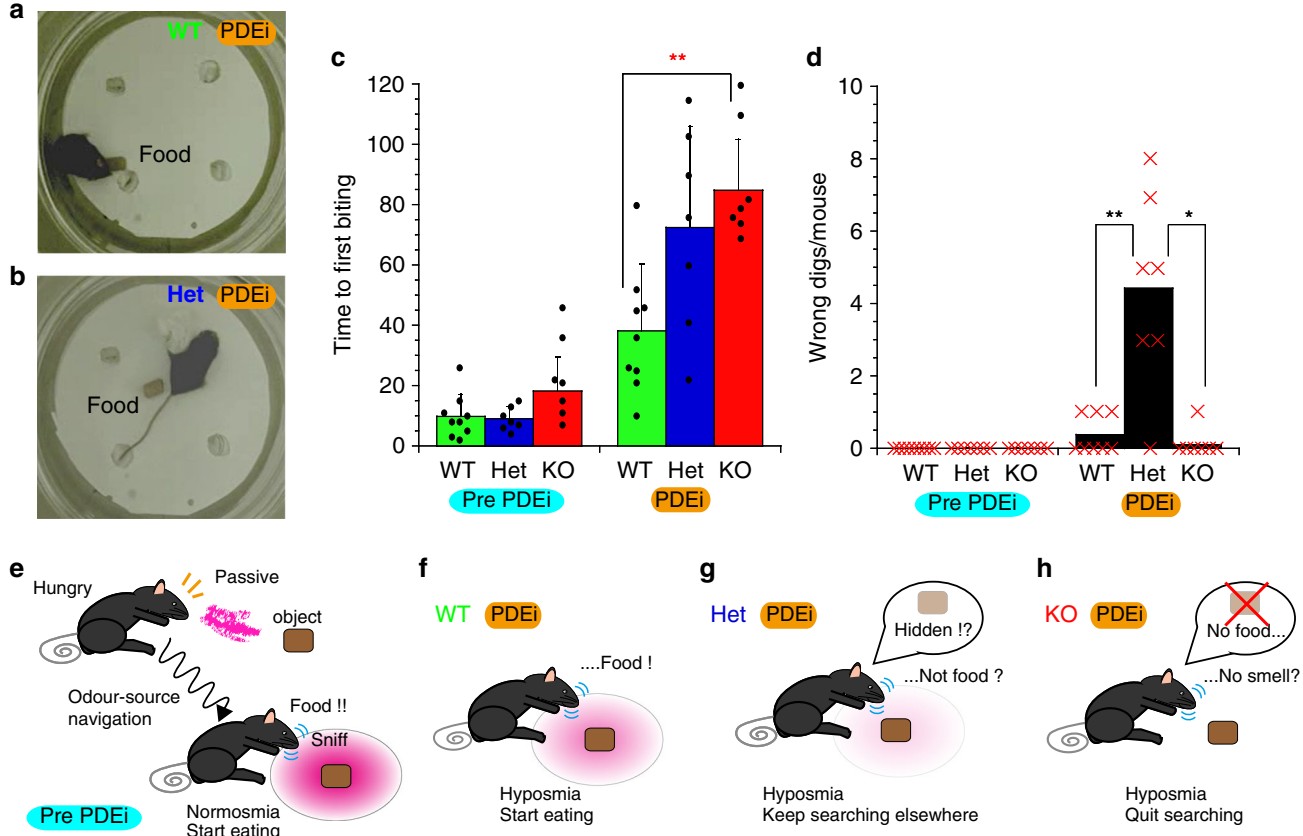

**Fig. 10 OMP-deficient mice lose olfactory sensitivity by cAMP overload. a** Eating behaviour of a WT mouse challenged with a PDEi. **b** Random-search behaviour of a Het mouse challenged with a PDEi. **c** Time to the first biting of visible food without and with a PDEi (control vs post PDEi). Mean ± s.d. **d** The total number of wrong digs by mice challenged without and with a PDEi (control vs post PDEi). Diamonds and X-marks indicate the numbers of mice used in (**c**, **d**). $n = 9$, 7 and 7 animals in (**c**) and (**d**) for WT, Het and KO mice, respectively. *$P < 0.05$; **$P < 0.01$. **e** Strategy of the visible odour-source navigation of WT, Het and KO mice in (**c**) prior to PDEi administration. **f–h** Interpretation of the behaviours of **f** WT, **g** Het and **h** KO mice challenged with a PDEi. *$P < 0.05$, **$P < 0.01$. Statistics: **c** across groups (Pre); one-way ANOVA; $F(3, 22) = 4.999$, $P = 0.017$; post hoc Bonferroni test, $P_{WT\,vs\,Het} > 0.999$, $P_{WT\,vs\,KO} = 0.036$ and $P_{Het\,vs\,KO} = 0.036$: across groups (Post); one-way ANOVA, $F(3, 22) = 8.196$, $P = 0.0025$; post hoc Bonferroni test, $P_{WT\,vs\,Het} = 0.04$, $P_{WT\,vs\,KO} = 0.0028$ and $P_{Het\,vs\,KO} = 0.85$: among groups (Pre vs Post); paired two-sided $T$ test; $P_{Control\,vs\,PDEi} = 0.0031$, 0.0034 and 0.00074; $n = 9$, 7 and 7 for WT, Het and KO mice, respectively. **d** Across groups (Pre); ND (not determined): across groups (Post); two-sided Kruskal–Wallis test, $P = 0.00041$, Mann–Whitney $U$ test, $P_{WT\,vs\,Het} = 0.0012$, $P_{WT\,vs\,KO} = 0.44$ and $P_{Het\,vs\,KO} = 0.0017$: among groups (Pre vs Post); $P_{Control\,vs\,PDEi} = 0.076$, 0.0017 and 0.39 for WT, Het and KO mice.

In this study, mechanical stimuli induced drastic silencing of KO neurons, whereas odourant mixtures merely suppressed firings of KO neurons, most likely due to the weak matching between the odourant species and the reception spectrum of recorded ORNs. Moreover, recent studies suggest that odourant perception is accomplished by utilising sniff-based phase information to identify the odour[20]. Mechanical stimuli might rather effectively induce cAMP production, and OMP could support such phase-dependent odour processing through the cAMP-buffering capacity.

In addition to sustained depolarisation of the membrane or inhibition of CNG channels, adaptation mechanisms in ORNs involve various signalling molecules that we did not specifically examine in this study, including calmodulin-dependent kinase II, ACIII, PDE, $Ca^{2+}$-ATPase and cGMP. In addition to the factors within ORNs, even synaptic transmission at olfactory glomeruli is modified during sustained stimulation[9–14,18]. Among the various adaptation pathways, OMP may indirectly enhance NCX activity[1], as it can also counterbalance $Ca^{2+}$-dependent adaptation after $Ca^{2+}$ influx in concert with $Ca^{2+}$-ATPase[50]. Cyclic AMP buffering by OMP contributes to the relatively short-term protection against repetitive stimuli during sniffing on the order of

seconds in concert with the basal action of a PDEi at tens of seconds. Moreover, the interactions of OMP with molecules in different types of adaptation, including those upstream[31,32] and downstream[9–14] of cAMP production, need further investigation.

To determine the biochemical parameters, we applied several approximations, including curve fitting (Fig. 2d, e; Supplementary Fig. 3i). Because OMP seems to have two sites for binding cAMP with different affinities, it might be possible that the initial binding of cAMP to OMP affects the next cAMP-binding process. Such a sequential binding process can induce hysteresis during association and dissociation, which may account for the partial inhibitory effects observed in the competitive assay (Fig. 2e; Supplementary Fig. 3i). Further structural and biochemical investigations of the binding dynamics will more accurately elucidate the actions of OMP at certain phases of olfaction.

It is also known that the genetic ablation of OMP disturbs the refinement of an olfactory neuronal circuit[51,52], while this histological process requires basal activities of cAMP-gated channels and cAMP-dependent kinase[31,43,53–55]. Reportedly, the basal cAMP pool might be disturbed in OMP-KO neurons[31]. Because the cAMP-binding capacities of OMP appear to differ at different basal concentrations of cAMP (Fig. 3f, g), OMP may participate

in cAMP-dependent axonal targeting by modulating the basal cAMP pool[31,47,51,54–56].

Hence, cAMP buffering by OMP should play a fundamental role in stabilising the basal perturbation of cAMP, although the previously hypothesised possible interactions between OMP and other molecules[1,31,32,52] are certainly possible.

Every mature ORN expresses a unique odourant receptor, which may affect firing patterns[31,34,35,43,54,55]. Variations in the spontaneous firing frequencies of ORNs have been shown in slice preparations[31,45]. In this study, the difference in firing pattern was statistically significant between Het and KO neurons. Moreover, there remains the possibility of suctioning axons[41] for measurement that are adjacent to ORNs, contributing to the heterogeneity of firing activity, as these axons likely have different firing patterns. Accordingly, further experiments on the heterogeneity of spontaneous or evoked firing patterns of ORNs are needed[2,31,34,35,43,54–57].

In this study, we used PDEi to disturb ORNs for in vivo investigation of odour-source searching ability. Although intranasal PDEi administration was employed to induce topical effects on the olfactory epithelium, side effects to the olfactory bulb or any other parts of the brain through the ethmoidal bone cannot be fully eliminated. Considering that the clinical use of PDEi induces side effects in olfaction[48], further detailed investigation of cAMP effects across olfaction-processing neural networks is necessary in animals and humans.

Finally, OMP exists solely in vertebrates, and its genome-wide duplication occurred in concert with teleost evolution[58]. Such phylogenetic conservation of the CNB sites also accentuates the physiological importance of OMP. The maintenance of a resilient neural response by cAMP buffering should have evolutionary advantages for survival in nature. For example, odour-source localisation, which was impaired in OMP-KO mice, must be important for searching for food and predators, as well as for long-distance migration for spawning[23,59].

Furthermore, OMP expression is not an exclusive marker of ORNs and is widely distributed in the brain[60–63]. Depending on the activation patterns and magnitudes of $G_s$ protein-coupled receptors in different subcellular parts of neurons, cAMP buffering could diversify the information flow through neural networks, i.e., counterbalancing adaptation, punctuating repetitive responses or smoothing minor perturbation. However, further investigation on this topic is needed.

## Methods

**Subject animals**. All experimental animals were treated in accordance with the guidelines of the Animal Experiments Committees and the Ethics Committees of Kyoto University and Kurume University under the committees' approval. The OMP-KO (RBRC02092) male mice[25] were obtained from RIKEN with permission from Prof. Peter Mombaerts at the Max Planck Research Unit for Neurogenetics. We used C57BL/6 male mice (Japan SLC, Inc., Shizuoka, Japan) as wild-type animals. The animals were housed in groups of six per cage in the same room under a 12/12-h light/dark photoperiod at 24 °C and 60% humidity. The animals used in the behavioural experiments were housed individually after weaning. The animals were fed ad libitum, except when otherwise specified during the odour-source-locating test. The animals used in the electrophysiological and molecular biological studies were sacrificed via rapid decapitation with sharp blades. The animals used in the histological experiments were deeply anaesthetised via inhalation of isoflurane (1–5% v/v in air) in a secure plastic container prior to perfusion fixation and decapitated after fixation. The animals were genotyped post experimentally and blindly.

**Taste preference test**. Preference for sucrose (Sigma-Aldrich, MO, USA) was measured by a handmade lickometer[64]. Briefly, sucrose was dissolved in water (100 mM), and the sucrose solution was applied in a plastic bottle plugged with a silicon cap and a leak-resistant metal spout containing two steel balls (CLEA Japan, Inc., Tokyo, Japan). A solution spout was inserted into a 10 × 10 × 10-cm plastic container, and mice were free to approach the spout. Each lick established an electrical circuit between the mouse and the spout of a drinking solution container, generating junction potentials between the tongue and the spout, which were

amplified and recorded using the PowerLab26T amplifier and LabChart 10 software (ADInstruments, Sydney, Australia). The number of licks within the 5-s period was counted from the first lick.

**Odour-source-locating test**. To evaluate the abilities of the mice to locate the odour source, mice were trained to associate the 100 mM sucrose reward with the odour of $(R)-(+)$-limonene (Sigma-Aldrich, USA). Limonene was prepared at 10 v/v% in mineral oil (Sigma-Aldrich, USA). Mice at the age of 8 weeks or more were deprived of water for approximately 8 h prior to the training session. The training session comprised a 1-min presentation of the sucrose solution to a water-deprived mouse as a reward five times at 1-min intervals. At each presentation, a swab containing 50 μL of limonene/mineral oil was presented close to the drinking spout. After training, the mouse was given a freely accessible water bottle until the next day, and this training was repeated for 8 days. The odour-source localising ability was tested in a cage (width × depth × height = 25 cm × 40 cm × 20 cm), wherein a bottle containing the sucrose solution (reward) and a metal container with limonene-odour-emitting filter paper (100 μL of 10 v/v% limonene) were suspended in one corner, and an empty bottle (no reward) was suspended in the other corner of one end of the cage. For the first session of the test, a trained mouse deprived of water for 8 h prior to the test was placed at the other end of the cage. The time that elapsed before the mouse first licked the reward spout and the number of licks of the empty spout were recorded. For the next session of the test, the mice were pre-exposed to 1 v/v% limonene in a waiting cage. Then, the mouse was placed in a cage presaturated with 1 v/v% limonene in the background that was wrapped with a polyethylene film (Saran Wrap, Asahi Kasei Corporation, Chiyoda, Tokyo) and allowed to search for the reward spout situated close to the 10 v/v% limonene odour. The locations of the sucrose and the empty bottle were switched in different experiments. To further evaluate the abilities of the mice to locate the food reward by olfaction without visible cues, we designed the following experiments based on the Barnes Maze Test[65]. The apparatus is a round glass chamber with a diameter of 220 mm and a height of 160 mm. The floor of the chamber has four holes, each of which has an orifice with a diameter of 25 mm and a depth of 70 mm. The apparatus is screened with 800-mm-high plain cardboard. All sessions were recorded with a digital video recorder and analysed offline. A piece of food (CRF-1, Oriental Yeast Co., LTD, Kyoto, Japan) was placed deep within one of the four disposable spirometry mouthpieces made of paper, which were plugged with two wads of tissue (KimWipes, Kimberly Clark Corp., TX, USA). Mice deprived of food for 2 days were individually placed at the centre of the platform and allowed to search for the food via the odour for 2 min during each session. Each session was repeated five times. The mouse was tasked with removing the two wads of tissue to reach the food reward deep inside one of the four holes; the correct hole was changed each session. Generally, the mouse poked their noses into the tube and sniffed without removing the wads of tissue. Once the mouse located the food reward, it removed the wad of tissue with its mouth and forelimbs, which is a process referred to as digging. The first removal of the wad was defined as a single dig, and the successful removal of both wads to reach the food reward was defined as a double dig. The KO mice occasionally removed the wads from the wrong holes, which contained no food, and this action was defined as a wrong dig. None of the mice were acclimated to the chamber prior to the sessions because learning the paradigm of searching for the reward was a part of the behavioural assay. In the dig count analysis, single, double or wrong digs were each independently assigned a score of 1, whereas no dig was assigned a score of 0 for the one-sided $T$ test. The video data were analysed offline. The 60- or 120-s video data were separated into pictures at the rate of one frame per second in JPEG format, converted into binary data and stacked into one image to construct a heatmap of the time spent in the assay chambers. The maximal signals were normalised to correspond to 15 /60 s or longer.

In the preliminary experiments, following 1 mg/kg i.p. administration, the mice were sedated to prevent mobility, and the olfactory-specific PDE4 inhibitor rolipram (300 μM, Sigma-Aldrich, MO, USA) was administered intranasally[66]. The mice were trained approximately ten times to correctly dig two wads prior to intranasal drug administration. We held each mouse by hand and loaded a pipettor with 20 μL total per mouse (four drops × 2.5 μL = 10 μL/nostril) of saline with rolipram or saline containing 0.5% DMSO. Between the administration procedures, the mice were held backwards for 15 s to allow the droplet to fully contact the olfactory epithelium and then allowed to rest for 10 min before starting the odour-source-locating test session using the same protocols described above. The mice were retested after 24 h to confirm recovery. Once a food chunk was explicitly placed in front of the mouse, the mouse was free to eat within a 2-min session; if the mouse ultimately did not eat, the search time was recorded as 2 min.

**In vivo recording from the olfactory epithelium and nerve**. To record the electrical activities from the main olfactory epithelium or the nerves of ORNs in vivo, we modified a previously described method[44]. Mice at the age of 8 weeks or more were anaesthetised with an intraperitoneal injection of 200 μL of a mixture of medetomidine hydrochloride, midazolam and butorphanol tartrate (0.075 mg/mL, 2 mg/mL and 0.1 mg/mL, respectively) and kept warm during experiments. The left nasal bone was carefully scraped with a dental drill, and the olfactory nerve bundle layer of the olfactory epithelium was exposed, which was kept humid with normal Ringer solution. The skin was ground. A plastic puffer pipette tip was inserted into

the left nostril, and an air puff scented with food odours was applied from a diaphragm pump (Picospritzer II; General Valve Corp., NJ, USA) under the control of an electrical stimulator (Nihon Kohden, Tokyo, Japan). A ground electrode was placed beneath the skin of the head. A test odour was prepared as a 1-cm$^2$ piece of filter paper containing 100 μL of 10 v/v% limonene in mineral oil. Food odours were prepared by using 2 g of freshly crushed food chunks to which the mice were accustomed (CRF-1, Oriental Yeast Co., LTD, Kyoto, Japan). The prepared odour samples were placed in a sealed 5-mL plastic tube with paper filter (KimWipes, Kimberly Clark Corp., TX, USA), which was interconnected between a puffer pipette and Picospritzer II via silicon tubing (inner $\phi = 1$ mm). For EOG recording, a tungsten electrode (Nihon Kohden, Tokyo, Japan) was inserted into the main olfactory epithelium by penetrating though the epithelial layer. For nerve recording, the multiunit recording pipette, which had a tip size of ~20 μm, was filled with Ringer solution and placed on the nerve layer with slight suction without penetrating though the epithelial layer. The signals were amplified with a current-clamp amplifier (Axoclamp 900A, Axon CNS, Molecular Devices, CA, USA) with acquisition software and an AD/DA converter (pClamp10 and Digidata 1440 A, Molecular Devices, CA, USA) and digitally filtered between 10 Hz and 10 kHz offline. The multiple unit activities were confirmed by spike clustering using pCLAMP by separating the data into 10-ms episodic events and then overlaying the results. The amplitudes, locations, widths and rise times of the spikes were measured, and no characteristic relationships across the clusters of spikes were observed. ORNs are supposedly heterogeneous even when the same odourant receptors are expressed[45]. Thus, segregation of the different unit activities by spike sorting was difficult. Alternatively, we detected spike peaks by thresholding[34,44]. Most of the signals recorded were signals of small baseline fluctuations, and only 2% at most were the signals corresponding to spikes. Therefore, we defined the threshold level as +3 s.d. (99% of all data points) of all signal fluctuations. The s.d. values were stable before, during and after stimulation (Supplementary Fig. 7e). Thus, the threshold during the 20-s pre-stimulatory period was used to evaluate the basal noise level to detect the subsequent peaks in the recording. A separation valley of less than 50% of that of the adjacent peaks, including those of complex action potentials, was accepted. The spontaneous events were stable before and after stimulation (Supplementary Fig. 7f). On these criteria, the relative odourant responses were determined from the number of spikes during a 20-s period of stimulation compared with that in the prestimulus 20-s bin. Note that the responses of multiunit activities were not always recorded, especially when using odours, implying that ORNs insensitive to limonene or food odours and mechanical stress exist.

**Ligand-binding computer simulation.** We used the Protein Data Bank[67] (PDB) to obtain the NMR-based solution structure of mouse OMP[28] (Model 1 from a 1ZRI file). The OMP data were further modified by adding polar hydrogen atoms and rendered into an analysis grid using AutoDockTools on a Python platform[68]. The structural data of cAMP (CID 6076) and cGMP (CID 92823) were obtained from the NCBI PubChem Database in a 3D-structure-data file format, which was then converted into PDB format using OpenBabel[69]. The docking simulations were performed using AutoDock Vina[70] (The Scripps Research Institute, CA, USA). The results were analysed with Python Molecular Viewer[68], yielding molecular surface rendering and rendering into PNG image files. We tested the validity of AutoDock Vina by predicting the affinity values of the following cAMP-binding proteins (PDB-ID): *M. musculus* hyperpolarisation-activated cyclic nucleotide-gated channel subtype 2 (HCN2: 1Q43), *B. taurus* protein kinase A (PKA; 1RL3) and *E. coli* catabolite gene activator protein (CAP: 1G6N). The values obtained in the most compatible conformations with cAMP-binding domains was as follows: $-25.9$ kJ/mol for HCN2, $-30.5$ kJ/mol for PKA and $-39.3$ kJ/mol for CAP (Supplementary Fig. 1b–f). Here, the PDB data of HCN2, PKA and CAP were modified by removing the water molecules and coexisting cAMP. Selenomethionine (MSE in the PDB data) was not edited because the N terminus of OMP was located at the edge of the globular structure. All other selenomethionine entries in the PDB text files were changed to methionines by performing a series of edits. We did not consider secondary conformational changes resulting from consecutive cAMP binding to OMP.

**Comparison of OMP across vertebrate species.** The amino acid sequences were aligned with STRAP[71] (http://www.bioinformatics.org/strap/). An evolutionary genetic analysis was conducted using MEGA5[72]. The protein data, including some predicted sequences, were obtained in FASTA format from NCBI. The accession numbers are as follows: *Mus musculus* (mouse, NP_035140.1), *Rattus norvegicus* (rat, NP_036748.1), *Homo sapiens* (human, NP_006180.1), *Trichechus manatus latirostris* (manatee, XP_004382047), *Canis lupus familiaris* (dog, XP_005633517.1), *Dasypus novemcinctus* (armadillo, XP_004458104), *Gallus* (chicken, XP_004938999.1), *Anolis carolinensis* (anole lizard, XP_003227578), *Alligator sinensis* (alligator, XP_006022032.1), *Xenopus laevis* (African clawed frog, NP_001079238.1), *Danio rerio* (zebrafish, AAL87664.1), *Oncorhynchus nerka* (salmon, AB490250) and *Latimeria chalumnae* (coelacanth, XP_006004323).

**Generation of cDNA constructs.** We used PfuUltra Fusion DNA polymerase (Agilent, CA, USA) and Q5 High-Fidelity DNA polymerase (New England Biolabs,

MA, USA) for the PCR analysis. We used a pCI mammalian expression vector (Promega, WI, USA) for exogenous expression in HEK293 and HEK293T cells (ATCC, VA, USA). The primers and the step-by-step methods used to construct the vectors are graphically summarised in Supplementary Fig. 2a and 2b. Briefly, cDNAs of mouse OMP (NM_011010) and Renilla luciferase (Rluc) were amplified via PCR and then inserted into a pCI vector. Rluc lacking the stop codon was inserted immediately upstream of OMP in-frame. Gene cassettes of IRES-GFP or IRES-DsRed were digested from pIRES2-EGFP or pIRES2-DsRed (Clontech, CA, USA) and inserted into a pCI vector containing the PCR products of CNGA2 channel cDNA (NM_007724). The OMP mutants were generated via overlap PCR using the corresponding combinations of primers by referencing a previous report[73]. The plasmids containing Rluc cDNA (pGL4.74 [hRluc/TK]; Promega, WI, USA) were the same as those used in the dual-reporter assay described below.

**Heterologous cDNA expression.** HEK293T cells were cultured in Dulbecco's modified Eagle's medium (D-MEM, Wako Pure Chemical, Osaka, Japan) supplemented with 10% foetal bovine serum (FBS: Sigma-Aldrich, MO, USA) without antibiotics at 37 °C and 5% $CO_2$. Different combinations of a total of 2 μg of plasmids at equivalent ratios were transfected into HEK293T cells using Lipofectamine 2000 (Thermo Fisher Scientific, MA, USA). After transfection, the HEK293T cells were incubated for 24 h, dispersed with TRYple (Thermo Fisher Scientific, MA, USA) and plated on coverslips at 10–20% confluency in D-MEM with 10% FBS. Then, the cells were incubated for at least 8 h to allow settling prior to the electrophysiological and histological experiments unless otherwise indicated.

**Western blotting.** To confirm heterologous OMP expression, HEK293T cells were transfected with plasmids expressing OMP$^{Wt}$ and OMP$^{Mut}$, incubated overnight, resuspended in divalent-ion free PBS (PBS(−)) and lysed via sonication in ice-cold water for 5 min. The cell lysates were centrifuged for 5 min at 20,000 g and 4 °C, and the supernatants (5 μL) were collected, mixed with Laemmli sample buffer (Bio-Rad Laboratories, CA, USA), heated at 95 °C for 5 min and used for subsequent Western blotting. The samples were separated via 12% SDS-PAGE. Then, the proteins were transferred to nitrocellulose membranes using a semidry blotting apparatus (Bio-Rad Laboratories, CA, USA) for 90 min at 16 V. The nitrocellulose membranes were pretreated with 5% skim milk/0.1% Tween in Tris-buffered saline (TBST: 10 mM Tris, 150 mM NaCl, pH 7.6) blocking solution for 30 min at 25 °C, washed with TBST and incubated overnight with an anti-OMP antibody (raised in goat, 1:1000) at 4 °C. The membranes were washed, incubated with an anti-goat secondary antibody conjugated with horseradish peroxidase (HRP, ab6885, Lot#GR3296933-2, Abcam, Cambridge, UK) at a 1:10,000 dilution for 1 h at 25 °C and then washed with TBST. Then, the membranes were washed meticulously. The immunoreactivity was detected using ECL prime Western blot detection reagent (GE Healthcare Biosciences, NJ, USA). The images were captured using a ChemiDoc XRS (Bio-Rad Laboratories, CA, USA).

**Immunohistochemistry and immunocytochemistry.** Male C57BL/6 mice (postnatal 4 weeks) were perfusion-fixed with PBS containing 4% paraformaldehyde (4% PFA). The collected samples were post-fixed with 4% PFA for 4 h, cryoprotected via overnight incubation in a 30% w/v sucrose solution containing PBS, mounted in Tissue-Tek O.C.T. Compound (Sakura Finetek, Tokyo, Japan) and sectioned into 30-μm slices using a cryostat (CM3050S, Leica Microsystems, Wetzlar, Germany). The resulting sections were incubated overnight at room temperature in an appropriate blocking solution containing the primary antibodies. Then, the sections were washed with PBS containing 0.3% Triton X-100 (PBS-X) and incubated with the anti-goat secondary antibody (Alexa Fluor 594 conjugated, 1:200, A11058, Lot# 1842799, Thermo Fisher Scientific, MA, USA, in blocking solution) for 1.5–2 h. After the secondary incubation, the sections were washed with PBS-X, mounted onto MAS-coated glass slides (Matsunami Glass Ind., Ltd., Osaka, Japan), coverslipped using Vectashield antifade reagent (Vector Labs, CA, USA) and tightly sealed. For immunocytochemistry, the transfected HEK293T cells cultured on coverslips were immersion-fixed in ice-cold 4% PFA for 15 min, washed three times with PBS for more than 30 min, incubated with primary antibodies at 24 °C for 6 h, washed three times with PBS and incubated with secondary antibodies at 24 °C for 1 h. Then, the coverslips were washed with distilled water, dried, mounted on a slide (Matsunami Glass Ind., Ltd., Osaka, Japan) with 50% glycerol and tightly sealed. The fluorescent signals were detected using an FV1000 confocal laser scanning microscope (Olympus Corporation, Tokyo, Japan) or a fluorescence microscopy unit and analysed with the appropriate software (BIOREVO BZ-9000, KEYENCE, Osaka, Japan). Anti-OMP primary antibodies (019-22291, Lot#PDP1534, Wako Pure Chemical, Osaka, Japan) were used at a 1:400 dilution. Omitting the primary antibodies from the incubation solution resulted in the lack of significant fluorescent signals.

**Optical spectrum analysis of BRET.** The plasmids containing cDNAs of the Rluc-OMP fusion genes (5 μg) were transfected into HEK293T cells plated in 10-cm dishes using Lipofectamine 2000 according to the manufacturer's protocol. The cells were incubated at 37 °C and 5% $CO_2$ for 24 h and collected via centrifugation at 400 g for 5 min; the supernatant was discarded. The cell pellet was resuspended in divalent cation-free PBS (PBS(−):100 μL) and disrupted by sonication at 3.1 kHz

for 5 min in an ice-cold water bath. The lysate was centrifuged to remove the cell debris. The supernatant containing the proteins was further rinsed using Vivaspin ultrafiltration columns (3 kDa: Sartorius Stedim Biotech, Germany) and then resuspended in 100 μL of PBS(–) for the following experiments. The relative OMP concentration in the supernatant was determined by Western blotting using an anti-OMP antibody (Wako Pure Chemical, Osaka, Japan). For the BRET analysis, the 1-mL test solution was prepared in divalent cation-free PBS (~1000-fold dilution of the original lysate to yield calibrated BRET signals greater than $10^5$ in arbitrary units) and supplemented with 8NBD-cAMP (BIOLOG Life Science Institute, Bremen, Germany) in a series of dilutions (200 nM–10 μM) or TNP-ATP (Wako Pure Chemical, Osaka, Japan). The test tubes were placed in a block thermostat maintained at 24 °C, and the thermostat block was completely covered with a C-mount laser housing attached to detecting fibre optics. The experiments were performed in a self-built darkroom. The background noise signals were detected through the fibre optics, relayed to a spectrometer (Optics 250is, Bruker K. K., USA), enhanced with an image intensifier (M7971-81, Hamamatsu Photonics, Shizuoka, Japan), captured with an infrared digital CCD camera (ORCA-R2, Hamamatsu photonics, Shizuoka, Japan) and recorded at 5-s intervals with a tenfold gain. We confirmed that 8NBD-cAMP had no autoluminescence in the absence of Rluc or the presence of Rluc lacking substrate (the basal level was subtracted where necessary). After the background calibration, 1 μL of the Rluc substrate (coelenterazine: Stop & Glo, Promega, WI, USA) was quickly applied, and the emitted light at wavelengths of 350–650 nm was detected in the same manner. The signals were corrected for background noise to yield the BRET ratio (Fig. 2c; Supplementary Fig. 3a, b), and the nonspecific components associated with Rluc alone were further subtracted from the BRET ratio (ΔBRET: Supplementary Fig. 3c–f) and analysed offline. The BRET ratio was calculated from the values at 536/480 nm[74]. The spectrum for each recording was the average of three samples over five sessions. No PDE inhibitor, AC inhibitor or proteinase inhibitor cocktail was added to the OMP-containing solutions because these molecules might also interact with OMP as nucleotide analogues. A small ΔBRET signal remained in all mutants at 1–2 μM (Fig. 2i). According to the docking simulations, OMP might have other cAMP-binding sites with much lower binding affinities (the simulated ΔG values were ~−24 kJ/mol) on its other facets (Supplementary Fig. 1i, j). These extra motifs, steric interactions between pockets A and B or conformational allostery of OMP might contribute to the residual ΔBRET.

**Competitive assay**. First, Rluc-OMP$^{Wt}$ was incubated with 8NBD-cAMP at a near-saturating concentration ($L = 5$ μM) and ΔBRET was measured at 24 °C in the presence of luminescent substrates as described above. Then, serially diluted nucleotides (cAMP and cGMP from Sigma-Aldrich, MO, USA) were added as competitive inhibitors. For cAMP and cGMP, the inhibition of ΔBRET (ΔΔBRET) was plotted and fitted with a single exponential curve to yield the IC$_{50}$ (1 μM for cAMP and 11 μM for cGMP; Supplementary Fig. 3i). The dissociation constants of cAMP and cGMP as inhibitors ($Ki$) were obtained with the Cheng–Prusoff equation $Ki = IC_{50} \times Kd / (L + Kd)$. The apparent dissociation constant ($Kd$) of 8NBD-cAMP as a ligand ($L$) was approximated as the $EC_{50}$ (1.9 μM) from the dose–response curve of Rluc-OMP$^{Wt}$ (Fig. 2d). The experimental values of ΔG$_{binding}$ were calculated with the Gibbs free energy equation (ΔG = −RT•ln $Ki$), where R is the gas constant (8.3 J/mol/K), T is the absolute temperature (297 K in this study) and ln is the natural logarithm. Thus, the IC$_{50}$ values above correspond to the changes in the free energy of binding (−37.3 kJ/mol for cAMP; −31.4 kJ/mol for cGMP) that are slightly larger than but mostly the same as the simulated ΔG.

**Computation of differential equations**. The three-parameter ordinary differential equations in mathematical modelling (Supplementary Note) were solved using the fourth-order Runge–Kutta method.

**Electrophysiology using HEK293T cells**. Transfected cells were identified with coexpressed fluorescent proteins via fluorescence microscopy. We employed cell-attached or whole-cell voltage-clamp configurations at 25 °C using either a patch- (Axoclamp 200B, Molecular Devices, CA, USA) or a current-clamp amplifier (Axoclamp 900 A, Axon CNS, Molecular Devices, CA, USA) with acquisition software and an AD/DA converter (pClamp10 and Digidata 1440 A, Molecular Devices, CA, USA). The extracellular solution contained 110 mM NaCl, 0.5 mM MgCl$_2$, 30 mM KCl, 1.8 mM CaCl$_2$, 5 mM HEPES and 14 mM glucose, and the pH was adjusted to 7.4 with NaOH. The patch pipette solution contained 130 mM KCl, 10 mM NaCl, 5 mM EGTA-KOH, 0.5 mM MgCl$_2$, 2 mM Na$_2$-ATP and 5 mM HEPES, and the pH was adjusted to 7.2 with KOH. The pipettes had resistances of 3–5 MΩ. Recordings were made by holding the pipette potential at −80 mV to observe the single-channel activity of CNG and record the whole-cell CNG currents, which were sampled at a rate of 10 kHz, high-pass filtered at 3 Hz and low-pass filtered at 1 kHz to analyse the unitary activities or low-pass filtered at 5 kHz to analyse the whole-cell currents.

**Manipulation of cAMP production in HEK293T cells**. For the UV uncaging of photosensitive caged cAMP (Wako Pure Chemical, Osaka, Japan), we principally

followed the methods described in a previous report[75]. Caged cAMP was prepared at 400 mM in DMSO and stored in the dark at −20 °C. The stock was diluted in a KCl-based internal solution at 1 mM immediately prior to use. The recording pipette was filled with the working solution under dim light. After the whole-cell recording configuration was completed, the caged cAMP transfused from the recording pipette was photolysed with 330–385-nm light from a xenon short-arc lamp (75 W) through a dichroic mirror (400-nm cut-off, U-MWU2, Olympus Corporation, Tokyo, Japan) by opening an electromagnetic shutter controlled by a stimulator for 1 s (Nihon Kohden, Tokyo, Japan). The spontaneous decomposition of caged cAMP by transillumination was avoided as much as possible by illuminating the preparation through a green interference filter (500–530 nm, bandpass and 90% transmission, Nikon, Tokyo, Japan) during electrode positioning. The recordings were performed at 25 °C.

**Loose patch recording of ORNs**. The olfactory epithelia of P0-4 mice of either sex from the OMP-KO and WT mouse groups were frontally sliced at a thickness of 300 μm using a slicer (PRO10, DOSAKA EM Co., Ltd., Kyoto, Japan) in ice-cold dissecting solution containing the following compounds (mM): 130 mM NaCl, 4.5 mM KCl, 2 mM CaCl$_2$, 5 mM PIPES-Na and 34 mM glucose at a pH = 7.4 adjusted with NaOH. In this study, the OMP gene was knocked out by replacing the locus with GFP. All recordings were obtained at 36 °C using a patch-clamp amplifier (Axoclamp 700B, Molecular Devices, CA, USA) with pClamp10 and Digidata 1440 A. The bath and pipette solution contained the following compounds: 155 mM NaCl, 2.5 mM CaCl$_2$, 1 mM MgCl$_2$, 17 mM glucose, 10 mM HEPES and 5 mM KOH at a pH = 7.4 adjusted with NaOH. The extracellular recordings were sampled at a rate of 10 kHz, low-pass filtered at 5 kHz and high-pass filtered at 3 Hz. The recording pipettes had a resistance of approximately 5 MΩ. The absolute values of negative peaks were automatically measured. The continuous recordings were subdivided into 15-ms episodic events using pCLAMP, and spikes were overlaid at the onset. The waveforms varied between spontaneous and evoked firing. Thus, all the spikes were detected by thresholding according to the following criteria: peaks greater than the noise level and a separation valley of less than 50% of that of the adjacent peaks. Thus, the doublet and the contaminated firings of adjacent ORNs were all collected, but small field potentials (below 50% of the preceding firing) of adjacent neurons were mostly eliminated from the analysed data. For flash uncaging of cAMP, we started the recording under dim light with an external solution containing 1 mM membrane-permeable caged cAMP ((7-dimethylaminocoumarin-4-yl)methyl-8-bromoadenosine-3',5'-monophosphate; DMACM-caged 8-Br-cAMP; BIOLOG Life Science Institute, Bremen, Germany). The UV was illuminated for 20 s. The transient local flow with external solution containing odourant mixtures was applied to the ciliary surface at 2 Hz with 200-ms duration and 300-ms intervals for 10 s using a puffer pipette (tip diameter, ~2 μm) that had been tangentially arranged in front of the surface of the sliced olfactory epithelium at a distance of ~20–50 μm. The estimated flow volume was approximately 2 μl per second. Odourant mixtures were freshly prepared, including amyl acetate, cineole, limonene, acetophenone, heptanal and lilial (Sigma-Aldrich, MO, USA) at 100 μM each in the extracellular solution according to a previous report with modifications[47]. Mechanical stimulation with the external solution was applied from another puffer pipette (tip diameter, ~20 μm) placed in front of the surface of the sliced olfactory epithelium at an appropriate distance. The puffer pipette was connected to Picospritzer II (General Valve Corp., NJ, USA). The solution flow was controlled by a stimulator (Nihon Kohden, Tokyo, Japan). Stock solutions of BPIPP (R&D Systems, MN, USA), SQ 22,536 and IBMX (Sigma-Aldrich, MO, USA) were produced in DMSO and diluted in the bath solution for use; db-cAMP (Sigma-Aldrich, MO, USA) was dissolved in water and freshly prepared. The supplemented solution was added to the recording chamber via peristaltic perfusion, and the recording was performed within 5 min of the onset of perfusion.

**Illustration**. The illustrations in Fig. 3a, k, l, Fig.8a, d–f, Fig. 9a, Fig. 10e-h are originally created by NN and ATN using Adobe Illustrator.

**Statistical analyses**. The statistical analyses were performed using either Microsoft Excel (Microsoft, WA, USA) or KaleidaGraph 4 (Synergy Software, PA, USA). Statistical values and methods are indicated in the figure legends.

**Reporting summary**. Further information on research design is available in the Nature Research Reporting Summary linked to this article.

## Data availability
The relevant data are found in the paper or provided as a Source Data file; otherwise, the data are available upon reasonable request.

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

## Acknowledgements

For their assistance with the experimental preparation, documentary filing and paper proofreading, we thank Hideko Yoshitake and Akemi Sakamoto at Kurume University and Midori Sakiyama, Shuko Hashimoto, Kimiko Hayashi and Yumiko Nishinaka at Kyoto University. We would also like to acknowledge the professional manuscript services of American Journal Experts. For their constructive comments regarding our paper, we thank Dr Takeshi Imai at Kyushu University, Dr Hiroshi Kuba at Nagoya University, Dr Hiroko Takeuchi at Osaka University, Dr Koji Sato at the National Institute for Physiological Sciences, Dr Tadashi Nakamura at the University of Electro-Communications and Dr Frank L. Margolis at the University of Maryland. This work was financially supported by the Kaibara Morikazu Medical Science Promotion Foundation and Kyoto University Step-up Grants-in-Aid for Young Investigators, both to Noriyuki Nakashima, and the JSPS KAKENHI to Noriyuki Nakashima (JP15K18967, JP18K15018), Harunori Ohmori (JP20220008), Akiko Taura (JP26506010) and Makoto Takano (JP26670292).

## Author contributions

N.N., M.T. and H.O. designed and performed the experiments, collected, analysed and interpreted the data, discussed the results and wrote the paper. N.N., H.O. and M.T. designed and performed the BRET experiments. K.N. generated the mutant OMP cDNAs, performed the immunohistochemistry and western blotting analyses and processed the sequence alignment data. A.T. performed the immunostaining. A.T.-N. programmed and computed the simulation for the binding reactions and performed the behavioural experiments.

## Competing interests

The authors declare no competing interests.
