## [Peer Review File · Nature Communications]

Reviewers' Comments:

Reviewer #1:

Remarks to the Author:

OMP is a highly abundant protein in ORNs. While previous studies have demonstrated the defective desensitization of odor responses, as well as the development of ORNs, in the OMP mutant animals, the exact actions of OMP in olfaction have remained elusive. In this study, Nakashima et al demonstrated that OMP directly binds to cAMP, which facilitates a quick desensitization and sharpening of CNG activity. The loss of OMP led to excessive Ca²⁺ influx and feedback adaptation, particularly in the presence of PDE inhibitors, leading to the silencing of odor responses. At the behavioral level, the OMP mutant mice demonstrated defects in odor source searching. Together, these results show that OMP is critical for olfaction based on repetitive sniffing.

While OMP has been known for many years, this study provides the first clear-cut biochemical evidence for its role in olfaction. Desensitization is an important issue in olfaction as well as in sensory systems in general, and this study, to my knowledge, provides the first evidence that a cAMP-binding protein mediates the desensitization, by directly buffering cytosolic cAMP. As olfaction is coupled to sniff cycles, quick desensitization by OMP may be useful for repetitive odor detection. This is an important contribution toward our understanding of sensory adaptation.

Major comments:

- 1) It is quite striking that loss of OMP leads to an almost complete loss of prolonged responses. As this has never been observed in previous EOG or OB imaging studies, it is important to establish how this happens. Is this due to the feedback inhibition of CNG by CaM or depolarization-induced inactivation of sodium channels?
- 2) The behavioral experiments were performed with PDE inhibitors, but is it possible to see some effect without the inhibitor? Also, with regards to the proposed mechanism, it would be interesting to perform the same experiment with different odor concentrations. Odor concentrations should be indicated in figure legends.
- 3) The desensitization of ORNs has been extensively studied. PDEs, CaM, AC3, and CNG channels are examples. How is the OMP-dependent mechanism different from them mechanistically and physiologically? As the authors claim a new type of desensitization, they need to discuss more about this point, for example, in terms of temporal kinetics and target molecules.

Minor comments:

- 4) Overall, more experimental details should be included in figure legends. For example, it is not fully described how mechanical stimuli were applied in Figure 2.
- 5) It has been reported that mechanical responses in ORN are heterogeneous and OR-specific. For example, if nearly half of the ORNs do not respond to mechanical stimuli, how did you control for that? Were there any criteria to include or exclude a particular type of ORNs in the analysis?
- 6) In Figure 3b, 3g, 4c, and 4g, the definition of peak, I, II, and III should be indicated. Just indicate time from the odor application in these panels (e.g., 0, 5, 15, 25, 35 sec).
- 7) Figure 5d and 5e obviously should not follow a normal distribution. It is not appropriate to show mean +/- sd with these bar graphs.
- 8) Supplementary Figure 6 can be moved to main figures. Traces in d and e are too dense. Supplementary Figure 1 may also be moved too.
- 9) Typos and mistakes in English needs to be carefully corrected in the final version. For example in page 17, fundamentally should be fundamental. Also, "ecological importance of cAMP buffering" does not make sense.

Reviewer #2:

Remarks to the Author:

The absence of olfactory marker protein (OMP) in olfactory sensory neurons (OSNs) causes sustained responses to single odorant stimulation and decreased responses to repeated stimulation. The decreased response to repeated stimulation is thought to underlie behavioral olfactory deficits in OMP knockout (KO) and heterozygous (Het) mice. However, how OMP alters olfactory transduction is unknown. The article by Nakashima and co-workers presents the interesting hypothesis that OMP buffers cAMP in the cilia, and that this underlies the physiological and behavioral phenotypes. The authors present data supporting binding of cAMP to OMP, complementary modeling and experimental data on responses of OSNs and a study of the behavioral phenotypes of the OMP KO mice. This is potentially an important contribution to the understanding of olfactory transduction, but unfortunately I find substantial deficits in the study that make it unclear whether the data and modeling support the hypothesis.

Major

1. The BRET study of the binding of cAMP to OMP is useful and potentially relevant to explain the phenotype of OSN responses in OMP Hets/KOs. Unfortunately, there is no information on the kinetics of binding, that would be useful to evaluate the validity of the conclusion that binding of cAMP to OMP is responsible for the differences in phenotype. Providing information on the on (and off) binding kinetics would be useful for modeling transduction.
2. Modeling the binding of cAMP to OMP could be extremely useful. However, the model shown in Fig. 2 may be incorrect. The binding of cAMP to OMP should be reversible. Is this correct? The authors do not present modeling in sufficient detail to evaluate its validity. Indeed, the modeling results are presented as cartoons without details and with no discussion in the results of how altering the different variables affects the CNG current. Does OMP buffering result in a smaller cAMP increase in WT mice?
3. The electrophysiological study in HEK293 cells shown in Fig. 2 is incomplete. The evidence for CNGA2 channel expression in Fig. 2g and the presentation of the whole cell patch clamp data in Fig. 2h,I are deficient. For Fig. 2g please show traces in the absence/presence of cAMP as well as histograms of channel openings. Was a control performed in HEK293 cells not expressing CNGA2? Did the authors perform an I-V curve analysis of channel opening? Similarly for Fig. 2h was this whole cell patch clamp, what was the holding voltage? Please show an I-V curve for the whole cell voltage clamp. Is there a difference in the peak current between OMP(+) and control? Why does the current decrease to zero in the presence of OMP?
4. The experiments in Fig. 3 do not support the title of the figure "OMP-KO neurons are silenced by cAMP overload". Did the authors test repeated stimulation by uncaging cAMP? How does the modeling in Figs. 3i,j support the conclusion of this figure? Modeling is not presented in sufficient detail to be evaluated by the reader.
5. In reference to Fig. 4 Reisert et al 2007 found that the response kinetics of OSNs were virtually indistinguishable between OMP^{-/-} and wild-type ORNs when intracellular cAMP level was elevated by the phosphodiesterase inhibitor, IBMX. On the basis of that observation they concluded that OMP acts upstream of cAMP production. This contradicts the conclusion of this manuscript. Fig. 4 does not show the response of the OSNs to IBMX. How did the OSNs respond to addition of IBMX? Have the authors modeled IBMX action? How (and why) does it differ from IBMX + mechanical stimulation? How do the kinetics of binding of cAMP to OMP explain this potential discrepancy with Reisert?
6. There is a problem with the presentation of the statistical analysis in this manuscript. The statistics should be explicitly stated in the figure legends (and/or the text). For example, a p value is stated in the legend for Fig. 1i. However, neither the test that was used nor the number of samples was stated. The statistical analysis is detailed in the supplementary data sheet, but the test used, p and F values and degrees of freedom should be stated in the results and/or the figure legends. This comment holds for all statistic tests presented in the manuscript.

Minor

1. Line 6 of page 8. Change: "The previous reports show that odorant stimulation to..." to

"Previous reports show that odorant stimulation of..."

2. References 30 and 31 in line 8 of page 8 are not relevant to this sentence. Please modify.
3. Line 7, page 10 change "overloaded..." to "increased".
4. Line 5, page 12, delete "stable"
5. The scale of the y axis in Figs. 1d and e are different. Did cAMP decrease delta BRET to background levels, or was there partial inhibition?

Reviewer #3:

Remarks to the Author:

Dear Editor,

In the manuscript entitled "Cyclic AMP buffering sharpens olfactory signal transduction and ensures repetitive olfactory neural responses during odour source searching", Nakashima and colleagues report that the olfactory marker protein (OMP) directly captures cAMP during odour stimulation within the ciliary cytosolic compartment of olfactory sensory neurons (OSNs). Moreover, the authors show data that support the notion that this process serves as a buffer for cAMP, which, during sustained activity, would otherwise drive OSNs into complete adaptation. Intriguingly, Nakashima and colleagues demonstrate that OMP-deficient mice show serious impairment in sniffing-based searches for an odour source. The authors, thus, conclude that cAMP buffering operates to maintain discrete olfactory responses.

The authors use in vitro BRET assays to assess direct interactions between OMP and cAMP. In heterologous cells, cAMP uncaging during simultaneous CNG channel measurements indicate that OMP is able to sequester a cytosolic cAMP surge. Finally, the authors provide electrophysiological and behavioral evidence that supports a role for OMP-mediated cAMP buffering in OSN adaptation. The findings presented by the authors are of interest to a wide range of individuals working in the field of (chemo)sensory neuroscience. The role of OMP in olfaction has remained a mystery for decades and only recently studies have identified physiological functions for this unique and neural population-specific protein. Nakashima et al. add an intriguing mechanism to the emerging picture. Their results could therefore be a substantial advance of the field. However, several concerns should be addressed before publication in Nature communications (see below).

Concerns:

My major concern is that the physiological recordings from OSNs (as shown in Figs. 3 and 4) are not easy to interpret and not sufficient (at least in the current stage of the manuscript) to support the authors' main claims. The following concerns need to be addressed:

a) Fig. 3a: Spontaneous firing frequency (at least in KO mice, but maybe also in Het animals) is not normally distributed, indicating that olfactory sensory neurons might represent physiologically different populations that should be analyzed independently. Fig. 3b is a prime example: basal firing rates and quality is already different, making it hard to assess the different 'response' patterns.

b) Fig. 3b: The time-course of the 'response' is somewhat unclear. Is the stimulus really applied ~30s before the peak response? It seems a bit arbitrary how to determine the actual peak then.

c) Fig. 3a, c & e: While spontaneous firing is highly heterogeneous (as shown in a), variability is almost not existent when experiments b and d are quantified (as shown in c and e). Something does not add up.

d) Fig. 3f-h: It is unclear why the authors now switch to mechanical stimulation. While it has been shown that OSNs also function as mechanosensors, this physiological role is much less understood and far from being a universal model for OSN activity. To not compare apples with pears, the

authors should rather opt for an odour stimulation paradigm. This is essential since the interpretation of their overall finding is currently hampered by the lack of any evidence in response to actual odour stimulation. While I am aware that this is technically challenging, it is definitely doable (e.g., using genetically targeted OSN populations with optically identifiable receptor expression, such as M71 or mOR-EG or other receptors). For me, this is the state-of-the-art in the field and corresponding experiments should be done to substantiate the authors results.

e) Fig. 3g: The original trace appears as if this might be a multi-unit recording. Have the authors applied spike-sorting? And, if so, which unit has been analyzed?

f) Figs. 3 & 4: the models shown in Fig. 3i,j as well as in Fig. 4f-h are not necessary and, as they are not results of modeling (or other approaches), they are also confusing. The reader tends to look for information in those schemes (e.g., looking at kinetics, etc.), while it is at this point mere speculation.

g) Histology / immunochemistry shown in Fig. 4a is hardly convincing. Figure quality is poor (maybe resulting from compression during file conversion?).

h) Fig. 4c-e: Again, the authors somewhat switch models. Now, wild-type OSNs are compared to HETs. This makes comparability between experiments almost impossible. A major concern is the 'chronic' IBMX treatment. At a concentration of 30 μ M, IBMX alone should have dramatic effects on OSNs. It is very hard to believe that prolonged IBMX incubation is without side effects.

Minor points:

1) main text; p. 4: "Adaptation adjusts basal states by reducing sensitivities to constant stimuli over time by utilizing Ca²⁺ influx during signal transduction cascades." This general statement is supported by rather context-specific references (1-4). Additionally, not all adaptation mechanisms function via Ca²⁺ influx.

2) p.4, ll. 11-13: "Meanwhile, odour-identification among background odours and odour-source searching requires the sufficient sensitivity during repetitive sniffing without excessive adaptation13-17."

The choice of references is somewhat misleading. How has this central claim of the authors (i.e., repetitive sniffing does not entail adaptation) been shown previously?

3) Fig. 1d: The DR curve should be presented with a sigmoidal fit.

4) p. 8; ll. 14-17: The authors should also cite Michalakis et al. (2006) Loss of CNGB1 protein leads to olfactory dysfunction and subciliary cyclic nucleotide-gated channel trapping. *J Biol Chem.* 281(46): 35156-66.

5) Fig. 2: the sequence of panels (a-k) is rather confusing. Following the logic of the figure is not easy.

6) Fig. 2g: How can the authors be sure that the measured activity is CNG single channel activity. Controls (pos. / neg.)?

7) Fig. 2h-k: Again, I am missing the relevant controls. Has this whole-cell patch-clamp experiment also been performed in cells where transfection with the CNG-encoding plasmid was omitted? Particularly for the models shown in j and k, it is unclear why the authors bring up Ca²⁺ at this stage. The CNG channel is an unselective cation channel and the current the authors measured will be of mixed nature.

8) Fig. 2: IBMX controls should be performed to strengthen data from physiological recordings.

9) The concentrations used for pharmacological activation / inhibition (IBMX; db-cAMP; SQ 22,536) are rather high. Proof-of-principle experiments should be performed with concentrations closer to the EC90 / IC90 values of the different agents.

10) p. 12; ll. 2-4: I do not understand why Fig. 4a,b is mentioned.

11) Supplementary Fig. 6: The quality of the recording is poor.

12) Discussion: the discussion is rather brief (which is not necessarily bad) and unfocused. Facts about phylogeny, brain-wide OMP expression, etc. are just stated, without any apparent logic. This section definitely needs some work.

13) Manuscript text: there are a number of typos and several other grammar / language issues that should be addressed in a revised manuscript.

14) Note: this reviewer lacks expertise in molecular simulation and docking. Therefore, I cannot assess the quality of those results presented in Fig. 1a and in supplementary Figs. 1 and 2.

Reply letter 1 from Noriyuki Nakashima et al., related to:

Cyclic AMP buffering sharpens olfactory signal transduction during chemo-mechanical stimulation to ensure repetitive neural responses in food odour-source searching (NCOMMS-19-01557A)

Reviewers' comments:

Reviewer #1 (Remarks to the Author):

OMP is a highly abundant protein in ORNs. While previous studies have demonstrated the defective desensitization of odor responses, as well as the development of ORNs, in the OMP mutant animals, the exact actions of OMP in olfaction have remained elusive. In this study, Nakashima et al demonstrated that OMP directly binds to cAMP, which facilitates a quick desensitization and sharpening of CNG activity. The loss of OMP led to excessive Ca²⁺ influx and feedback adaptation, particularly in the presence of PDE inhibitors, leading to the silencing of odor responses. At the behavioral level, the OMP mutant mice demonstrated defects in odor source searching. Together, these results show that OMP is critical for olfaction based on repetitive sniffing.

While OMP has been known for many years, this study provides the first clear-cut biochemical evidence for its role in olfaction. Desensitization is an important issue in olfaction as well as in sensory systems in general, and this study, to my knowledge, provides the first evidence that a cAMP-binding protein mediates the desensitization, by directly buffering cytosolic cAMP. As olfaction is coupled to sniff cycles, quick desensitization by OMP may be useful for repetitive odor detection. This is an important contribution toward our understanding of sensory adaptation.

Major comments:

1) It is quite striking that loss of OMP leads to an almost complete loss of prolonged responses. As this has never been observed in previous EOG or OB imaging studies, it is important to establish how this happens. Is this due to the feedback inhibition of CNG by CaM or depolarization-induced inactivation of sodium channels?

Reply:

Thank you for spending your time carefully reading our manuscript.

We appreciate you for pointing this out.

We speculate that the silencing of ORNs is due to the depolarization-induced inactivation of sodium channels (new Fig. 4).

Regarding the silencing, we initially had the same impression.

However, prolonged CNGA2 currents were observed in response to a single stimulus (new Fig. 3d-g), which is the same phenomenon previously reported by Dr. Reisert.

We also added the response of ORNs to a single stimulus, and OMP-KO neurons showed a prolonged response. Thus, we included only the data representing the prolonged response in our manuscript for comparison with the previous report (new Fig. 4g,h). This response occasionally resulted in decreased firing (adaptation). The repetitive stimuli finally led to the loss of prolonged (or resilient) firing responses but silencing. In this context, our observation is completely consistent with the previous report.

For slice recordings, we added experiments in which the repetitive stimulation induced adaptation in KO neurons, which was reversed by hyperpolarization of the membrane by external field stimulation (new Fig. 4l,m). With a rather strong stimulation by $V_p = +60$ mV, however, the firing rate returned to only the basal level and not to the evoked level, indicating that the continuous Ca^{2+} influx induced membrane depolarization, and CNG channels were also most likely inhibited by adaptation during the late phase of repetitive stimulation.

The shapes of action potentials, which were shrunken during the puff stimulation, also recovered. Therefore, the activation speed of voltage-gated Na channels was recovered by the reversal of inactivation induced by the membrane-hyperpolarizing test pulse (new Supplementary Fig. 5a-c).

We also exemplified the responses of OMP^(+/-) neurons and KO neurons to a single puff stimulation; OMP^(+/-) neurons resiliently and sharply responded, whereas KO neurons responded in a prolonged manner with an initial delay to a weak stimulus and gradual adaptation to a stronger stimulus, similar to the responses to repetitive stimuli.

In vivo nerve recordings under the repetitive stimulus condition, on the other hand,

showed the multiunit recording with reduced firing in KO mice (new Fig. 6), which was consistent with the direct recording from KO neurons.

Thus, the prolonged responses in the olfactory bulb should be a collective summation of multiple ORNs with prolonged firing responses.

We speculate that this dispersed summation could be due to the nonselective, retarded and diverse responses of each ORN. However, because we do not have evidence of nonselective responses in KO neurons, we did not include this speculation in our manuscript.

2) The behavioral experiments were performed with PDE inhibitors, but is it possible to see some effect without the inhibitor? Also, with regards to the proposed mechanism, it would be interesting to perform the same experiment with different odor concentrations. Odor concentrations should be indicated in figure legends.

Reply:

We apologize for the confusing figures.

All the behavioural assays were basically conducted without PDEi in control (pretreatment) experiments. We have explicitly indicated this information in the new Fig. 7.

The concentrations of odours in the “food odour” could not be specified because the food chunk was a mixture of various ingredients. We described the experimental protocol in detail in the figure legend, including the amount of food used (new Fig. 6).

Several difficulties were related to concentration, in addition to the unknown ingredients mentioned above.

In general, the motivation for olfactory searching in mice seems to decline during repeated sessions by acclimation. On the other hand, in our experiment, the mice were motivated by starvation and most likely began searching for hidden food within a few seconds of sniffing; thus, they may have become more eager to search for food during the learning session. Thus, it is difficult to parametrize olfactory sensitivity with regard to the amount of food.

We also considered reproducing the behavioural experiments using artificial odours, which turned out to be difficult to perform because the animals must learn to correlate

their search for objects with artificial odours. However, the detection sensitivities of OMP-KO mice actually depend on sniffing, and it is difficult to faithfully train OMP-KO mice to eat food scented with artificial “food odours” prior to the experiments.

We are certainly interested in the olfactory behaviours of OMP-KO mice from the viewpoint of odour concentrations.

We herein provide our preliminary data from another project investigating the artificial odour sensitivities of OMP-KO mice.

Both WT and KO mice showed acclimation to new odours by sequential presentation of limonene (+), and KO mice could not discriminate the structurally similar odour (limonene (-)). WT mice became interested in limonene (-) as a new odour at the 4th presentation. KO mice showed less sensitivity but approached the odour spot with a visual aid (scented swabs). However, the KO mice were not very accustomed to the odour at the 4th presentation of the swab. Furthermore, the KO mice were not highly interested in the new odour (limonene (-)).

Thus, when using artificial odours, the mice could utilize other senses, including visual aids.

Again, to eliminate such collateral information, we used food odours, which could instantly be linked to food after 8 weeks of being ordinarily stored with foods.

We will further investigate the impact of OMP by considering more general odourants in the future.

3) *The desensitization of ORNs has been extensively studied. PDEs, CaM, AC3, and CNG channels are examples. How is the OMP-dependent mechanism different from them mechanistically and physiologically? As the authors claim a new type of desensitization, they need to discuss more about this point, for example, in terms of temporal kinetics and target molecules.*

Reply:

Thank you for your comments.

We have added an experiment evaluating the binding kinetics of cAMP and OMP (new Fig. 3d-g, new Supplementary Fig. 3j) and a discussion of the other molecular adaptation target (Page 10, Line 1 ~ Page 11, Line 13)

Kinetic analysis revealed the approximate action phases of each cAMP-dependent molecule as follows: CNG activation, ~100 ms; OMP binding, ~1 s; and PDEi elimination, ~10 s or gradually at the basal level.

This observation further elucidates the action of OMP in eliminating cAMP in parallel with PDEi. With regards to kinetics, cAMP buffering is a short-term mechanism, which is also mentioned in the discussion (Page 21, Line 16 – Page 22 Line 10).

We also added experiments using PDEi, which only slightly slowed the CNG currents in the presence of OMP. These results indicate that the elevation of PDEi changed the basal cAMP-buffering capacity of OMP, but OMP operates independently from PDEi (new Fig. 3f-j).

Minor comments:

4) *Overall, more experimental details should be included in figure legends. For example, it is not fully described how mechanical stimuli were applied in Figure 2.*

Reply:

Thank you for pointing this out.

We have added details regarding the methods in the figure legends.

Additionally, we have incorporated the statistical values in the legends according to the style of the journal.

5) *It has been reported that mechanical responses in ORN are heterogeneous and OR-specific. For example, if nearly half of the ORNs do not respond to mechanical stimuli, how did you control for that? Were there any criteria to include or exclude a particular type of ORNs in the analysis?*

Reply:

Thank you for this constructive comment.

In the present manuscript, we focused on chemo-mechanical stimulation. Therefore, we used only the ORNs sensitive to mechanical (puff) stimulation. We added the rationale (Page 5, Lines 4-8) and an explanation of the selection criteria (Page 12, Lines 12-15).

In the comparison analyses, we tried to use only the ORNs with spontaneous firing rates of approximately 4 Hz (the mean), although we sometimes used ORNs with firing rates ranging from 2-8 Hz, which largely corresponded to the mean +/- s.d.

For mechanical stimulation, we used ORNs in the dorsal part of the olfactory epithelium, and the responsive ORNs were used for subsequent experiments. In our experiments, we did not detect mechanically-insensitive ORNs. Sometimes, ORNs showed phase-locked responses to stimuli over time. However, the mechanically induced firings were not always timed with the solution puffs, which might be due to differences in odourant receptor subtypes. As we have not elucidated the mechanisms of heterogeneity in responses towards mechanical stimulation, these data have been added in new Supplementary Fig. 5a-d.

In most cases, the mechanically induced responses corresponded to continuously increased or decreased firings; thus, we averaged the firing rates in discrete bins in the new Figs. 4 and 5, and relevant Supplementary Figs. 5 and 6.

We noticed that spontaneous firing itself was rather heterogenous (new Fig. 4a, new Supplementary Fig. 5a-c), and some ORNs showed spontaneous activity with very high or low frequency. In this context, we cannot describe the OR-specific regulation of mechanical responses.

Our study focused on the roles of OMP in regulating ORN cAMP-dependent processes induced specifically by chemo-mechanical stimulation. To clarify this purpose, we

added the rationale in our revised manuscript (Page 6, Line 3). The heterogeneity of ORN is mentioned in the discussion section (Page 23, Line 8- Page24, Line 3).

Considering the diverse spatiotemporal roles of cAMP, we cannot narrate all the possible outcomes in this manuscript, and we would like to investigate these outcomes individually in the future.

6) In Figure 3b, 3g, 4c, and 4g, the definition of peak, I, II, and III should be indicated. Just indicate time from the odor application in these panels (e.g., 0, 5, 15, 25, 35 sec).

Reply:

Thank you for your suggestion.

We have indicated the chemical and stimulus application times in the new Fig. 4b,d,g,h,j,l and Fig. 5c,d.

For Fig. 4b, we could not specify the time at which the chemical perfusion started because we used a peristaltic pump to maintain and switch solutions. Thus, the time that the stimulation solution reached the chamber did not accurately correspond to the actual start of the effect of the solution. The solution was replaced 20-30 seconds after entering the chamber, and the effect fluctuated until this occurred. For precise timing, we collected data on the UV uncaging of membrane-permeable cAMP, shown in the new Fig. 4d.

7) Figure 5d and 5e obviously should not follow a normal distribution. It is not appropriate to show mean +/- sd with these bar graphs.

Reply:

Thank you for your suggestion.

We deleted the SD error bars, and only the mean values are shown on the actual plot.

8) Supplementary Figure 6 can be moved to main figures. Traces in d and e are too dense. Supplementary Figure 1 may also be moved too.

Reply:

Thank you for this suggestion.

We have moved these supplementary figures to main figures as new Figs. 1 and 6.

The traces in the new Fig. 6 were replaced by other data at a more appropriate magnification, and the detailed experimental procedures are provided in the legend.

9) Typos and mistakes in English needs to be carefully corrected in the final version. For example in page 17, fundamentally should be fundamental. Also, “ecological importance of cAMP buffering” does not make sense.

Reply:

Thank you for spending your time in carefully reading through our manuscript. Our revised manuscript has been proofread by a professional English language editing service (American Journal Experts).

Reply letter 2 from Noriyuki Nakashima et al., related to:

Cyclic AMP buffering sharpens olfactory signal transduction during chemo-mechanical stimulation to ensure repetitive neural responses in food odour-source searching (NCOMMS-19-01557A)

Reviewers' comments:

Reviewer #2 (Remarks to the Author):

The absence of olfactory marker protein (OMP) in olfactory sensory neurons (OSNs) causes sustained responses to single odorant stimulation and decreased responses to repeated stimulation. The decreased response to repeated stimulation is thought to underlie behavioral olfactory deficits in OMP knockout (KO) and heterozygous (Het) mice. However, how OMP alters olfactory transduction is unknown. The article by Nakashima and co-workers presents the interesting hypothesis that OMP buffers cAMP in the cilia, and that this underlies the physiological and behavioral phenotypes. The authors present data supporting binding of cAMP to OMP, complementary modeling and experimental data on responses of OSNs and a study of the behavioral phenotypes of the OMP KO mice. This is potentially an important contribution to the understanding of olfactory transduction, but unfortunately I find substantial deficits in the study that make it unclear whether the data and modeling support the hypothesis.

Major

1. The BRET study of the binding of cAMP to OMP is useful and potentially relevant to explain the phenotype of OSN responses in OMP Hets/KOs. Unfortunately, there is no information on the kinetics of binding, that would be useful to evaluate the validity of the conclusion that binding of cAMP to OMP is responsible for the differences in phenotype. Providing information on the on (and off) binding kinetics would be useful for modeling transduction.

Reply:

Thank you for spending your time in carefully reading through our manuscript.

We agree that the estimated reaction rates should be useful.

We applied several approaches to estimate the magnitudes of reaction rates for the binding (on-) kinetics. The deactivation kinetics of CNG channels were useful for estimating the binding kinetics, which were as follows (new Fig. 3d-g: 1.67 s with OMP alone and 2.26 s with OMP and PDEi. From the BRET experiments, steady state was reached with a tau of 1.5 s (new Supplementary Fig. 3j), which was comparable to the results of the electrophysiological recordings.

Regarding your reference to “(and off)”, off-responses are difficult to measure. The competitive assay showed the partial inhibitory responses probably because OMP contains two binding sites. We speculate that the sequential binding of cAMP to these binding sites would exhibit hysteresis in a dissociation reaction. We added a discussion of these data to the revised manuscript (Page 22, Lines 5-18).

2. Modeling the binding of cAMP to OMP could be extremely useful. However, the model shown in Fig. 2 may be incorrect. The binding of cAMP to OMP should be reversible. Is this correct? The authors do not present modeling in sufficient detail to evaluate it's validity. Indeed, the modeling results are presented as cartoons without details and with no discussion in the results of how altering the different variables affects the CNG current. Does OMP buffering result in a smaller cAMP increase in WT mice?

Reply:

Thank you for the comments regarding the release of cAMP from OMP. Yes, the initial binding of cAMP to OMP is presumably reversible. As a distinct project, we employed reverse modelling to assess the tonic action of OMP, and trends in cAMP-sequestering effects were expected, as you suggested.

In the present experiment, we focused on the phasic reactions in the vicinity of the

membrane. OMP is estimated as a large drain of cAMP, as modelled in this study (new Fig. 3i). The refined model in the (a) absence or presence (b) of OMP will be studied in the subsequent project.

Regarding the cAMP surge (or the peak current in the next comment), the CNG currents in the presence of OMP were indeed smaller than those in the absence of OMP (shown below). Using repetitive flashing, we confirmed that the CNG currents were smaller and took longer to reach saturation in the presence of OMP than in the absence of OMP (shown below). However, this experiment was somewhat difficult to repeat because of the excessive cAMP uncaging and unstable recovery of photosensitivity even in the presence of a 6% ND filter. We present only a representative trace here.

3. The electrophysiological study in HEK293 cells shown in Fig. 2 is incomplete. The evidence for CNGA2 channel expression in Fig. 2g and the presentation of the whole cell patch clamp data in Fig. 2h,I are deficient. For Fig. 2g please show traces in the absence/presence of cAMP as well as histograms of channel openings. Was a control performed in HEK293 cells not expressing CNGA2? Did the authors perform an I-V curve analysis of channel opening? Similarly for Fig. 2h was this whole cell patch clamp, what was the holding voltage? Please show an I-V curve for the whole cell voltage clamp. Is there a difference in the peak current between OMP(+) and control? Why does the current decrease to zero in the presence of OMP?

Reply:

Thank you for your constructive comments.

We have added experiments confirming that the channels expressed in the transfected cells were in fact CNG channels. Using excised patch recordings, we altered the holding potentials (V_p) to show that single-channel activities exhibited an approximately 40-pS conductance, which was similar to that presented in a previous report (new Supplementary Fig. 4a-d).

We evaluated cell-attached recordings to show that these single channel activities were gated upon application of the adenylate cyclase activator forskolin, which was not observed in cells not transfected with cDNA (new Supplementary Fig. 4e,f).

We then evaluated whole-cell recordings to show that the whole-cell currents in the cDNA-transfected cells were increased upon application of the adenylate cyclase activator forskolin, which was not observed in cells not transfected with cDNA (new Supplementary Fig. 4g,h).

We confirmed that cells not containing CNG cDNA showed no flash cAMP-sensitive currents at $V_p = -80$ mV in the whole-cell configuration. We also confirmed that OMP alone did not produce the flash cAMP-sensitive currents at $V_p = -80$ mV in the whole-cell configuration. (new Supplementary Fig. 4i,j).

Regarding the return of CNG currents to zero, this process was delayed in the presence of PDEi. We discussed that OMP possessed a high basal cAMP-buffering capacity during the initial few seconds at rest, and this capacity deteriorated when the basal cAMP pool was elevated in the presence of PDEi (new Fig. 3f,g; Page 10, Lines 7-18).

We discussed the peak currents in the previous reply.

4. The experiments in Fig. 3 do not support the title of the figure “OMP-KO neurons are silenced by cAMP overload”. Did the authors test repeated stimulation by uncaging cAMP? How does the modeling in Figs. 3i,j support the conclusion of this figure? Modeling is not presented in sufficient detail to be evaluated by the reader.

Reply:

Thank you for pointing this out. We changed the title to OMP-KO neurons are silenced by repetitive stimuli (new Fig. 4).

Furthermore, we added the results of electrical field effects to hyperpolarize the

membrane to test whether the firing activity is recovered (new Fig. 4l).

Therefore, we modified our hypothesis that the repetitive receptor currents resulted in prolonged membrane depolarization and were silenced in the ORNs after cAMP overload. The previous schemes were omitted because we have modified our conclusions.

5. In reference to Fig. 4 Reisert et al 2007 found that the response kinetics of OSNs were virtually indistinguishable between OMP-/- and wild-type ORNs when intracellular cAMP level was elevated by the phosphodiesterase inhibitor, IBMX. On the basis of that observation they concluded that OMP acts upstream of cAMP production. This contradicts the conclusion of this manuscript. Fig. 4 does not show the response of the OSNs to IBMX. How did the OSNs respond to addition of IBMX? Have the authors modeled IBMX action? How (and why) does it differ from IBMX + mechanical stimulation? How do the kinetics of binding of cAMP to OMP explain this potential discrepancy with Reisert?

Reply:

We believe that our observation is consistent with the finding previously reported by Dr. Reisert.

Similar to the phenomenon observed by Dr. Reisert, OMP alone can increase the rate of deactivation kinetics. IBMX only slightly slowed the decay tau of CNG channels within a magnitude of 10^0 s: 1.67 s with OMP alone and 2.26 s with OMP and PDEi (Fig. 3d-g). Thus, these results point to the conclusion that OMP has independent effects from PDE.

Dr. Reisert's data suggest that OMP acts on the processes of odorant binding, OR activation, AC activation, cAMP production, cAMP diffusion and CNG activation and that OMP acts independently of PDE actions, but not necessarily in the upstream of AC production.

This phenomenon can be also explained by the cAMP buffering by OMP, as we showed that the produced cAMP was swiftly removed by OMP (new Fig. 3d).

On the other hand, we performed an experiment to measure the CNG currents in HEK cells treated with IBMX, revealing attenuated OMP action, suggesting that OMP was equilibrated by the basal cAMP pools and that the buffering capacity was diminished.

We added these experimental details to the results and discussion sections as well as to the new Fig. 3f-j (Page10, Line7- Page11, Line11).

Dr. Reisert's conclusion of an association between OMP and AC3 is certainly interesting, and we do not deny its potential existence.

6. There is a problem with the presentation of the statistical analysis in this manuscript. The statistics should be explicitly stated in the figure legends (and/or the text). For example, a p value is stated in the legend for Fig. 1i. However, neither the test that was used nor the number of samples was stated. The statistical analysis is detailed in the supplementary data sheet, but the test used, p and F values and degrees of freedom should be stated in the results and/or the figure legends. This comment holds for all statistic tests presented in the manuscript.

Reply:

Thank you for this constructive suggestion.

We have added all the relevant statistical information to the legends.

Minor

1. Line 6 of page 8. Change: "The previous reports show that odorant stimulation to..." to "Previous reports show that odorant stimulation of..."

Reply:

Thank you for this comment. We have modified this sentence accordingly, and the English language in our manuscript has also been edited by a professional editing service.

2. References 30 and 31 in line 8 of page 8 are not relevant to this sentence. Please modify.

Reply:

Thank you for this comment. We have modified the sentence accordingly.

3. Line 7, page 10 change "overloaded..." to "increased".

Reply:

Thank you for this comment. We have modified the sentence accordingly.

4. Line 5, page 12, delete “stable”

Reply:

Thank you for this comment. We have modified this sentence accordingly.

5. The scale of the y axis in Figs. 1d and e are different. Did cAMP decrease delta BRET to background levels, or was there partial inhibition?

Reply:

As you pointed out, the inhibition was partial. An explanation of this result has been added to the revised manuscript (Page 22, Lines 5-10).

We speculate that the two binding sites function cooperatively. In other experiments using ATP, the effect seems to be rather complex. Based on the current results, we can report only that the binding of the cAMP surge is damped by OMP.

Reply letter 3 from Noriyuki Nakashima et al., related to:

Cyclic AMP buffering sharpens olfactory signal transduction during chemo-mechanical stimulation to ensure repetitive neural responses in food odour-source searching (NCOMMS-19-01557A)

Reviewers' comments:

Reviewer #3 (Remarks to the Author):

Dear Editor,

In the manuscript entitled “Cyclic AMP buffering sharpens olfactory signal transduction and ensures repetitive olfactory neural responses during odour source searching”, Nakashima and colleagues report that the olfactory marker protein (OMP) directly captures cAMP during odour stimulation within the ciliary cytosolic compartment of olfactory sensory neurons (OSNs). Moreover, the authors show data that support the notion that this process serves as a buffer for cAMP, which, during sustained activity, would otherwise drive OSNs into complete adaptation. Intriguingly, Nakashima and colleagues demonstrate that OMP-deficient mice show serious impairment in sniffing-based searches for an odour source. The authors, thus, conclude that cAMP buffering operates to maintain discrete olfactory responses.

The authors use in vitro BRET assays to assess direct interactions between OMP and cAMP. In heterologous cells, cAMP uncaging during simultaneous CNG channel measurements indicate that OMP is able to sequester a cytosolic cAMP surge. Finally, the authors provide electrophysiological and behavioral evidence that supports a role for OMP-mediated cAMP buffering in OSN adaptation.

The findings presented by the authors are of interest to a wide range of individuals working in the field of (chemo)sensory neuroscience. The role of OMP in olfaction has remained a mystery for decades and only recently studies have identified physiological functions for this unique and neural population-specific protein. Nakashima et al. add an intriguing mechanism to the emerging picture. Their results could therefore be a substantial advance of the field. However, several concerns should be addressed before publication in Nature communications (see below).

Concerns:

My major concern is that the physiological recordings from OSNs (as shown in Figs. 3 and 4) are not easy to interpret and not sufficient (at least in the current stage of the manuscript) to support the authors' main claims. The following concerns need to be addressed:

a) Fig. 3a: Spontaneous firing frequency (at least in KO mice, but maybe also in Het animals) is not normally distributed, indicating that olfactory sensory neurons might represent physiologically different populations that should be analyzed independently. Fig. 3b is a prime example: basal firing rates and quality is already different, making it hard to assess the different 'response' patterns.

Reply:

Thank you for spending your time carefully reading our manuscript.

As you mentioned, the firing patterns of OMP-KO ORNs are more intermittent than those of WT ORNs. While we do not know the exact mechanisms underlying this discrepancy, tonic and burst responses were previously reported to be frequently observed in olfactory neurons. The difference in the basal activation patterns of odorant receptors seems to play a role in the background. We added this discussion with additional references (Page 23, Lines 8-18).

Added references

J. Reisert, J Gen Physiol. 2010 Nov; 136(5): 529–540.

A. Nakashima et al., Cell. 2013 Sep 12;154(6):1314-25.

T. Connelly et al, J Neurophysiol. 2013 Jul 1; 110(1): 55–62.

CR. Yu et al., Neuron 2004;42:553–566.

We have added the F-test results to show the difference in variance (new Fig. 4a).

To compare these ORNs with different spontaneous firing rates, in subsequent experiments, we chose ORNs with firing frequencies in the range of 2-8 Hz, particularly attempting to use ORNs at 4 Hz (the mean). We evaluated the average firing frequencies in bins several seconds wide rather than the actual firing patterns, including basal firing and evoked firing. Thus, the increase or decrease in the average firing in the defined bins was analysed as “response patterns” in this manuscript (Page 12, Lines 7-15).

We also added representative traces with changes in amplitude during the spontaneous firing (new Supplementary Fig. 5a-c). We performed experiments to ameliorate the silenced firing of KO neurons by employing field stimulation to hyperpolarize the membrane. The recovered firing was rather abrupt, implying the fluctuation in membrane potential (new Fig. 4l). Considering that ORNs have extraordinarily high input impedance (~ giga-ohm), the slight fluctuation of ion channels would lead to drastic changes in the membrane potential. OMP might also contribute to smoothing such variance in the membrane potentials, which needs further investigations in the future.

b) Fig. 3b: The time-course of the ‘response’ is somewhat unclear. Is the stimulus really applied ~30s before the peak response? It seems a bit arbitrary how to determine the actual peak then.

Reply:

Thank you for this suggestion.

We added the definition of the word “response” (Page 12, Lines 7-15).

In the legend for the new Fig. 4c, the word was changed to average firing rate as a more explanatory term.

As you suggested, the stimulation start time was not definable in the chamber perfusion system. In new Fig. 4a, the stimuli were defined as the start of the perfusion into the chamber. To avoid applying the mechanical stimuli by bolus, we used a peristaltic pump, which resulted in delays in the time (approximately 30 seconds) the solution took to reach the chamber and in the time that the maximum concentrations were reached in the chamber. The precise time the peak was reached is unknown, and the 5-bin yielded the maximal averaged frequency, which was then defined as the peak.

To avoid this delay in stimulation time, we adopted flash UV uncaging, as shown in new Fig. 4d.

c) Fig. 3a, c & e: While spontaneous firing is highly heterogeneous (as shown in a), variability is almost not existent when experiments b and d are quantified (as shown in c and e). Something does not add up.

Reply:

We appreciate your careful reading of our manuscript and figures.

In the previous Fig. 3c and e, the error bars indicated SEM. We have replaced the SEM error bars in the figures with SD error bars.

d) Fig. 3f–h: It is unclear why the authors now switch to mechanical stimulation. While it has been shown that OSNs also function as mechanosensors, this physiological role is much less understood and far from being a universal model for OSN activity. To not compare apples with pears, the authors should rather opt for an odour stimulation paradigm. This is essential since the interpretation of their overall finding is currently hampered by the lack of any evidence in response to actual odour stimulation. While I am aware that this is technically challenging, it is definitely doable (e.g., using genetically targeted OSN populations with optically identifiable receptor expression, such as M71 or mOR-EG or other receptors). For me, this is the state-of-the-art in the field and corresponding experiments should be done to substantiate the authors results.

Reply:

Thank you for your important suggestion. We agree that confirming the physiological stimuli by odorant puffs is fundamentally essential.

As you indicated, it remains very difficult to target ORNs that selectively respond to specific odours (ligands) in our experiments. Thus, we applied in vivo multiunit recordings from ORNs in the dorsal region of the olfactory epithelium using natural odorant (food odours).

In this manuscript, we investigated sniffing-related olfaction, that is, the chemo-mechanical sensitivities of ORNs using only mechanically sensitive ORNs. Thus, we focused on the mechanical-related phenomenon throughout the study, and the title and abstract have been modified accordingly. We also added an explanation of the mechano-sensitivities of odourant receptors (ORs) in the introduction (Page 5, Lines 4-8).

While mechanical sensitivities might be not universal to ORNs, a substantial number of studies have reported the functions of mechanical sensitivity that we have referenced. We agree that we would need OR-specific experiments to completely generalize our

finding to all ORNs with different ORs.

We have added a related discussion to the current manuscript (Page 23, Lines 8-18).

We obtained additional approval from the Ethical Committee of Kurume University for related experiments using these animals. We consulted the Riken Bioresource Center about the M71-GFP strain mice, whose sperm and embryos are cryopreserved. We should receive these mice in six months, and we will then cross these mice to yield mice on an OMP-KO background. As this process will require an extensive amount of time, we plan to perform this experiment in future studies. Thank you for your constructive suggestion.

e) Fig. 3g: The original trace appears as if this might be a multi-unit recording. Have the authors applied spike-sorting? And, if so, which unit has been analyzed?

Reply:

As you mentioned, the KO-ORNs tended to present different “patterns” of spontaneous firing, namely, the burst type. Thus, this unit recording corresponds to doublet-type firing. We have added representative traces with changes in amplitude during spontaneous firing (Supplementary Fig. 5a-c).

The firing amplitude decreased depending on the firing frequency, and the action potential shapes changed, becoming smaller as Nav inactivation progressed with depolarization of the membrane potentials.

Our newly added experiments showed that the inactivation of sodium channels was directly responsible for the silencing of OMP-KO neurons (new Fig. 4I), and firing was recovered by employing field stimulation to hyperpolarize the membrane. As shown in this figure, the firing became abrupt, and the amplitudes fluctuated (similar to a doublet), suggesting unstable membrane potentials.

Because ORNs are surrounded by nonfiring supporting cells, multiunit recordings are rarely observed with a patch pipette with a resistance of approximately 5 mOhm. Of course, multiunit recordings occasionally happen, most likely due to the suctioning of one or two axons or somas of adjacent ORNs. The amplitudes are highly different in such cases and can easily be discriminated from other recordings by thresholding.

The absolute values of negative peaks were automatically measured. The detection criteria were set as follows: peaks greater than the noise level and a separation valley of less than 50% of that of the adjacent peaks. Thus, the doublet firings were all collected, but the small field potentials (below 50% of the preceding firing) of adjacent neurons were mostly eliminated from the analysed data.

We added this text to the method section (Page 59, Lines 13-18)

Moreover, a multiunit recording in the olfactory epithelium is rather difficult to obtain. We previously succeeded in effectively obtaining a loose multiunit recording using a pipette with a 20- μ m diameter (JP, 2013).

Again, in this study, we used only single-unit recordings regardless of burst or tonic firing because OMP-KO ORNs seem to have differences in spontaneous firing patterns by nature. This phenomenon could be investigated in future studies.

f) Figs. 3 & 4: the models shown in Fig. 3i,j as well as in Fig. 4f-h are not necessary and, as they are not results of modeling (or other approaches), they are also confusing. The reader tends to look for information in those schemes (e.g., looking at kinetics, etc.), while it is at this point mere speculation.

Reply:

Thank you for your suggestion.

These speculative schematics have been removed.

g) Histology / immunochemistry shown in Fig. 4a is hardly convincing. Figure quality is poor (maybe resulting from compression during file conversion?).

Reply:

Thank you for pointing this out.

We have replaced these images with larger, clearer images.

h) Fig. 4c-e: Again, the authors somewhat switch models. Now, wild-type OSNs are compared to HETs. This makes comparability between experiments almost impossible. A major concern is the 'chronic' IBMX treatment. At a concentration of 30 μ M, IBMX alone should have dramatic effects on OSNs. It is very hard to believe that prolonged

IBMX incubation is without side effects.

Reply:

We apologize for this confusion.

In the new Fig. 4, we examined the qualitative role of OMP. We thus compared OMP^(+/GFP)-Het neurons with OMP^(GFP/GFP)-KO neurons because Het- and KO-ORNs could be easily identified by GFP fluorescence.

In the new Fig. 5, we examined the quantitative role of OMP.

We did not compare KO neurons because their firings ceased under our experimental paradigm, namely, under the chemical stimuli used to produce cAMP overload; these neurons were not suitable for comparisons (new Fig. 3b,d, Supplementary Fig. 6a,b), and we could not compare the increase or decrease in firing by using OMP-KO ORNs. Thus, we compared OMP-WT neurons and OMP-Het neurons to investigate haploinsufficiency (Page 16, Line 18). Therefore, we believe that we did not switch models but rather expanded upon the viewpoint of the OMP expression level.

According to your comments, we alternatively used a different stimulus, PDEi (rolipram), and we obtained similar results for WT and Het neurons, whereas KO neurons were finally silenced at rest (new Supplementary Fig. 6). Given this context, we consider that the model was not switched but rather expanded.

We confirmed a similar phenomenon using rolipram, which elevated spontaneous firing in a previous study (see below; the mean frequency was normalized to 1), and we observed no obvious effects on firing other than on frequency during a 10-min incubation period.

We agree that in the presence of PDEi, ORNs undergo dramatic events via the activation of cAMP- and PKA-dependent mechanisms, including CNG channels and other modules that generate firing. However, these mechanisms were not the focus of this study.

Here, we used the same IBMX concentration used by Dr. Stephen Frings (below), who observed that 100 μM IBMX silenced wild-type ORNs after a few minutes of application. We observed little silencing at 30 μM IBMX or 10 μM rolipram in wild-type ORNs or Het-ORNs, but the Het-ORNs showed decreased firing in response to puff stimulation, as we show in the new Fig. 5c-e and Supplementary Fig. 6c-f.

Current Recording from Sensory Cilia of Olfactory Receptor Cells In Situ I. The Neuronal Response to Cyclic Nucleotides. STEPHAN FRINGS and BERND LINDEMANN. J Gen Physiol. 1991 Jan 1; 97(1): 1-16.

Minor points:

1) main text; p. 4: "Adaptation adjusts basal states by reducing sensitivities to constant stimuli over time by utilizing Ca^{2+} influx during signal transduction cascades." This general statement is supported by rather context-specific references (1-4). Additionally, not all adaptation mechanisms function via Ca^{2+} influx.

Reply:

We agree with the reviewer's comment and have omitted the expression "by utilizing Ca²⁺ influx".

Regarding the adaptation mechanisms of ORNs, we added new references related to other mechanisms observed in mammalian ORNs.

Kurahashi T, Menini A. Mechanism of odorant adaptation in the olfactory receptor cell. *Nature* 1997;385:725–729.

Bradley, J., Bonigk, W., Yau, K.-W. & Frings, S. Calmodulin permanently associates with rat olfactory CNG channels under native conditions. *Nat. Neurosci.* **7**, 705–710 (2004).

Wei, J. *et al.* Phosphorylation and inhibition of olfactory adenylyl cyclase by CaM kinase II in Neurons: a mechanism for attenuation of olfactory signals. *Neuron* **21**, 495–504 (1998).

Li, R.-C., Ben-Chaim, Y., Yau, K.-W. & Lin, C.-C. Cyclic-nucleotide-gated cation current and Ca²⁺-activated Cl current elicited by odorant in vertebrate olfactory receptor neurons. *Proc. Natl. Acad. Sci. U. S. A.* **113**, 11078–11087 (2016).

Zufall, F. & Leinders-Zufall, T. Role of cyclic GMP in olfactory transduction and adaptation. *Ann. N. Y. Acad. Sci.* **855**, 199–204 (1998).

Lecoq, J., Tiret, P. & Charpak, S. Peripheral adaptation codes for high odor concentration in glomeruli. *J. Neurosci.* **29**, 3067–3072 (2009).

Castillo, K., Delgado, R. & Bacigalupo, J. Plasma membrane Ca(2+)-ATPase in the cilia of olfactory receptor neurons: possible role in Ca(2+) clearance. *Eur. J. Neurosci.* **26**, 2524–2531 (2007).

2) p.4, ll. 11–13: "*Meanwhile, odour-identification among background odours and odour-source searching requires the sufficient sensitivity during repetitive sniffing without excessive adaptation*13-17."

The choice of references is somewhat misleading. How has this central claim of the authors (i.e., repetitive sniffing does not entail adaptation) been shown previously?

Reply:

Thank you for the constructive comments.

We modified this paragraph and included a new reference of a study in which Dr.

Verhagen et al. investigated the adaptive response of ORNs during sniff stimulation.

Verhagen, J.V., Wesson, D.W., Netoff, T.I., White, J.A., and Wachowiak, M. (2007). Sniffing controls an adaptive filter of sensory input to the olfactory bulb. *Nat. Neurosci.* 10, 631–639.

3) *Fig. 1d: The DR curve should be presented with a sigmoidal fit.*

Reply:

Thank you for carefully reading our study.

We have replaced the curve with a sigmoidal fit.

4) *p. 8; ll. 14-17: The authors should also cite Michalakis et al. (2006) Loss of CNGB1 protein leads to olfactory dysfunction and subciliary cyclic nucleotide-gated channel trapping. J Biol Chem. 281(46): 35156-66.*

Reply:

Thank you for your suggestion.

We have added this reference.

5) *Fig. 2: the sequence of panels (a–k) is rather confusing. Following the logic of the figure is not easy.*

Reply:

We have modified the sequences of panels with additional experiments.

We performed several experiments involving IBMX to confirm the principal contribution of OMP rather than PDE. With these new results, we have added a discussion of the kinetics of OMP, and the modelling has been moved to the last part of the new Fig. 3 (Page 10, Line 7- Page 11, Line 11).

6) *Fig. 2g: How can the authors be sure that the measured activity is CNG single channel activity. Controls (pos. / neg.)?*

Reply:

We performed experiments with positive and negative controls to show that only HEK293T cells transfected with CNG channel cDNA exhibited cAMP-sensitive

channel activities (new Supplementary Fig. 4).

7) Fig. 2h-k: Again, I am missing the relevant controls. Has this whole-cell patch-clamp experiment also been performed in cells where transfection with the CNG-encoding plasmid was omitted? Particularly for the models shown in j and k, it is unclear why the authors bring up Ca²⁺ at this stage. The CNG channel is an unselective cation channel and the current the authors measured will be of mixed nature.

Reply:

We have performed experiments with positive and negative controls, which are shown in the new Supplementary Fig. 4.

We performed the control experiments using HEK cells transfected with and without OMP but in the absence of the CNG-encoding plasmid. In both cells, no currents were induced by UV flashing.

As you suggested, CNG channel currents contain Na, Ca, and other cations. Thus, we modified our conclusion to state that the membrane depolarization most likely induced voltage-gated Na channel inactivation (new Fig. 4l and m).

8) Fig. 2: IBMX controls should be performed to strengthen data from physiological recordings.

Reply:

Thank you for your important suggestion.

We recorded the effect of IBMX on CNG currents with or without OMP (new Fig. 3f,g). This experiment indeed strengthened the data and yielded results that contributed to our discussion of the kinetics and the cAMP-buffering capacity of OMP at different basal cAMP pool sizes (Page 10, Line 7 - Page 11, Line 11).

9) The concentrations used for pharmacological activation / inhibition (IBMX; db-cAMP; SQ 22,536) are rather high. Proof-of-principle experiments should be performed with concentrations closer to the EC₉₀ / IC₉₀ values of the different agents.

Reply:

Thank you for your suggestion.

For db-cAMP, we have already experimentally examined a different agent. In the initial experiment, we used db-cAMP to emphasize the contribution of cAMP. While this was an introductory experiment, temporal precision of the stimulation was not achieved. We next used the photosensitive membrane-permeable 8-Br-cAMP analogue with UV uncaging to achieve temporal precision (new Fig. 4d).

Regarding other agents, we have added experiments performed using different chemicals as follows: rolipram as a PDE inhibitor and BPIPP as an AC inhibitor (new Supplementary Fig. 5 and 6).

10) p. 12; ll. 2-4: I do not understand why Fig. 4a,b is mentioned.

Reply:

We apologize for this confusion.

In the new Fig. 4 (old Fig. 3), we evaluated the firing qualitatively in the presence or absence of OMP.

In the new Fig. 5 (old Fig. 4), we evaluated the firing depending on the quantity of OMP expressed.

We added this rationale for the investigation of the quantitative role of OMP in the paragraph related to the new Fig. 5a,b (Page 15, Line 7 - Page 16, Line 4). The conclusion for this observation was indicated as haploinsufficiency in Page 16, Line 18.

11) Supplementary Fig. 6: The quality of the recording is poor.

Reply:

Thank you for pointing this out.

We have added new traces with enlarged images. These figures were moved to the main figures (new Fig. 6) to link the cellular-level study with the behavioural experiments.

12) Discussion: the discussion is rather brief (which is not necessarily bad) and unfocused. Facts about phylogeny, brain-wide OMP expression, etc. are just stated, without any apparent logic. This section definitely needs some work.

Reply:

We added a discussion of the main facts about adaptation mechanisms.

We also added the limitations of our study regarding the chemo-mechanical stimuli, hysteresis in the association/dissociation between cAMP/OMP, long-term effects and odourant receptor specificity (Page 21, Line 1 - Page 24, Line 3).

Because other facts about the phylogeny, etc., provide clues for researchers in other fields, this section was retained and only minimally modified, although it was placed at the end of the manuscript (Page 24, Lines 4-14).

13) Manuscript text: there are a number of typos and several other grammar / language issues that should be addressed in a revised manuscript.

Reply:

Thank you for your careful and detailed reading of our manuscript.

Our revised manuscript has been proofread by a professional English language editing service (American Journal Experts).

14) Note: this reviewer lacks expertise in molecular simulation and docking. Therefore, I cannot assess the quality of those results presented in Fig. 1a and in supplementary Figs. 1 and 2.

Reply:

I appreciate your fair comments.

Reviewers' Comments:

Reviewer #1:

Remarks to the Author:

This study demonstrates an important piece of adaptation mechanisms in mammalian ORNs. Authors have addressed some of important concerns raised in the previous round of review. Particularly, the newly added Fig. 4l and m are very important in establishing that the depolarization-induced inactivation of APs (most likely by the inactivation of voltage-gated sodium channels) is a major cause of the sensory-evoked ORN silencing in the OMP KO animal. I continue to strongly support this manuscript; however, there are still remaining issues that need to be addressed before publication.

Major

1) As I wrote in my previous comment #2, I still believe that the behavioral analysis and deficits found under PDEi are artificial. It would be nice if authors could mimic the situation in Fig. 4 in their behavioral assay using physiological sensory stimuli. For example, running mice are known to show higher mechanosensory responses (Kato et al., *Neuron* 2013; Iwata et al., *Neuron* 2017). It may also be possible to perform behavioral analysis under higher background odors (e.g., Rokni et al., *Nat Neuro* 2014). Regarding concentration dependence, it is not particularly difficult to associate an odor with a sugar reward, and test variable odor concentrations. If authors want to test the impact of OMP KO in "repetitive" sniffing, odor trail tracking assay may be useful (Khan et al., *Nat Comm* 2012).

2) I echo Reviewer #2's comment d. The authors need to test both mechanical and odor-evoked responses, even though both are mediated by cAMP pathway. I understand that it would be time consuming to use OR-GFP mice; however, authors could easily record an EOG. Alternatively, the authors could add odor response data in Fig. 6.

Minor

3) Adaptation and depolarization-induced silencing are mechanistically different. The authors show that OMP is important for adaptation by buffering cAMP. Failure in this adaptation process results in the silencing of ORNs, but this is not an adaptation mechanism. They should be carefully distinguished throughout the manuscript (e.g., Fig. 4, 6, page 15 line 12, page 17 line 18).

4) Related to my previous comment #7, do not use ANOVA for non-normally distributed data. You need to use non-parametric tests in this case. In Fig. 4a, the statistical analysis is not described in figure legends.

5) The title does not convey the major conclusion of this study. Shorter is better. Also, as mentioned by Reviewer #2 comment #4, it is not really demonstrated whether OMP is important for the responses to "repeated" stimuli, particularly in the physiological context. To argue the specific role in "repeated" sensation, authors would have to compare single sniff vs multiple sniff tasks (not essential in this paper, though). It is also unclear whether OMP is specifically important for food odor-source searching. The major conclusions of this study is that a) OMP buffers cAMP by direct binding and b) avoid sustained depolarization of ORNs upon sensory stimuli. The true value of this study is the discovery of a novel type of desensitization in sensory signal transduction.

6) The term "chemo-mechanical stimulation" is weird and inaccurate. I would simply use "sensory stimulation".

7) Page 4, line 15. CACC is important to "amplify" the sensory signals, not for adaptation. PDE is also important for adaptation. Also correct page 22, line 1.

8) Page 5, line 5. For those who are not familiar with the mechanosensation in ORNs, it would be more helpful to explain that mechanosensation is mediated by OR, G-protein, AC3, and is important for the sniff-coupled temporal coding.

9) Page 5, line 18; page 10, line 11; and page 11, line 1. It is true that PDE was not so effective in HEK293 cells. However, ORNs express PDEs at a much higher level. The authors cannot comment on the relative contribution of OMP vs PDE based on these experiments. Indeed, the authors did see a difference under PDEi in Fig. 7. Similarly in page 10, line 10, this is not accurate.

10) Overall, the writing needs to be significantly improved. There are many inaccurate descriptions

(below).

- 11) Page 15, line 16. This experiment does not demonstrate the sufficiency.
- 12) Fig. 4d. It would be more appropriate to label it "cAMP uncaging" rather than "caged cAMP".
- 13) Page 19, line 4. It is unclear whether OMP-deficient mice are anosmic. Hyposmia would be better.
- 14) Page 20, line 3. It would be extremely difficult to judge whether mice "hesitated".
- 15) Page 20, line 12. The authors cannot tell whether this is based on the trained memory.
- 16) Page 20, line 17. This is not accurate. Basal sensitivity should increase in theory.
- 17) Page 21, line 13-15. This is not really demonstrated. Facts and assumptions should be distinguished.
- 18) Page 24, line 5-8. There is a leap of logic.

Reviewer #2:

Remarks to the Author:

This manuscript presents a novel finding on the role of OMP as a buffer of cAMP. The authors have answered my questions. This is an interesting contribution to the understanding of olfactory transduction.

Reviewer #3:

Remarks to the Author:

Dear Editor,

In their revised manuscript entitled "Cyclic AMP buffering sharpens olfactory signal transduction during chemo-mechanical stimulation to ensure repetitive neural responses in food odour-source searching", Nakashima and colleagues added new experimental data and made a number of changes to the text and figures to meet the concerns raised during initial review. While many aspects of the manuscript have improved, a few major points remain to be addressed.

In new Fig. 4I, a hyperpolarization experiment is introduced, in part, to address my previous concern (a). I do not exactly follow how both are related. In fact, as the authors state ("The recovered firing was rather abrupt, implying the fluctuation in membrane potential (new Fig. 4I). Considering that ORNs have extraordinarily high input impedance (~ giga-ohm), the slight fluctuation of ion channels would lead to drastic changes in the membrane potential. OMP might also contribute to smoothening such variance in the membrane potentials, which needs further investigations in the future.") these new findings rather add a novel layer of complexity rather than clarifying my point.

The *in vivo* olfactory nerve recordings (Fig. 6) describe a technique that the authors apparently established in this manuscript (I do not find references to previously published studies). While it could be a very powerful technique, I feel that reliability and data quality need to be established before drawing any conclusions. The reader needs to be aware of the S/N and the general suitability of this method. Positive and negative controls (e.g., by performing standard adaptation experiments according to conventional experimental paradigms) need to be performed and the results have to be quantified to allow interpretation.

I am also still somewhat hesitant to support the notion of "chemo-mechanical stimulation"

As I explained before, the authors still appear to "compare apples with pears" (previous concern (d)). I can understand that authors point out that "as this process will require an extensive amount of time, we plan to perform this experiment in future studies." However, this issue is central to the authors' claims and either needs to be addressed (could also be done by EOG or whole-cell patch-clamp recordings from isolated neurons) or the authors should limit their report

to the mechanical stimulation part.

Regarding my previous point (e), the authors argue that "Because ORNs are surrounded by nonfiring supporting cells, multiunit recordings are rarely observed with a patch pipette with a resistance of approximately 5 mOhm. Of course, multiunit recordings occasionally happen, most likely due to the suctioning of one or two axons or somas of adjacent ORNs. The amplitudes are highly different in such cases and can easily be discriminated from other recordings by thresholding." I do disagree. In the slice preparation, it is relatively easy to record simultaneously from several OSNs, since the soma layer is targeted. Why are the authors hesitant to perform waveform-based spike sorting?

Reply to Reviewer #1

Reviewers' comments:

Reviewer #1 (Remarks to the Author):

This study demonstrates an important piece of adaptation mechanisms in mammalian ORNs. Authors have addressed some of important concerns raised in the previous round of review. Particularly, the newly added Fig. 4l and m are very important in establishing that the depolarization-induced inactivation of APs (most likely by the inactivation of voltage-gated sodium channels) is a major cause of the sensory-evoked ORN silencing in the OMP KO animal. I continue to strongly support this manuscript; however, there are still remaining issues that need to be addressed before publication.

Major

1) As I wrote in my previous comment #2, I still believe that the behavioral analysis and deficits found under PDEi are artificial. It would be nice if authors could mimic the situation in Fig. 4 in their behavioral assay using physiological sensory stimuli. For example, running mice are known to show higher mechanosensory responses (Kato et al., Neuron 2013; Iwata et al., Neuron 2017). It may also be possible to perform behavioral analysis under higher background odors (e.g., Rokni et al., Nat Neuro 2014). Regarding concentration dependence, it is not particularly difficult to associate an odor with a sugar reward, and test variable odor concentrations. If authors want to test the impact of OMP KO in “repetitive” sniffing, odor trail tracking assay may be useful (Khan et al., Nat Comm 2012).

Reply:

Thank you very much for your suggestion to include the detailed behavioural experiments.

Inspired by the sucrose-reward association, we trained water-deprived mice to associate limonene odour with sucrose water (new Fig. 7a-d in the revised manuscript).

As you mentioned, background odour stimulation deteriorated the behaviour of KO mice in a manner different from that induced by cAMP loading using the PDE inhibitor; the KO mice continued to erroneously search for targets (Fig. 7e-j).

In the presence of a background odour, the Het mice also made more errors in attempting to drink from the empty bottle (Fig. 7f,h,i,j).

We thereby included these odour-evoked experiments in Fig. 7, and the hole-digging experiments were included in the new Fig. 8 as experimental cAMP-overloaded experiments.

These results and methods were added to the revised manuscript (Page 18, Line 10 - Page 19, Line 12, Results; and Pages 72 – 74; Method)

2) I echo Reviewer #2's comment d. The authors need to test both mechanical and odor-evoked responses, even though both are mediated by cAMP pathway. I understand that it would be time consuming to use OR-GFP mice; however, authors could easily record an EOG. Alternatively, the authors could add odor response data in Fig. 6.

Reply:

Thank you for your comment.

To first comment on the experiments suggested by reviewer 3, these suggested experiments could help to associate types of ORNs with specific odour responses. However, we have hesitation conducting the suggested experiments because (1) we currently do not have mice of which OMP^(-/-)-ORNs are labelled with GFP in an OR-specific manner to allow these cells to be identified for whole-cell recording in slices and (2) the *in vivo* nerve recording serves as an alternative to recording EOGs from the main olfactory epithelium. We believe the finding and conclusion of the present paper is the role of OMP in the firing activity of ORNs, on which we believe we have presented sufficient evidence.

In relation to the newly added behavioural experiments explained above (new Fig. 7), we have modified the entire *in vivo* recording section by adding the limonene odour-induced responses together with food odour mixture and mechanical puff stimulation in Fig. 6b,h (Page 17, Line 15 – Page 18, Line 9, Results).

To strengthen the data suitability, we added the quantification of the multiunit recordings by defining S/N, setting controls (puff with limonene odour or food odours,

puff without odours) and examining the stability of recording as negative controls (no stimuli) (Page 17, Line 15 – Page 18, Line 5, Results; and Page 78, Line 5- Page 79, Line 6, Method).

In these recordings, we succeeded in detecting the increase in the collective spike number in some nerve fibres in WT and Het mice and the decrease in the collective spike number in KO mice (Fig. 6f-h; and Page 18, Line 5-9, Results).

Minor

3) *Adaptation and depolarization-induced silencing are mechanistically different. The authors show that OMP is important for adaptation by buffering cAMP. Failure in this adaptation process results in the silencing of ORNs, but this is not an adaptation mechanism. They should be carefully distinguished throughout the manuscript (e.g., Fig. 4, 6, page 15 line 12, page 17 line 18).*

Reply:

Thank you for your understanding of our findings.

As you also pointed out in your comment #5, as *a novel type of desensitization*, we have discriminated the actions of OMP as cAMP-buffering, which transiently desensitizes CNG channels by uncoupling signal transduction and prevents the sustained depolarization-induced silencing of ORNs.

We have modified the core results in Fig. 4 as follows.

“These results indicate that cAMP buffering by OMP prevents the silencing of ORNs due to sustained membrane depolarization, which is a consequence of Na⁺ and Ca²⁺ overload through the open CNG channels.” (Page 15, Lines 13-15).

We have also made the following changes:

We have changed “adaptation” to “silenced” (Fig. 4l).

We have changed “adaptation” to “decrease” (Fig. 6e).

We have deleted “decreased” in Fig. 4h and in the related text (Page 13, Line 16 – Page 14, Line 10).

In the *in vivo* nerve recording section, we similarly analysed the total increase or

decrease in collective firing activity in the multiunit recordings. Thus, we have removed “adaptive decrease” and described the responses as only decreased or increased firing activity (Page 17, Line 15 - Page 18, Line 9).

4) *Related to my previous comment #7, do not use ANOVA for non-normally distributed data. You need to use non-parametric tests in this case. In Fig. 4a, the statistical analysis is not described in figure legends.*

Reply:

Thank you for your correction.

We used the Kruskal-Wallis test for non-normally distributed data, including those in the new Fig. 8 (previously Fig. 7) (Page 45, Line 22 – Page 46, Line 3; and Page 46 Lines 11-12).

We also used the Kruskal-Wallis test for non-normally distributed data and the Mann-Whitney U test to compare the pre-post effects of ambient limonene in the newly added Fig. 7 (Page 42, Line 16 – Page 43, Line 14; Fig. 7 Statistics).

We also added the statistical analysis to Fig. 4a. (Page 35, Line 2-3).

5) *The title does not convey the major conclusion of this study. Shorter is better. Also, as mentioned by Reviewer #2 comment #4, it is not really demonstrated whether OMP is important for the responses to “repeated” stimuli, particularly in the physiological context. To argue the specific role in “repeated” sensation, authors would have to compare single sniff vs multiple sniff tasks (not essential in this paper, though). It is also unclear whether OMP is specifically important for food odor-source searching. The major conclusions of this study is that a) OMP buffers cAMP by direct binding and b) avoid sustained depolarization of ORNs upon sensory stimuli. The true value of this study is the discovery of a novel type of desensitization in sensory signal transduction.*

Reply:

Thank you very much for understanding the essence of our study.

We have modified the title, abstract, conclusion and related text of the manuscript to focus on the cAMP buffering activity of OMP (Page 1; Title; Page 3, Abstract; Page 5, Lines 14-16; Page 15, Lines 13-15).

We have changed the Title as follows.

New title

Olfactory marker protein buffers cAMP by direct binding to avoid depolarization-induced silencing of olfactory receptor neurons

Accordingly, we have changed the Abstract and the related conclusion as follows.

New abstract:

Sensory cells respond to external stimuli with resilient firing properties. Olfactory receptor neurons (ORNs) use odour-induced intracellular cAMP surge to gate cyclic nucleotide-gated nonselective cation (CNG) channels in the limited ciliary space. The prolonged exposure to cAMP causes adaptation of CNG channels and attenuates neural responses. On the other hand, the odour-source searching behaviour of animals requires ORNs to be sensitive to the odour when approaching the targets. How ORNs accommodate these conflicting aspects of cAMP responses remains unknown. Here, we show that the cAMP-buffering machinery swiftly bypasses surplus cAMP during signal transduction to maintain the sensitivity of ORNs. We discovered that the cytosolic olfactory marker protein (OMP) directly captured cAMP, which transiently desensitized CNG channel activity and prevented sustained membrane depolarization upon the application of sensory stimuli. Under repetitive stimulation, *OMP*^{-/-} ORNs were immediately silenced after burst firing due to sustained depolarization and inactivated firing machinery. Consequently, *OMP*^{-/-} mice showed serious impairment during odour-source searching tasks. Therefore, cAMP buffering by OMP transiently desensitizes CNG channels by uncoupling the signal transduction but maintains resilient sensory responses in ORNs.

6) *The term “chemo-mechanical stimulation” is weird and inaccurate. I would simply use “sensory stimulation”.*

Reply:

Thank you for your suggestion.

We have replaced the related term with “sensory” in the following lines.

Page 5, Line 9; Page 5, Line 13; Page 11, Line 7; Page 12, Line 8.

7) Page 4, line 15. CACC is important to “amplify” the sensory signals, not for adaptation. PDE is also important for adaptation. Also correct page 22, line 1.

Reply:

Thank you for your comment.

We deleted CACC and added the contribution of PDE in those two parts by referencing the following article (Page 4, Line 15; and Page 23, Line 10).

Reference added:

Borisy, F. F. *et al.* Calcium/calmodulin-activated phosphodiesterase expressed in olfactory receptor neurons. *J. Neurosci.* **12**, 915–923 (1992).

8) Page 5, line 5. For those who are not familiar with the mechanosensation in ORNs, it would be more helpful to explain that mechanosensation is mediated by OR, G-protein, AC3, and is important for the sniff-coupled temporal coding.

Reply:

Thank you for your suggestion.

We added information about mechanosensation on cAMP-related signals via OR, G_{off} and ACIII as well as their physiological roles in sniff-coupled temporal coding (Page 5 Lines 4-7).

9) Page 5, line 18; page 10, line 11; and page 11, line 1. It is true that PDE was not so effective in HEK293 cells. However, ORNs express PDEs at a much higher level. The authors cannot comment on the relative contribution of OMP vs PDE based on these experiments. Indeed, the authors did see a difference under PDEi in Fig. 7. Similarly in page 10, line 10, this is not accurate.

Reply:

Thank you for pointing this out.

We have removed the following sentences:

Page 5: “was more effective than the enzymatic elimination of cAMP”

Page 10: “indicating that PDE was not involved in the elimination of cAMP.”

Page 11: “faster than PDE”

Instead, we have modified the related sentences, as shown below, to highlight the role of OMP in rapid buffering and uncoupling signal transduction to desensitize CNG channels, which is operationally independent from cAMP elimination by PDE.

Page 10, Lines 3-4:

The CNG channel currents were sustained for 3 s in the absence OMP (Ctrl; Fig. 3d-g).

Page 10, Lines 12-14:

These results suggest that OMP swiftly eliminates cAMP to deactivate CNG channels and that the cAMP-buffering capacity of OMP depends on the size of the basal cAMP pools, and a substantial population of OMP might be pre-equilibrated.

10) Overall, the writing needs to be significantly improved. There are many inaccurate descriptions (below).

Reply:

We truly appreciate your suggestions on our manuscript.

We followed all your suggestions as described below.

11) Page 15, line 16. This experiment does not demonstrate the sufficiency.

Reply:

Thank you for carefully reading our manuscript. We have concluded this paragraph focusing on haploinsufficiency to avoid silencing of ORNs, and the subheading has been changed as follows:

Page 15, Line 17; Subheading

OMP haploinsufficiency results in a smaller cAMP-buffering capacity

Page 17, Lines 7-9:

The results presented thus far depict a scenario in which the cAMP-buffering capacity of OMP depends on its expression level and provides a cAMP signalling bypass that avoids ORN silencing to ensure resilient olfactory responses under repetitive stimuli.

12) Fig. 4d. It would be more appropriate to label it “cAMP uncaging” rather than “caged cAMP”.

Reply:

Thank you for carefully reading our manuscript.

We have made this change to Fig. 4d according to your suggestion: from “Caged cAMP” to “cAMP uncaging”

13) Page 19, line 4. It is unclear whether OMP-deficient mice are anosmic. Hyposmia would be better.

Reply:

Thank you for your suggestion.

We used the term hyposmia in the new Fig. 8k-m and in all related text on the following pages: Page 20, Lines 13, 15; Page 21, Lines 6, 9; and Page 22, Lines 1, 6.

14) Page 20, line 3. It would be extremely difficult to judge whether mice “hesitated”.

Reply:

Thank you for pointing this out.

Page 21, Line 14

We have removed “with hesitation”.

15) Page 20, line 12. The authors cannot tell whether this is based on the trained memory.

Reply:

Thank you for pointing this out.

Indeed, it is difficult to list all the possible motivations of mice underlying this searching behaviour.

Page 22, Line 3

We have deleted the phrase “the trained memory”.

16) Page 20, line 17. *This is not accurate. Basal sensitivity should increase in theory.*

Reply:

Thank you for pointing this out while comprehensively understanding our results.

We have changed the phrase as follows.

Page 22, Line 9

increased basal sensitivity during olfaction.

17) Page 21, line 13-15. *This is not really demonstrated. Facts and assumptions should be distinguished.*

Reply:

Thank you for pointing this out.

We have changed the phrase.

Page 23, Lines 5,6

OMP could support such phase-dependent odour processing through the cAMP-buffering capacity.

18) Page 24, line 5-8. *There is a leap of logic.*

Reply:

Thank you for pointing this out.

We have separated the related sentences as follows.

Page 25 Lines 10 – 15:

Such phylogenetic conservation of the CNB sites also accentuates the physiological importance of OMP. The maintenance of a resilient neural response by cAMP-buffering should have evolutionary advantages for surviving in nature. For example, odour-source

localization, which was impaired in OMP-KO mice, must be important for searching for food and predators as well as for long-distance migration for spawning.

To conclude our reply, we sincerely thank the reviewer for the comments, which have certainly improved our manuscript.

Reply to Reviewer #3

In new Fig. 4l, a hyperpolarization experiment is introduced, in part, to address my previous concern (a). I do not exactly follow how both are related. In fact, as the authors state (“The recovered firing was rather abrupt, implying the fluctuation in membrane potential (new Fig. 4l). Considering that ORNs have extraordinarily high input impedance (~ giga-ohm), the slight fluctuation of ion channels would lead to drastic changes in the membrane potential. OMP might also contribute to smoothing such variance in the membrane potentials, which needs further investigations in the future.”) these new findings rather add a novel layer of complexity rather than clarifying my point.

Original comment

*a) Fig. 3a: Spontaneous firing frequency (at least in KO mice, but maybe also in Het animals) is **not normally distributed**, indicating that olfactory sensory neurons might represent physiologically different populations that should be analyzed independently. Fig. 3b is a prime example: basal firing rates and quality is already different, making it hard to assess the different **‘response’ patterns**.*

Reply:

We apologize for our confusing remarks in the last reply.

The variations in waveforms by spike sorting and different patterns of firing are shown in **Supplementary Fig. 5a-c**. Amplitudes and shapes changed even at rest during a burst firing (**Supplementary Fig.5b**). When the sorted spikes were ranked by amplitude, we acknowledged that the amplitudes varied (**Supplementary Fig. 5c and legends**) and some spikes were 50% of the maximal amplitudes. We agree with you that separating different response patterns of single-unit activity is difficult.

Since recording from single units was not practical, we avoided expressions and discussions related to single-unit recordings or response patterns. Rather, we analysed the collective firing activity that occurred in a certain time bin (*e.g.*, basal, I, II, III, post) as “averaged changes” in firing frequencies (**Page 11 ~, Text related to Fig. 4**).

This analysis might contain the spikes of one or a few ORNs, but nevertheless, the recordings at least detect the increase (in WT and Het) or decrease (in KO) in ORN

firing under stimulation. We believe that by these experiments, we have succeeded in characterizing the firing patterns of ORNs in relation to the level of OMP in the cell: OMP^(+/+), OMP^(+/-) and OMP^(-/-).

Namely, ORNs with OMP^(+/-) fired resiliently during stimulation; ORNs without OMP rapidly ceased their firing activity on stimulation (Fig. 4k).

Furthermore, in the presence of basal cAMP, OMP^(+/+) ORNs still maintained resilient firing during stimulation, whereas OMP^(+/-) ORNs increased firing instantaneously followed by a decrease (Fig. 5e).

Accordingly, we drew the current conclusion that OMP prevents silencing of ORNs under stimulation (Page 15, Lines 13-15).

*The in vivo olfactory nerve recordings (Fig. 6) describe a technique that the authors apparently established in this manuscript (I do not find references to previously published studies). While it could be a very powerful technique, I feel that reliability and data quality need to be established before drawing any conclusions. The reader needs to be aware of the **S/N** and the **general suitability** of this method. **Positive and negative controls** (e.g., by performing standard adaptation experiments according to conventional experimental paradigms) need to be performed and the results have to be quantified to allow interpretation.*

I am also still somewhat hesitant to support the notion of “chemo-mechanical stimulation”

As I explained before, the authors still appear to “compare apples with pears” (previous concern (d)). I can understand that authors point out that “as this process will require an extensive amount of time, we plan to perform this experiment in future studies.” However, this issue is central to the authors’ claims and either needs to be addressed (could also be done by EOG or whole-cell patch-clamp recordings from isolated neurons) or the authors should limit their report to the mechanical stimulation part.

Original comment

*d) Fig. 3f–h: It is unclear why the authors now switch to mechanical stimulation. While it has been shown that OSNs also function as mechanosensors, this physiological role is much less understood and far from being a universal model for OSN activity. To not compare apples with pears, the authors should rather opt for an **odour stimulation** paradigm. This is essential since the interpretation of their overall finding is currently hampered by the lack of any evidence in response to actual odour stimulation.*

Reply:

Thank you for your comments to improve the experimental stringency of our manuscript.

We added the evaluation of the multiunit recordings based on the following points: signal-to-noise, suitability and controls. In this analysis, we added limonene-odour stimulation to the *in vivo* recording procedures. We believe this addition strengthens our findings regarding odour stimulation.

Let us provide point-to-point explanations to the individual evaluations below.

Multiple Clusters

First, we performed spike clustering on *in vivo* multiunit recordings, which certainly yielded some clusters (Supplementary fig. 7a-c). However, the shapes of the spikes were slightly changed before and during stimulation (Supplementary fig. 7b). Complex action potentials were also present (Supplementary fig. 7d). The widths and rise times of the spikes were also measured, but these features did not identify the characteristics of the clusters (Supplementary fig. 7c). It was difficult to count the number of spikes at a single-spike resolution from these results (Page 17, Lines 17-18 Results, and Page 78, Lines 5-12 Method).

In this study, we were principally interested in the total changes in the averaged firing frequencies of multiple nerves of ORNs. We quantitatively compared the collective spike numbers by detecting all the peaks of spikes larger than the noise level, defined as follows.

S/N and Quantification

We defined the basal noise level as +3 s.d. of all signals based on the consideration below.

On these criteria, each multi-unit recording contained 5~8 action potential clusters (Supplemental Fig.7b). At the lowest estimate to exclude the indistinguishable small clusters, we assumed each record contains 5 discrete clusters of ORNs, each of which has the averaged firing rate of 4 Hz; corresponding to approximately 20 spikes per second. The duration of spikes over the threshold level is approximately 1 ms (Supplementary Fig. 7b). Because a 1-s bin under 10 kHz recording contains 10000 sampling points, there are 10 points corresponding to a single spike of 1 ms duration (each spike has 1 ms duration with 10 points/ms); 20 spikes correspond to approximately 2% of the signals (200 points/10000 points). Conversely, approximately 98% of the signal corresponds to baseline fluctuations below the noise level. The s.d. values of noise were stable before, during and after the stimulation (Supplementary Fig. 7f and Legends).

This 98% exclusion of signals by the 3 s.d. criteria is very close to the general statistical understanding of 99% exclusion of signals as long as data are normally distributed. Accordingly, we adopted the 3 s.d. threshold that should be large enough to eliminate most of the noise fluctuation (Page 78, Lines 12- 15, Page 79, Lines 1- 6, Method).

Based on these principles, all peaks larger than the threshold of +3 s.d. were counted in this study. Complex action potentials that could be separated by a valley less than 50% between adjacent peaks were counted as two peaks, but unseparated peaks by a valley no less than 50% were simply counted as one (Page 78, Lines, 16-18, Method).

Then, we quantified the stimulus-induced changes by comparing the numbers of collective spikes obtained during the 20-s pre-stimulatory period and during the 20-s stimulation (Fig. 6; and Page 79, Lines 2-6, Method).

Although this thresholding procedure may exclude very small spikes overlapping the noise level, we detected hundreds of spikes within 20-s bins the multiunit recording analysis and quantitatively compared the features of the averaged spike numbers as the increase in WT and Het mice or the decrease in KO mice (a histogram in Fig. 6f,g, plots in Fig. 6h, and legends of Fig. 6f-h); these findings were consistent with those obtained from ORNs in the slice preparations (Fig. 4).

General suitability

We also confirmed that the firing activity of olfactory nerve fibres recorded by this technique was stable for at least 3 min, which was sufficiently long and suitable to analyse the effects of stimulation over the pre- and post-stimulatory periods, which were approximately 2 min (Supplementary fig. 7f; and Page 79, Lines 1,2, Method). Stability without stimulation serves as a negative control for the experimental procedures. Therefore, we believe that our comparison of the counted spike numbers is reliable for evaluating the increase or decrease before and during stimulation (Supplementary fig. 7 and legends, and Page 79, Lines 2-6, Method).

Controls (Odour stimulation)

The use of odourant stimulation was also suggested by Reviewer 1, who proposed a modification of the experiments shown in Fig. 6 by adding background odour stimulation. Thus, we added limonene odour to the *in vivo* recording procedures as a positive control (Limonene (+); Fig. 6h).

Air flow stimulation was included as the positive control to the mechanically evoked responses, while the negative control was free of background odours (Air; Fig. 6h).

Thereby, we quantitatively compared the firing changes in response to a background limonene odour, a food odour mixture and mechanical air flow. Limonene odour, food odours or air flow alone increased the firing rate in WT and Het mice but reduced the firing rate in KO mice (Fig. 6h), which is consistent with the findings in the slice preparations (Fig. 4).

We believe that these consistencies are good justification for the use of *in vivo* multiunit recordings for the evaluation of firing activity patterns of ORNs.

By the addition of these experiments incorporating limonene odour stimulation, we rephrased the ‘chemo-mechanical’ stimulation to ‘sensory’ stimulation sections in the revised manuscript. Modifications were made in the following lines: Page 5, Line 9; Page 5, Line 13; Page 11, Line 7; Page 12, Line 8.

Last, we did not include in this revision any experiments on whole-cell or EOG recordings. A whole-cell recording might involve the activity of OR-specifically GFP-labelled OMP(-/-) ORNs with precise identification of ORNs. However, we did not aim to investigate certain odour-specific responses of ORNs, which is beyond the scope of this study, and we believe that the revised conclusion is supported by the multiunit recordings. We also believe that the revised analysis of the *in vivo* nerve layer recordings with odour stimulation are improved according to your suggestions and that these experiments serve as an alternative to EOG recordings of the local field potentials generated in the main olfactory epithelium.

Regarding my previous point (e), the authors argue that “Because ORNs are surrounded by nonfiring supporting cells, multiunit recordings are rarely observed with a patch pipette with a resistance of approximately 5 mOhm. Of course, multiunit recordings occasionally happen, most likely due to the suctioning of one or two axons or somas of adjacent ORNs. The amplitudes are highly different in such cases and can easily be discriminated from other recordings by thresholding.” I do disagree. In the slice preparation, it is relatively easy to record simultaneously from several OSNs, since the soma layer is targeted. Why are the authors hesitant to perform waveform-based spike sorting?

Original comment

e) Fig. 3g: The original trace appears as if this might be a multi-unit recording. Have the authors applied spike-sorting? And, if so, which unit has been analyzed?

Reply:

We apologize for the confusing remarks in the last reply.

Related to our reply to your first comment, we added the spike sorting data related to the recordings from the soma of ORNs with varying amplitudes (Supplementary Fig. 5c). We agree with you that separating different response patterns of single-unit activity is difficult. The data correspond to multiunit recordings.

Moreover, as long as sorting is based on the spike size only, there are confusing observations to multiple unit recordings such as that spike shapes were drastically changed in a burst mode of firing or during stimulation, partly because of adaptation or accumulated inactivation. (Supplementary Fig. 5b). Similar observations have been reported elsewhere (Reference; Connely et al, 2013; Sato and Sorensen, 2018).

Therefore, in this revision, the firing patterns were analysed as the number of spikes in multiunit activity. This point has been described in detail above.

In the present study, we analysed all the detected spikes and simply described the changes in the number of firings of ORNs (Page 12, Lines 6-7, Results).

With this multiunit analysis, we succeeded in elucidating one aspect of OMP^(-/-)-ORNs in that they showed rapid silencing under cAMP overload (Fig. 4b,d,j,l,m).

Therefore, we believe our multiunit analysis was effective even in further elucidating the molecular mechanisms of OMP in cAMP buffering; OMP buffering of cAMP avoided stimulation-induced silencing of ORNs.

In the text, we included a sentence to describe the possible contribution of suctioning adjacent axons on the multiunit recordings (Page 25, Lines 4-6; Supplemental Fig. 5c and legends; and Page 91, Lines 14-17, Methods).

We also acknowledged the heterogeneity of the firing and response patterns of ORNs by adding an explanation that the ORNs in our data did not show a normal distribution of spontaneous firing (Page 11, Line 18 – Page 12, Line 3) with an additional discussion and a reference that even the ORNs that expressed the same odourant receptors show variance in firing patterns (Page 25, Lines 1-8; Discussion).

To conclude our reply, we must mention that we have modified the abstract, conclusion and title of the manuscript to focus on the cAMP buffering activity of OMP, which transiently desensitizes CNG channels by uncoupling signal transduction and avoids the silencing of ORNs via excessive membrane depolarization (Page 1, Title; and Page 3, Abstract).

We sincerely thank you for the comments, which have certainly improved our manuscript.

Reviewers' Comments:

Reviewer #1:

Remarks to the Author:

The newly added data have much improved this work. This is an important finding in the field. I only have some minor comments.

1) There are still some inaccurate and/or imprecise descriptions. Just to point out some examples in the new abstract:

Line 4-6: "The prolonged exposure to cAMP causes adaptation of CNG channels and attenuates neural responses."

This is wrong. The adaptation of CNG channel is mediated by Ca^{2+} -CaM, rather than cAMP. The attenuation of responses in OMP KO is mediated by the inactivation of voltage-gated ion channels, rather than CNG channels.

Line 9: "cAMP-buffering machinery swiftly bypasses surplus cAMP".

The phrase "bypass" is inaccurate. I would simply change it to: "OMP is a major cAMP buffer"

Line 11-12: "which transiently desensitized CNG channel activity~"; Line 16-17. "cAMP buffering by OMP transiently desensitizes CNG channels~".

These are also wrong. OMP does not directly desensitize CNG channels. They merely reduce free cAMP.

I am not pointing out every minor issue, but I recommend the authors to carefully proofread the text throughout to improve accuracy and readability.

2) I still believe that PDEi in the behavioral test is artificial and the newly added Figure 7 is more physiological. I therefore would suggest switching Figures 7 and 8, but it is entirely up to the authors.

Reviewer #3:

Remarks to the Author:

Dear Editor,

In their revised manuscript now entitled "Olfactory marker protein buffers cAMP by direct binding to avoid depolarization-induced silencing of olfactory receptor neurons", Nakashima and colleagues have not (or only partly) addressed two of my major concerns that I raised in my original review. I am referring to the following points:

A) The in vivo olfactory nerve recordings (Fig. 6) describe a technique that the authors apparently established in this manuscript (I do not find references to previously published studies). While it could be a very powerful technique, I feel that reliability and data quality need to be established before drawing any conclusions. The reader needs to be aware of the S/N and the general suitability of this method. Positive and negative controls (e.g., by performing standard adaptation experiments according to conventional experimental paradigms) need to be performed and the results have to be quantified to allow interpretation.

I apologize that, apparently, I failed to clearly point out what I feel is missing. The authors have mistaken my request for more information about "S/N and the general suitability of this method" as a concern about their ability to do spike sorting and stability of spike detection throughout the recordings. I asked, however, about the suitability of the method as a faithful reporter of odor-evoked activity. New supplementary figure 7 shows that 'single unit' activity is stable throughout

the recording with respect to spike waveform. Apparently, eight units were detected. Fig. 6 (where the original traces are presented) now shows that, in KO animals, essentially eight units display reduced firing when compared to pre stimulation periods. By contrast, all units increased firing in WT mice (as assessed by the homogeneous increase across the distribution histogram (Fig. 6f). How is that possible? In other words, is this recording technique painting a picture that corresponds to a physiological setting? To answer this question, the authors would need to run a series of tests (i.e., experiments, not re-analysis of the present recordings) that validate the method. I am aware that this is (too) time-consuming, but without such controls this approach (in my view) is still in a preliminary state and, as such, it cannot substantiate the authors claim(s). That said, my second concern becomes critical. If addressed, the above issue would also be solved.

B) I am also still somewhat hesitant to support the notion of “chemo-mechanical stimulation.” As I explained before, the authors still appear to “compare apples with pears” (previous concern (d)). I can understand that authors point out that “as this process will require an extensive amount of time, we plan to perform this experiment in future studies.” However, this issue is central to the authors’ claims and either needs to be addressed (could also be done by EOG or whole-cell patch-clamp recordings from isolated neurons) or the authors should limit their report to the mechanical stimulation part.

Here, the authors explain that “Last, we did not include in this revision any experiments on whole-cell or EOG recordings. A whole-cell recording might involve the activity of OR-specifically GFP-labelled OMP(-/-) ORNs with precise identification of ORNs. However, we did not aim to investigate certain odour-specific responses of ORNs, which is beyond the scope of this study, and we believe that the revised conclusion is supported by the multiunit recordings. We also believe that the revised analysis of the in vivo nerve layer recordings with odour stimulation are improved according to your suggestions and that these experiments serve as an alternative to EOG recordings of the local field potentials generated in the main olfactory epithelium.”

We are somewhat going round in circles. The authors believe that this concern is disposed / mitigated by the data shown in Fig. 6 (and suppl. Fig. 7) and I do not concur. The easiest solution would be to perform EOG recordings (e.g., in collaboration with another group that routinely does such experiments – the Zhao lab, the Zufall group, etc.). Why the authors are reluctant is honestly beyond me. This is a standard and reliable technique that could solve the issue at hand within a few weeks. Crossing mOR-EG or another defined OR-GFP line with the OMP KO strain for whole-cell recordings is truly more time-consuming, but it is also doable (animals can be purchased from JAX or provided by several labs in Japan if necessary). Since these approaches are the state-of-the-art, the authors should follow either of the two strategies.

Reviewer #4:

Remarks to the Author:

I have read the previous reviews, the responses from the authors and the current version of the manuscript. As an arbitrating reviewer, my comments are focused primarily on the remaining concerns from Reviewer #2 and the reply to these concerns from the authors.

In general I agree with the concerns of Reviewer #2 with respect to the two points (interpretation of the olfactory nerve recordings and the reliance on mechanosensory responses in the in vivo and ex vivo electrophysiology recordings). As to the first issue, I think there may be some misunderstanding about the supplementary figure 7 and its support of the method, as well as interpretation of Figure 6. An important clarifying point is that I believe the authors use 'spike' frequency, NOT amplitude, as their activity measure; the y-axis label in Figure 6f makes that clear ('spikes per 20 sec'), although breaking these up into a histogram of 'amplitudes' I think

introduces a crucial point of confusion. While supplementary Figure 7 talks about spike waveform shapes and amplitudes, as these are compound action potentials from a recording of many small-diameter axons evoked by an asynchronous stimulus (odor or airpuff), amplitude has little meaning here.

Thus, as I understand it, the important point is that this is a multi-unit spiking measurement and the authors are using the frequency of such events as a measure of overall olfactory nerve output. My concern is that, given the large number of heterogeneous events apparent in the raw traces, and the low S/N of this measurement, there is less confidence about how robust these measurements are - in this respect I agree with Reviewer # 2's concern. I would have more faith in the results if the analysis had been performed blind to genotype, but I could not find this stated anywhere. At the same time, while a weakness, I did not think that this result was crucial to the main conclusions of the paper.

With respect to the second concern, I also agree that it is a limitation to use air puff-driven responses as a proxy for odor-evoked activity in interpreting the effects in the OMP-KO mice. Aside from potential differences in transduction pathway, the kinetics of these different responses may be quite distinct - in addition, the degree to which airpuff stimulation even mimics any mechanosensory stimulation that may occur during breathing or sniffing *in vivo* remains unclear (in my opinion, at least). I was surprised that odorant stimulation was not used in the cell-attached recordings presented in Figures 4 and 5, which would be a stronger result in my opinion.

As to the difficulty in addressing these concerns: I agree with the authors that EOG recordings would not be appropriate given their proposed mechanism of OMP preventing spike adaptation. But I would find additional cell-attached spike recordings using odorants in the *ex vivo* preparation to be convincing, and should not be too difficult to obtain. Even more convincing would be to image calcium activity from ORN axon terminals *in vivo*, although generating the appropriate crosses would be quite time-consuming.

Overall, I feel that these experiments as presented are legitimate limitations in the paper, and should be qualified as such, but also that the key conclusions as outlined in the title and abstract are well supported by the preceding results.

REPLY TO REVIEWERS

To Reviewer #1

Thank you for your tremendous support, which has helped improve our manuscript. We are appreciative of the straightforward suggestions.

Comment 1

1) There are still some inaccurate and/or imprecise descriptions. Just to point out some examples in the new abstract:

Line 4-6: “The prolonged exposure to cAMP causes adaptation of CNG channels and attenuates neural responses.”

This is wrong. The adaptation of CNG channel is mediated by Ca²⁺-CaM, rather than cAMP. The attenuation of responses in OMP KO is mediated by the inactivation of voltage-gated ion channels, rather than CNG channels.

Line 9: “cAMP-buffering machinery swiftly bypasses surplus cAMP”.

The phrase “bypass” is inaccurate. I would simply change it to: “OMP is a major cAMP buffer”

Line 11-12: “which transiently desensitized CNG channel activity~”; Line 16-17. “cAMP buffering by OMP transiently desensitizes CNG channels~”.

These are also wrong. OMP does not directly desensitize CNG channels. They merely reduce free cAMP.

I am not pointing out every minor issue, but I recommend the authors to carefully proofread the text throughout to improve accuracy and readability.

Reply:

Thank you for your careful reading of our manuscript.

We have modified the phrases and terms in the manuscript as follows.

ABSTRACT

Sensory cells resiliently respond to external stimuli. Olfactory receptor neurons (ORNs) use odour-induced intracellular cAMP surge to gate cyclic nucleotide-gated

nonselective cation (CNG) channels in the limited ciliary space. Prolonged exposure to cAMP causes calmodulin-dependent feedback adaptation of CNG channels and attenuates neural responses. On the other hand, the odour-source searching behaviour of animals requires ORNs to be sensitive to odours when approaching targets. How ORNs accommodate these conflicting aspects of cAMP responses remains unknown. Here, we discover that olfactory marker protein (OMP) is a major cAMP buffer that maintains the sensitivity of ORNs. Upon the application of sensory stimuli, OMP directly captured and swiftly reduced freely available cAMP, which transiently uncoupled downstream CNG channel activity and prevented persistent depolarization. Under repetitive stimulation, *OMP*^{-/-} ORNs were immediately silenced after burst firing due to sustained depolarization and inactivated firing machinery. Consequently, *OMP*^{-/-} mice showed serious impairment during odour-source searching tasks. Therefore, cAMP buffering by OMP is a novel mechanism for maintaining resilient firing of ORNs.

Other changes

The last paragraph of the **Introduction** was shortened (Page 5, Lines 69-75).

We also changes terms and phrases to improve the readability in the following pages (highlighted in the manuscript):

Page 4,

Page 5,

Page 10,

Page 11,

Page 13

Page 17

Pages 20-23

Page 31

Page 47

Page 57

Pages 79-83

Page 80

Page 87

Comment 2

2) I still believe that PDEi in the behavioral test is artificial and the newly added Figure 7 is more physiological. I therefore would suggest switching Figures 7 and 8, but it is entirely up to the authors.

Reply:

Thank you for your comments. We truly appreciate your suggestions.

We showed the physiological findings (Figure 7) and the intervention experiments (Figure 8). Therefore, Figure 8 could be considered a perspective for future studies, as the PDEi could have side effects. However, Figure 8 was designed to eliminate the visual tasks to highlight the olfactory ability alone. Thus, we have kept the same figure sequence.

Nonetheless, we added and modified the discussion to describe the possibility of unwanted PDEi effects (Page 27, Lines 460-466).

Additional experiments

We also added some experiments in Fig.4 and Fig.5 by using odourants to stimulate ORNs as suggested by Reviewer #4 and Reviewer#3.

To further demonstrate the physiological properties of OMP-KO neurons towards odourant application, we have included additional experiments to investigate odourant-induced firing changes in OMP-KO neurons (Text, Pages 12-13, Lines 192-209; Fig. 4b-d, Supplementary Fig. 5d, e; and Legends; Method, Page 97).

To compare with previous studies, we also added *in vivo* EOG recording experiments (Text, Page 12-13, Lines 200 and 206; Supplementary Fig. 5d,e; and Legends;).

We added comments on the previous EOG in the Discussion (Text, Page 24, Lines 406-413; Method, Page 83).

We also added experiments to investigate haploinsufficiency using odourants (Text; Page 18, Lines 292-297; Fig. 5f-h; Supplementary Fig. 6g-j; and Legends).

We believe these experiments have reinforced the conclusions.

Finally, we sincerely thank you for your suggestions and support throughout the revision processes.

Noriyuki Nakashima
Kurume University

Reviewer #3

Dear Reviewer,

Thank you for your sincere comments.

Comments

A)

The in vivo olfactory nerve recordings

...this approach (in my view) is still in a preliminary state and, as such, it cannot substantiate the authors claim(s). That said, my second concern becomes critical. If addressed, the above issue would also be solved.

B)

... this issue is central to the authors' claims and either needs to be addressed (could also be done by EOG or whole-cell patch-clamp recordings from isolated neurons) or the authors should limit their report to the mechanical stimulation part.

Reply:

We apologize that we did not fully address your suggestions in the previous revision.

We have included additional experiments using odourants in the main experiments to supplement the nerve recording experiments.

To directly substantiate the odourant-induced changes in the firing properties of OMP-KO neurons, we added extracellular recording from the semi-intact ORNs *ex vivo* using odourant mixtures (Text, Pages 12-13, Lines 192-209; Figure 4c,d; and Legends; Method, Page 97)

We also added IBMX and rolipram challenge experiments to ensure comparability with the mechanically induced responses of OMP-KO neurons (Text, Page 18, Lines 292-297; Figure 5f-h; Supplementary Fig. 6g-j; and Legends).

Finally, we also added EOG recordings for comparison with the extracellular recordings (Supplementary Fig. 5d, e; and Legends).

Odourant-induced stimulation did not completely cause silencing in OMP-KO neurons, most likely due to the unmatched odourant-receptor spectrum. However, the trend in the

firing changes was similar to that observed under pharmacological cAMP overload and mechanical stimulation. Accordingly, we added an evaluation of these results to the Discussion section (Page 24, Lines 414-419).

We believe these additional experiments have reinforced the logic and substantiated the context from the viewpoint of odourant sensation.

Thank you for your comments and suggestions for improving our manuscript.

Sincerely,

Noriyuki Nakashima
Kurume University

Reviewers' Comments:

Reviewer #4:

Remarks to the Author:

The addition of new experiments using odor stimulation to activate OSNs during repetitive stimulation is important, and the results - while not as compelling as with the mechanical stimulation - are consistent with the authors' model regarding the function of OMP. Thus I find these revisions to generally well-enough address the previous comments from the reviewers.

RESPONSE TO REVIEWER

Reviewer #4 (Remarks to the Author):

The addition of new experiments using odor stimulation to activate OSNs during repetitive stimulation is important, and the results - while not as compelling as with the mechanical stimulation - are consistent with the authors' model regarding the function of OMP. Thus I find these revisions to generally well-enough address the previous comments from the reviewers.

Reply:

Dear Reviewer #4

Thank you for your arbitration on the final reviewing process.

We show our cordial gratitude to you.

Noriyuki Nakashima

Kurume University